# Canopy-scale biophysical controls of transpiration and evaporation in the Amazon Basin

Kaniska Mallick[1], Ivonne Trebs[1], Eva Boegh[2], Laura Giustarini[1], Martin Schlerf[1], Darren T. Drewry[3], Lucien Hoffmann[1], Celso von Randow[4], Bart Kruijt[5], Alessandro Araùjo[6], Scott Saleska[7], James R. Ehleringer[8], Tomas F. Domingues[9], Jean Pierre H. B. Ometto[4], Antonio D. Nobre[4], Osvaldo Luiz Leal de Moraes[10], Matthew Hayek[11], J. William Munger[11], Steve Wofsy[11]

[1]Department of Environmental Research and Innovation, Luxembourg Institute of Science and Technology (LIST), Belvaux, Luxembourg

[2]Department of Environmental, Social and Spatial Change, Roskilde University, Roskilde, Denmark

[3]Jet Propulsion Laboratory, California Institute of Technology, 4800 Oak Grove Drive, Pasadena, 91109, USA

[4]Instituto Nacional de Pesquisas Espaciais (INPE), Centro de Ciência do Sistema Terrestre, São José dos Campos, SP, Brazil

[5]Alterra, Wageningen University and Research Centre, Wageningen, The Netherlands

[6]Empresa Brasileira de Pesquisa Agropecuária (EMBRAPA), Belém-PA, Brazil

[7]Department of Ecology and Evolutionary Biology, University of Arizona, Tucson, AZ, USA

[8]Department of Biology, University of Utah, Salt Lake City, UT, USA

[9]Faculdade de Filosofia Ciências e Letras de Ribeirão Preto, Universidade de São Paulo (USP), São Paulo, SP, Brazil

[10]Centro Nacional de Monitoramento e Alertas de Desastres Naturais, SP-RJ, Brazil

[11]Harvard University, Cambridge, MA, USA

**Corresponding Authors:** Kaniska Mallick (Phone: +352 275888425; Email: kaniska.mallick@gmail.com); Ivonne Trebs (Phone: +352 275888880; Email: ivonne.trebs@list.lu)

**Running head:**

Bio-physical controls on evapotranspiration

**Abstract:**

Canopy and aerodynamic conductances ($g_C$ and $g_A$) are two of the key land surface biophysical variables that control the land surface response of land surface schemes in climate models. Their representation is crucial for predicting transpiration ($\lambda E_T$) and evaporation ($\lambda E_E$) flux components of the terrestrial latent heat flux ($\lambda E$), which has important implications for global climate change and water resource management. By physical integration of radiometric surface temperature ($T_R$) into an integrated framework of the Penman-Monteith and Shuttleworth-Wallace model, we present a novel approach to directly quantify the canopy-scale biophysical controls on $\lambda E_T$ and $\lambda E_E$ over multiple plant functional types (PFTs) in the Amazon Basin. Combining data from six LBA (Large-scale Biosphere-Atmosphere Experiment in Amazonia) eddy covariance tower sites and a $T_R$-driven physically-based modeling approach, we identified the canopy-scale feedback-response mechanism between $g_C$, $\lambda E_T$, and atmospheric vapor pressure deficit ($D_A$), without using any leaf-scale empirical parameterizations for the modelling. The $T_R$-based model shows minor biophysical control on $\lambda E_T$ during the wet (rainy) seasons where $\lambda E_T$ becomes predominantly radiation driven and net radiation ($R_N$) determines 75% to 80% of the variances of $\lambda E_T$. However, biophysical control on $\lambda E_T$ is dramatically increased during the dry seasons, and particularly the 2005 drought year, explaining 50% to 65% of the variances of $\lambda E_T$ and indicates $\lambda E_T$ to be substantially soil moisture driven during rainfall deficit phase. Despite substantial differences in $g_A$ between forests and pastures, very similar canopy-atmosphere 'coupling' was found in these two biomes due to soil moisture induced decrease in $g_C$ in the pasture. This revealed the pragmatic aspect of the $T_R$-driven model behavior which exhibits a high sensitivity of $g_C$ to per unit change in wetness as opposed to $g_A$ that is not sensitive to surface wetness variability. Our results reveal the occurrence of a significant hysteresis effect between $\lambda E_T$ and $g_C$ during the dry season for the pasture sites, which is

attributed to relatively low soil water availability as compared to the rainforests, likely due to differences in rooting depth between the two systems. Evaporation was significantly influenced by $g_A$ for all the PFTs and across all wetness conditions. Our analytical framework accurately captures the responses of $g_C$ and $g_A$ to changes in radiation forcings, $D_A$, and surface radiometric temperature, and thus appears to be promising for the improvement of existing land-surface-atmosphere exchange parameterisations across a range of spatial scales.

*Keywords*: Canopy conductance, aerodynamic conductance, transpiration, evaporation, Penman-Monteith, Shuttleworth-Wallace, coupling, Amazon, LBA

# 1 Introduction

The Amazon rainforest is one of the world's most extensive natural ecosystems influencing the Earth's water, energy, and carbon cycles (Malhi et al., 2012), and also a major source of global terrestrial evapotranspiration ($E$) or latent heat flux ($\lambda E$) (Costa et al., 2010; Harper et al., 2014). An intensification of the Amazon hydrological cycle was observed in the past two decades (Cox et al., 2000; Huntingford et al., 2008; Gloor et al., 2013). Recent Amazonian droughts have gained particular attention due to the sensitivity of the tropical forest $\lambda E$ to climate change (Hilker et al., 2014). If persistent precipitation extremes become more prevalent (Hilker et al., 2014); the Amazon rainforest may increasingly become a net source of carbon as a result of both the suppression of net biome exchange by drought and carbon emissions from fires (Gatti et al., 2014). Changes in land cover due to conversion of tropical forest to pastures significantly alters the energy partitioning of the region by decreasing $\lambda E$ and increasing sensible heat fluxes ($H$) over pasture sites (e.g. Priante-Filho et al., 2004). This will ultimately lead to severe consequences for the water balance in the region, with modifications to river discharge already observed in some parts of the Basin (Davidson et al., 2012). Evaluating the $\lambda E$ response to changing climate and land use in the Amazon basin is critical to understand the stability of the tropics within the Earth system (Lawrence and Vandecar, 2015). The control of $\lambda E$ can be viewed as complex supply-demand interactions, where net radiation and soil moisture represent the supply and the atmospheric vapor pressure deficit represents the demand. This supply-demand interaction accelerates the biophysical feedbacks in $\lambda E$ and understanding these biophysical feedbacks is necessary to assess the terrestrial biosphere response to water availability. Therefore, quantifying the critical role of biophysical variables on $\lambda E$ will add substantial insight to assessments of the resilience of the Amazon basin under global change.

The aerodynamic and canopy conductances ($g_A$ and $g_C$, hereafter) (unit m s$^{-1}$) are the two
most important biophysical variables regulating the evaporation ($\lambda E_E$) and transpiration ($\lambda E_T$)
flux components of $\lambda E$ (Monteith and Unsworth, 2008; Dolman et al., 2014; Raupach, 1995;
Colaizzi et al., 2012; Bonan et al., 2014). While $g_A$ controls the bulk aerodynamic transfer of
energy and water through the near-surface boundary layer, $g_C$ represents the restriction on
water vapour flow through the aggregated conductance from stomata of the leaves, in case of
a vegetated land surface. In case of partial vegetation cover, $g_C$ also includes soil surface
conductance for evaporation. At small $g_C/g_A$ ratio, the vapor pressure deficit close to the
canopy source/sink height ($D_0$) approximates the atmospheric vapor pressure deficit ($D_A$) due
to aerodynamic mixing and/or low transpiration. These results in a strong canopy-atmosphere
coupling and such conditions are prevalent under soil moisture deficits. On the contrary, large
$g_C/g_A$ ratio influences the gradients of vapor pressure deficit just above the canopy, such that
$D_0$ tend towards zero and thus remains different from $D_A$ (Jarvis and McNaughton, 1986).
This situation reflects a weak canopy-atmosphere coupling and such situations prevail under
predominantly wet conditions and/or poor aerodynamic mixing due to wetness induced low
aerodynamic roughness. The Penman-Monteith (PM) equation is a physically-based scheme
for quantifying such biophysical controls on canopy-scale $\lambda E_E$ and $\lambda E_T$ from terrestrial
ecosystems, treating the vegetation canopy as a 'big-leaf' (Monteith, 1965; 1981). Despite its
development based on biophysical principles controlling water vapour exchange, quantifying
the $g_A$ and $g_C$ controls on $\lambda E$ through the PM equation suffers from the continued
longstanding uncertainty over the aggregated stomatal and aerodynamic behaviour within the
soil-plant-atmosphere-continuum (Matheny et al., 2014; Prihodko et al., 2008).
One of the major sources of uncertainties in modeling $g_A$ is associated with the empirical (and
uncertain) parameterizations of near-surface boundary layer dynamics, which is invariably
confounded by space-time variability in atmospheric stability (van der Tol et al., 2009;

Shuttleworth, 1989; Gibson et al., 2011). For example, Monin-Obukhov Similarity Theory (MOST) used for $g_A$ modeling appears to be only valid over uniform, extensive, and flat surfaces (Monteith and Unsworth, 2008; van der Tol et al., 2009; Holwerda et al., 2012), and its application to complex 'real' canopy systems is problematic due to chaotic interactions between turbulence, canopy roughness and topography (Raupach and Finnigan, 1995; Shuttleworth, 2007; Holwerda et al., 2012). Similarly, $g_C$ varies in space and time due to variations in plant species, photosynthetic capacity, soil moisture variability and environmental drivers (Monteith and Unsworth, 2008; van der Tol et al., 2009). Despite the existence of several semi-mechanistic and empirical parameterisations for $g_C$ (e.g. Ball et al., 1987; Leuning, 1995; Tuzet et al., 2003; Medlyn et al., 2011), the adaptive tendencies of plant canopies severely compromises the efficacy of such approaches (Matheny et al., 2014), limiting their applicability over most landscapes. Thus, debate over the most appropriate model of canopy conductance has endured for decades.

Previous studies in the Amazon Basin focused on developing an observational understanding of the biogeochemical cycling of energy, water, carbon, trace gases, and aerosols in Amazonia (Andreae et al., 2002; Malhi et al., 2002; da Rocha et al., 2009), model-based understanding of surface ecophysiological behaviour and seasonality of $\lambda E$ (Baker et al., 2013; Christoffersen et al., 2014), modelling the environmental controls on $\lambda E$ (Hasler and Avissar, 2007; Costa et al., 2010), understanding the seasonality of photosynthesis and of $\lambda E$ (da Rocha et al., 2004; Restrepo-Coupe et al., 2013), and the impact of land use on hydrometeorology (Roy and Avissar, 2002; von Randow et al., 2012). However, the combination of climatic and ecohydrological disturbances will significantly affect stomatal functioning, the partitioning of $\lambda E_E$-$\lambda E_T$ and carbon-water-climate interactions of tropical vegetation (Cox et al., 2000; Mercado et al., 2009). Hence, investigation of the effects of drought and land cover changes on conductances, $\lambda E_E$, and $\lambda E_T$ is topic requiring urgent

attention (Blyth et al., 2010) both because of the cursory way it is handled in current
generation of parametric models (Matheny et al., 2014) and because of the centrality of $g_A$
and $g_C$ in controlling modelled flux behaviours (Villagarcía et al., 2010). The persistent risk
of deforestation is likely to alter the radiation interception, surface temperature, surface
moisture, associated meteorological conditions, and vegetation biophysical states of different
plant functional types (PFTs). Conversion from forest to pasture is expected to change the
$g_C/g_A$ ratio of these ecosystems and impact the evapotranspiration components. Besides
inverting the PM equation using field measurements of $\lambda E$, till date either photosynthesis-
dependent modeling or leaf-scale experiments were performed to directly quantify $g_C$ (Ball et
al., 1987; Meinzer et al., 1993, 1997; Monteith, 1995; Jones, 1998; Motzer et al., 2005).
However, an analytical or physical retrieval for $g_A$ and $g_C$ is required not only to better
understand the role of the canopy in regulating evaporation and transpiration, but to enable a
capability to characterize the conductances using remote observations, across large spatial
domains where in-situ observations are not available. This paper aims to leverage this
emerging opportunity by exploring data from the Large-scale Biosphere-Atmosphere
Experiment in Amazonia (LBA) eddy covariance (EC) observations (e.g., de Gonçalves et
al., 2013; Restrepo-Coupe et al., 2013) using a novel analytical modeling technique, the
Surface Temperature Initiated Closure (STIC) (STIC1.0 and STIC1.1) (Mallick et al., 2014,
2015) in order to quantify the biophysical control on $\lambda E_E$ and $\lambda E_T$ over several representative
PFTs of the Amazon Basin.
STIC provides a unique framework for simultaneously estimating $g_A$ and $g_C$, surface energy
balance fluxes, $\lambda E_E$ and $\lambda E_T$. It is based on finding analytical solutions for $g_A$ and $g_C$ by
physically integrating radiometric surface temperature ($T_R$) information (along with radiative
fluxes, meteorological variables) into the PM model (Mallick et al., 2014, 2015). The direct
estimates of canopy-scale conductances and $\lambda E$ obtained through STIC are independent of
any land surface parameterisation. *This contrasts with the multi-layer canopy models that*
*explicitly parameterize the leaf-scale conductances and perform bottom-up scaling to derive*
*the canopy-scale conductances (Baldocchi et al., 2002; Drewry et al., 2010).* A primary
advantage of the approach on which STIC is based is the ability to directly utilize remotely
sensed $T_R$ to estimate $E$, thereby providing a capability to estimate $E$ over large spatial scales
using a remotely sensed variable that is central to many ongoing and upcoming missions.
This study presents a detailed examination of the performance of STIC to better understand
land-atmosphere interactions in one of the most critical global ecosystems and addresses the
following science questions and objectives:
(1) How realistic are canopy-scale conductances when estimated analytically (or non-
parametrically) without involving any empirical leaf-scale parameterization?
(2) What are the controls of canopy-scale $g_A$ and $g_C$ on evaporation and transpiration in the
Amazon basin, as evaluated using STIC?
(3) How do the STIC-based canopy-scale conductances compare with known environmental
constraints?
(4) Is the biophysical response of $g_C$ consistent with the leaf-scale theory (Jarvis and
McNaughton, 1986; McNaughton and Jarvis, 1991; Monteith, 1995)?
The following section describes a brief methodology to retrieve $g_C$, $g_A$, $\lambda E_E$, and $\lambda E_T$. The
data sources used for the analysis are described after the methodology and will be followed
by a comparison of the results with fluxes derived from EC measurements. A detailed
discussion of the results and potential applicability of the method with implications for global
change research are elaborated at the end. A list of symbols and variables used in the present
study is given in Table 1.

## 2 Methodology

### 2.1 Theory

The retrievals of $g_A$, $g_C$, and $\lambda E$ are based on finding a 'closure' of the PM equation (eqn. 1 below) using the STIC framework (Fig. A1 in Appendix) (Mallick et al., 2015). STIC is a physically-based single-source surface energy balance scheme which includes internally consistent estimation of $g_A$ and $g_C$ (Mallick et al., 2014, 2015). Originally designed for application to thermal remote sensing data from Earth observation sensors, the STIC framework exploits observations of radiative ($T_R$), and environmental variables including net radiation ($R_N$), ground heat flux ($G$), air temperature ($T_A$), relative humidity ($R_H$) or vapor pressure ($e_A$) at a reference level above the surface.

The foundation of the development of STIC is based on the goal of finding an analytical solution of the two unobserved 'state variables' ($g_A$ and $g_C$) in the PM equation while exploiting the radiative ($R_N$ and $G$), meteorological ($T_A$, $R_H$), and radiometric surface temperature ($T_R$) as external inputs. The fundamental assumption in STIC is the first order dependence of $g_A$ and $g_C$ on the aerodynamic temperature ($T_0$) and soil moisture (through $T_R$). This assumption allows a direct integration of $T_R$ into the PM equation while simultaneously constraining the conductances through $T_R$. Although the $T_R$ signal is implicit in $R_N$, which appears in the numerator of the PM equation (eqn. 1), it may be noted that $R_N$ has a relatively weak dependence on $T_R$ (compared to the sensitivity of $T_R$ to soil moisture and $\lambda E$). Given $T_R$ is the direct signature of the soil moisture availability, inclusion of $T_R$ in the PM equation also works to add water-stress controls in $g_C$. Until now the explicit use of $T_R$ in the PM model was hindered due to the unavailability of any direct method to integrate $T_R$ into this model, and, furthermore, due to the lack of physical models expressing biophysical states of vegetation as a function of $T_R$. Therefore, the majority of the PM-based $\lambda E$ modeling approaches strongly rely on surface reflectance and meteorology while exploiting the

empirical leaf-scale parameterisations of the biophysical conductances (Prihodko et al., 2008;
Bonan et al., 2014; Ershadi et al., 2015).
The PM equation is commonly expressed as,

$$\lambda E = \frac{s\phi + \rho c_P g_A D_A}{s + \gamma \left(1 + \frac{g_A}{g_C}\right)} \tag{1}$$

where $\rho$ is the air density (kg m$^{-3}$), $c_P$ is the specific heat of air (J kg$^{-1}$ K$^{-1}$), $\gamma$ is the
psychrometric constant (hPa K$^{-1}$), $s$ is the slope of the saturation vapor pressure versus air
temperature (hPa K$^{-1}$), $D_A$ is the saturation deficit of the air (hPa) or vapor pressure deficit at
the reference level, and $\phi$ is the net available energy (W m$^{-2}$) (the difference between $R_N$ and
$G$). The units of all the surface fluxes and conductances are in W m$^{-2}$ and m s$^{-1}$, respectively.
For a dense canopy, $g_C$ in the PM equation represents the canopy surface conductance.
Although it is not equal to the canopy stomatal conductance, it contains integrated
information of the stomata. For a heterogeneous landscape, $g_C$ in the PM equation is an
aggregated surface conductance containing information on both canopy and soil.
Traditionally, the two unknown 'state variables' in eqn. (1) are $g_A$ and $g_C$, and the STIC
methodology is based on formulating 'state equations' for these conductances that satisfy the
PM model (Mallick et al., 2014, 2015). The PM equation is 'closed' upon the availability of
canopy-scale measurements of the two unobserved biophysical conductances, and if we
assume the empirical models of $g_A$ and $g_C$ to be reliable. However, neither $g_A$ nor $g_C$ can be
measured at the canopy-scale or at larger spatial scales. Furthermore, as shown by some
recent studies (Matheny et al., 2014; van Dijk et al., 2015), a more appropriate $g_A$ and $g_C$
model is currently not available. This implies that a true 'closure' of the PM equation is only
possible through an analytical estimation of the conductances.

## 2.2 State equations


By integrating $T_R$ with standard surface energy balance (SEB) theory and vegetation
biophysical principles, STIC formulates multiple 'state equations' that eliminate the need for
exogenous parametric submodels for $g_A$ and $g_C$, associated aerodynamic variables, and land-
atmosphere coupling. The state equations of STIC are as follows and their detailed
derivations are described Appendix (A1).

$$g_A = \frac{\phi}{\rho c_P \left[ (T_o - T_A) + \left( \frac{e_0 - e_A}{\gamma} \right) \right]} \tag{2}$$

$$g_C = g_A \frac{(e_0 - e_A)}{(e_0^* - e_0)} \tag{3}$$

$$T_o = T_A + \left( \frac{e_0 - e_A}{\gamma} \right) \left( \frac{1 - \Lambda}{\Lambda} \right) \tag{4}$$

$$\Lambda = \frac{2\alpha s}{2s + 2\gamma + \gamma \frac{g_A}{g_C}(1 + M)} \tag{5}$$

Here, $T_0$ is the temperature (°C) at the source/sink height (or at the roughness length ($z_0$) or
in-canopy air stream), $e_0$ is the atmospheric vapor pressure (hPa) at the source/sink height, $e_0^*$
is the saturation vapor pressure (hPa) at the source/sink height, $\Lambda$ is the evaporative fraction
(the ratio of $\lambda E$ and $\phi$), $\alpha$ is the Priestley-Taylor parameter (unitless) (Priestley and Taylor,
1972), and $M$ is a unitless quantity which describes the relative wetness (or moisture
availability) of the surface. $M$ controls the transition from potential to actual evaporation and
hence is critical for providing constraint against which the conductances can be estimated ($M$
estimation is explained in Appendix A2). Given values of $R_N$, $G$, $T_A$, and $R_H$ or $e_A$, the four
state equations (eqn. 2 to 5) can be solved simultaneously to derive analytical solutions for
the four state variables. This also produces a 'closure' of the PM model, which is independent
of empirical parameterizations for both $g_A$ and $g_C$. However, the analytical solution to the
above state equations have four accompanying unknowns; $M$ (surface moisture availability),
$e_0$ (vapor pressure at the source/sink height), $e_0^*$ (saturation vapor pressure at the source/sink
height), and Priestley-Taylor coefficient ($\alpha$), and as a result there are four equations with
eight unknowns. Consequently an iterative solution is needed to determine the four unknown
variables (as described in Appendix A2), which is a further modification of the STIC1.1
framework (Mallick et al., 2015). The present version of STIC is designated as STIC1.2 and
its uniqueness is the physical integration of $T_R$ into a combined structure of the PM and
Shuttleworth-Wallace (SW, hereafter) (Shuttleworth and Wallace, 1985) model to estimate
the source/sink height vapor pressures (Appendix A2). In addition to physically integrating
$T_R$ observations into a combined PM-SW framework, STIC1.2 also establishes a feedback
loop describing the relationship between $T_R$ and $\lambda E$, coupled with canopy-atmosphere
components relating $\lambda E$ to $T_0$ and $e_0$. For estimating $M$, the radiometric surface temperature
($T_R$) is extensively used in a physical retrieval framework, thus treating $T_R$ as an external
input. In eqn. (5), the Priestley-Taylor coefficient ($\alpha$) appeared due to the use of the
Advection-Aridity (AA) hypothesis (Brutsaert and Stricker, 1979) for deriving the state
equation of $\Lambda$ (Supplement S1). However, instead of optimising $\alpha$ as a 'fixed parameter', we
have developed a physical equation of $\alpha$ (eqn. A15 in the Appendix A2) and numerically
estimated $\alpha$ as a 'variable'. The derivation of the equation for $\alpha$ is described in Appendix A2.
The fundamental differences between STIC1.2 and earlier versions are described in Table
(A1).
In STIC1.2, $T_0$ is a function of $T_R$ and they are not assumed equal ($T_0 \neq T_R$). The analytical
expression of $T_0$ is dependent on $M$ and the estimation of $M$ is based on $T_R$. To further
elaborate this point on the inequality of $T_0$ and $T_R$, we show an intercomparison of retrieved
$T_0$ versus $T_R$ for forest and pasture (Fig. A2). This indicates the distinct difference of the
retrieved $T_0$ from $T_R$ for the two different biomes.

**2.3 Partitioning λE**

The terrestrial latent heat flux is an aggregate of both transpiration ($\lambda E_T$) and evaporation ($\lambda E_E$) (sum of soil evaporation and interception evaporation from canopy). During rain events the land surface becomes wet and $\lambda E$ tends to approach the potential evaporation ($\lambda E^*$), while surface drying after rainfall causes $\lambda E$ to approach the potential transpiration rate ($\lambda E_T^*$) in the presence of vegetation, or zero without any vegetation. Hence, $\lambda E$ at any time is a mixture of these two end member conditions depending on the degree of surface moisture availability or wetness ($M$) (Bosveld and Bouten, 2003; Loescher et al., 2005). Considering the general case of evaporation from an unsaturated surface at a rate less than the potential, $M$ is the ratio of the actual to the potential evaporation rate and is considered as an index of evaporation efficiency during a given time interval (Boulet et al., 2015). Partitioning of $\lambda E$ into $\lambda E_E$ and $\lambda E_T$ was performed according to Mallick et al. (2014) as follows:

$$\lambda E = \lambda E_E + \lambda E_T = M \lambda E^* + (1-M) \lambda E_T^* \qquad (6)$$

The estimates of $\lambda E_E$ in the current method consist of aggregated contribution from both 'interception' and 'soil evaporation', and no further attempt is made to separate these two components. In the Amazon forest, 'soil evaporation' has a negligible contribution while the 'interception evaporation' contributes substantially to the total evaporative fluxes, and, therefore the partitioning of $\lambda E$ into $\lambda E_E$ and $\lambda E_T$ is crucial. After estimating $g_A$, $\lambda E^*$ was estimated according to the Penman equation and $\lambda E_T$ was estimated as the residual in eqn. (6).

In this study, we use the term 'canopy conductance' instead of 'stomatal conductance' given the term 'stomata' is applicable at the leaf-scale only. As stated earlier, for a heterogeneous surface $g_C$ should principally be a mixture of the canopy surface (integrated stomatal information) and soil conductances. However, given the high vegetation density of the Amazon Basin, the soil surface exposure is negligible, and, hence we assume $g_C$ to be the

canopy-scale aggregate of the stomatal conductance. Similarly, different $g_A$ exists for soil-
canopy, sun-shade, and dry-wet conditions (Leuning, 1995); which is currently integrated
into a lumped $g_A$ (given the big-leaf nature of STIC). From the big-leaf perspective, it is
generally assumed that the aerodynamic conductance of water vapor and heat are equal
(Raupach, 1998). However, for obtaining partitioned aerodynamic conductances, explicit
partitioning of $\lambda E$ is needed, which is beyond the scope of the current manuscript.
**2.4 Evaluating $g_A$ and $g_C$**
Due to the lack of direct canopy-scale $g_A$ measurements, a rigorous evaluation of $g_A$ cannot be
performed. To evaluate the STIC retrievals of $g_A$ ($g_{A\text{-}STIC}$) we adopted three different methods:
(a) By using the measured friction velocity ($u^*$) and wind speed ($u$) at the EC towers and
using the equation of Baldocchi and Ma (2013) ($g_{A\text{-}BM13}$) in which $g_A$ was expressed as sum of
turbulent conductance and canopy (quasi-laminar) boundary layer conductance as,

$$g_{A\text{-}BM13} = [(u/u^{*2}) + (2/ku^{*2})(S_c/P_r)^{0.67}]^{-1} \qquad (7)$$

where $k$ is von Karman's constant, 0.4; $S_c$ is the Schmidt Number; $P_r$ is the Prandtl Number
and their ratio is generally considered to be unity. Here the conductances of momentum,
sensible and latent heat fluxes are assumed to be identical (Raupach, 1998).
(b) By inverting $\lambda E$ observations for wet conditions hence assuming $\lambda E \cong \lambda E^*$ and estimating
$g_A$ ($g_{A\text{-}INV}$) as,

$$g_{A\text{-}INV} = \gamma \lambda E / \rho c_P D_A \qquad (8)$$

(c) By inverting the aerodynamic equation of $H$ and estimating a hybrid $g_A$ ($g_{A\text{-}HYB}$) from
observed $H$ and STIC $T_0$ as ($T_{0\text{-}STIC}$),

$$g_{A\text{-}HYB} = H / \rho c_P (T_{0\text{-}STIC} - T_A) \qquad (9)$$

Like $g_{A-STIC}$, direct verification of STIC $g_C$ ($g_{C-STIC}$) could not be performed as canopy-scale
$g_C$ observations are not possible with current measurement techniques. Although leaf-scale $g_C$
measurements are relatively straightforward, these values are not comparable to values
retrieved at the canopy-scale. However, assuming $u^*$-based $g_A$ as baseline aerodynamic
conductance, we have estimated canopy-scale $g_C$ by inverting the PM equation ($g_{C-INV}$)
(Monteith, 1995) to evaluate $g_{C-STIC}$ by exploiting $g_{A-BM13}$ in conjunction with the available $\phi$,
$\lambda E$, $T_A$, and $D_A$ measurements from the EC towers.
**2.5 Decoupling coefficient and biophysical controls**
The decoupling coefficient or 'Omega' ($\Omega$) is a dimensionless coefficient ranging from 0.0 to
1.0 (Jarvis and McNaughton, 1986) and considered as an index of the degree of stomatal
control on transpiration relative to the environment. The equation of $\Omega$ is as follows:

$$\Omega = \frac{\frac{s}{\gamma} + 1}{\frac{s}{\gamma} + 1 + \frac{g_A}{g_C}} \tag{10}$$

Introducing $\Omega$ in the Penman-Monteith (PM) equation for $\lambda E$ results in:

$$\lambda E = \Omega \lambda E_{eq} + (1 - \Omega) \lambda E_{imp} \tag{11}$$

$$\lambda E_{eq} = \frac{s\phi}{s + \gamma} \tag{12}$$

$$\lambda E_{imp} = \frac{\rho c_P}{\gamma} g_C D_A \tag{13}$$

Where, $\lambda E_{eq}$ is the equilibrium latent heat flux, which depends only on $\phi$ and would be
obtained over an extensive surface of uniform moisture availability (Jarvis and McNaughton,
1986; Kumagai et al., 2004). $\lambda E_{imp}$ is the imposed latent heat flux, which is 'imposed' by the
atmosphere on the vegetation surface through the effects of vapor pressure deficit (triggered
under limited soil moisture availability) and $\lambda E$ becomes proportional to $g_C$.
When the $g_C/g_A$ ratio is very small (i.e., water stress conditions), stomata principally control
the water loss and a change in $g_C$ will result in a nearly proportional change in transpiration.
Such conditions trigger strong biophysical control on transpiration. In this case the $\Omega$ value
approaches zero and vegetation is believed to be fully coupled to the atmosphere. In contrast,
for a high $g_C/g_A$ ratio (i.e., high water availability), changes in $g_C$ will have little effects on the
transpiration rate, and transpiration is predominantly controlled by $\phi$. In this case the $\Omega$ value
approaches unity, and vegetation is considered to be poorly coupled to the atmosphere.
Given both $g_A$ and $g_C$ are the independent estimates in STIC1.2, the concept of $\Omega$ was used to
understand the degree of biophysical control on $\lambda E_T$, which indicates the extent to which the
transpiration fluxes are approaching the equilibrium limit. However, the biophysical
characterisation of $\lambda E_T$ and $\lambda E_E$ through STIC1.2 significantly differs from previous
approaches (Ma et al., 2015; Chen et al., 2011; Kumagai et al., 2004), and the fundamental
differences are  centered on the specifications of $g_A$ and $g_C$ (as described in Table A2). While
the estimation of $g_A$ in previous approaches was based on $u$ and $u^*$, the estimation of $g_C$ was
based on inversion of observed $\lambda E$ based on the PM equation (e.g. Stella et al., 2013).
However, none of these approaches allow independent quantification of biophysical controls
of $\lambda E$ as $g_C$ is constrained by $\lambda E$ itself.
## 3 Datasets
### 3.1 Eddy covariance and meteorological quantities
We used the LBA (Large-Scale Biosphere-Atmosphere Experiment in Amazonia) data for
quantifying the biophysical controls on the evaporative flux components. LBA was an
international research initiative conducted during 1995-2005 to study how Amazonia
functions as a regional entity within the larger Earth system, and how changes in land use and
climate will affect the hydrological and biogeochemical functioning of the Amazon
ecosystem (Andreae et al., 2002).
A network of eddy covariance (EC) towers was operational during the LBA experiment, such
that data from nine EC towers were obtained from the ORNL Distributed Archive Active
Centre (ftp://daac.ornl.gov/data/lba/carbon_dynamics/CD32_Brazil_Flux_Network/). These
are the quality controlled and harmonized surface flux and meteorological data from the
Brazilian Amazon flux network. Time series of surface fluxes ($\lambda E$, $H$, $G$), radiation ($R_N$,
shortwave and longwave), thermal ($T_R$), meteorological quantities ($T_A$, $R_H$, wind speed) as
well as soil moisture and rainfall were available from six (out of nine) EC towers. Three of
the EC towers had numerous missing data and were not included in the analysis. The surface
energy balance was closed by applying the Bowen ratio (Bowen, 1926) closure as described
in Chavez et al. (2005) and later adopted by Anderson et al. (2007) and Mallick et al. (2015).
In the absence of $G$ measurements, $\phi$ was assumed to be equal to the sum of $\lambda E$ and $H$ with
the assumption that a dense vegetation canopy restricts the energy incident on the soil
surface, thereby allowing us to assume negligible ground heat flux. For the present analysis,
data from six selected EC towers (Table 2) represent two different biomes (forest and
pasture) covering four different PFTs, namely, tropical rainforest (TRF), tropical moist forest
(TMF), tropical dry forest (TDF), and pasture (PAS), respectively. A general description of
the datasets can be found in Saleska et al. (2013). For all sites, monthly averages of the
diurnal cycle (hourly time resolution) were chosen for the present analysis.
**4 Results**
**4.1 Evaluating $g_A$, $g_C$, and surface energy balance fluxes**
Examples of monthly averages of the diurnal cycles of the four different $g_A$ estimates and
their corresponding $g_C$ estimates over two different PFTs (K34 for forest and FNS for
pasture) reveal that $g_{A\text{-}STIC}$ and $g_{C\text{-}STIC}$ tend to be generally higher for the forest than their
counterparts, varying from 0 to 0.06 m s$^{-1}$ and 0 to 0.04 m s$^{-1}$ respectively (Fig. 1a and 1b).
The magnitude of $g_{A\text{-}STIC}$ varied between 0 to 0.025 m s$^{-1}$ for the pasture (Fig. 1a), while $g_{C\text{-}}$
$_{STIC}$ values were less than half of those estimated over the forest (0 – 0.01 m s$^{-1}$) (Fig. 1b).
The conductances showed a marked diurnal variation expressing their overall dependence on
net radiation, vapor pressure deficit, and surface temperature. Despite the absolute differences
between the conductances from the different retrieval methods, their diurnal patterns were
comparable.
The canopy-scale evaluation of $g_{A\text{-}STIC}$ is illustrated in Fig. 2a (and Table 3) combining data
from the four PFTs. Estimated values range between zero and 0.1 m s$^{-1}$ and show modest
correlation (R$^2$ = 0.44) (R$^2$ range between 0.22 [±0.18] to 0.55 [±0.12]) between $g_{A\text{-}BM13}$ and
$g_{A\text{-}STIC}$ with regression parameters ranging between 0.81 (±0.023) and 1.07 (±0.047) for the
slope and 0.0019 (±0.0006) to 0.0006 (±0.0006) m s$^{-1}$ for the offset (Table 3). The root mean
squared deviation (RMSD) varied between 0.007 (TDF) and 0.013 m s$^{-1}$ (TRF). Statistical
comparisons between $g_{A\text{-}STIC}$ and $g_{A\text{-}HYB}$ revealed relatively low RMSD and high correlation
between them (RMSD = 0.007 m s$^{-1}$ and R$^2$ = 0.77) as compared to the error statistics
between $g_{A\text{-}STIC}$ and $g_{A\text{-}INV}$ (RMSD = 0.011 m s$^{-1}$ and R$^2$ = 0.50) (Fig. 2b, 2c). The residuals
between $g_{A\text{-}STIC}$ and $g_{A\text{-}BM13}$ are plotted as a function of $u$ and $u^*$ in Fig. (2d) with the aim to
ascertain whether significant biases are introduced by ignoring wind and shear information
within STIC1.2. As illustrated in Fig. 2d, there appears to be a weak systematic relationship
between the residual $g_A$ difference with either $u^*$ or $u$ ($r$ = -0.26 and -0.17). However, a
considerable relationship was found between wind and shear driven $g_A$ (i.e., $g_{A\text{-}BM13}$) versus $\phi$,
$T_R$ and $D_A$ ($r$ = 0.83, 0.48, and 0.42) (Fig. 2e and 2f), which indicates that these three energy
and water constraints can explain 69%, 23%, and 17% variance of $g_{A\text{-}BM13}$, respectively.
Canopy-scale evaluation of hourly $g_C$ is presented in Fig. 3a (and Table 3) combining data
from the four PFTs. Estimated values range between zero and 0.06 m s$^{-1}$ for $g_{C\text{-}STIC}$ and show
reasonable correlation ($R^2 = 0.39$) ($R^2$ range between 0.14 [±0.04] to 0.58 [±0.12]) between
$g_{C\text{-}STIC}$ and $g_{C\text{-}INV}$ with regression parameters ranging between 0.30 (±0.022) and 0.85
(±0.025) for the slope and 0.0024 (±0.0003) to 0.0097 (±0.0007) m s$^{-1}$ for the offset (Table
3). The RMSD varied between 0.007 (PAS) and 0.012 m s$^{-1}$ (TRF and TDF). Given $g_A$
significantly controls $g_C$, we also examined whether biases in $g_C$ are introduced by ignoring
wind and shear information within STIC. The scatterplots between residual $g_C$ difference ($g_{C\text{-}}$
$_{STIC} - g_{C\text{-}INV}$) versus both $u$ and $u^*$ (Fig. 3b) showed $g_C$ residuals to be evenly distributed
across the entire range of $u$ and $u^*$ and no systematic pattern was evident.
The reliability of STIC1.2-based $g_A$ and $g_C$ retrievals was further verified by evaluating $\lambda E$
and $H$ estimates (Fig. 4). Both the predicted $\lambda E$ and $H$ are generally in good agreement with
the observations, with substantial correlation ($r$) ($R^2$ from 0.61 to 0.94), reasonable RMSD of
33 and 37 W m$^{-2}$, and mean absolute percent deviation (MAPD) of 14% and 32% between
the observed and STIC fluxes (Fig. 4), respectively. Regression parameters varied between
0.96 (±0.008) to 1.14 (±0.010) for the slope and -16 (±2) to -2 (±2) W m$^{-2}$ for the offset for
$\lambda E$ (Table 4), whereas for $H$, these were 0.60 (±0.025) to 0.89 (±0.035) for the slope and 9
(±1) to 29 (±2) W m$^{-2}$ for the offset (Table 3), respectively. The RMSD in $\lambda E$ varied from 20
to 31 W m$^{-2}$ and 23 to 34 W m$^{-2}$ for $H$ (Table 3).
The evaluation of the conductances and surface energy fluxes indicates some efficacy for the
STIC derived fluxes and conductance estimates which represent a weighted average of these
variables over the source area around EC tower.
**4.2 Canopy coupling, transpiration and evaporation**
From Fig. 5a an overall weak to moderate relationship ($r$ = -0.31 to -0.42) is apparent
between the coupling (i.e., 1-$\Omega$) and $\lambda E_T$, where $\lambda E_T$ is negatively related to the coupling for
all the PFTs, thus indicating the influence of weak to moderate biophysical controls on $\lambda E_T$
throughout the year in addition to radiative controls. The biophysical control was
substantially enhanced in TRF ($r$ increased from -0.36 to -0.53 and -0.60) (47 to 67%
increase) and TMF ($r$ increased from -0.31 to -0.53 and -0.58) (70 to 85% increase) during
the dry seasons (July-September) (Fig. 5a). A profound increase of biophysical control on
$\lambda E_T$ during the dry season was also found in TDF (52% increase) and PAS (37% increase)
(Fig. 5a). The negative relationship ($r = -0.29$ to -0.45) between $(1-\Omega)$ and $\lambda E_E$ (Fig. 5b) in all
four PFTs indicated the role of aerodynamic control on $\lambda E_E$. The aerodynamic control was
also enhanced during the dry seasons as shown by the increased negative correlation ($r = -$
$0.50$ to -0.69) (Fig. 5b) between $(1-\Omega)$ and $\lambda E_E$.
Illustrative examples of the diurnal variations of $\lambda E_E$, $\lambda E_T$, and $\Omega$ for two different PFTs with
different annual rainfall (2329 mm in rainforest, K34 and 1597 mm in pasture, FNS) for three
consecutive days during both dry and wet seasons are shown in Fig. 5c to 5f. This shows
morning rise of $\Omega$ and a near-constant afternoon $\Omega$ in the wet season (Fig. 5c and 5d), thus
indicating no biophysical controls on $\lambda E_E$ and $\lambda E_T$ during this season. On the contrary, during
the dry season, the morning rise in $\Omega$ is followed by a decrease during noontime (15% to 25%
increase in coupling in forest and pasture) (Fig. 5e and 5f) due to dominant biophysical
control, which is further accompanied by a transient increase from mid-afternoon till late
afternoon and steadily declined thereafter. Interestingly, coupling was relatively higher in
pasture during the dry seasons and the reasons  are detailed in the following section and
discussion.
**4.3 $g_C$ and $g_A$ versus transpiration and evaporation**
Scatter plots between $\lambda E_T$ and $\lambda E_E$ versus $g_C$ and $g_A$ showed a triangular pattern which
became wider with increasing conductances (Fig. 6). To explain this behaviour of $\lambda E_T$ versus
$g_C$ and $g_A$, we further examined the entire mechanism of conductance-$\lambda E_T$ interactions
through two dimensional scatters between $\lambda E_T$ and conductances for two consecutive diurnal
cycles during wet and dry seasons over rainforest and pasture sites with different annual
rainfall (e.g., K34 as wet and FNS as dry site, annual rainfall 2329 mm and 1597 mm) (Fig.
7). Our results confirm the occurrence of diurnal hysteresis between $g_C$-$g_A$ and $\lambda E_T$ and
explain the reason for the shape of the curves obtained in Fig. 6. During the wet season, a
distinct environmental control is detectable on $g_C$ and $\lambda E_T$ in the morning hours (Fig. 7a and
7b) in both PFTs where $g_C$ and $\lambda E_T$ increased as a result of increasing $R_N$, $T_R$, and $D_A$. From
the late morning to afternoon, a near-constant (forest) or negligible increase (pasture) of $\lambda E_T$
is observed despite substantial reduction of both $g_C$ and $g_A$ (25 to 50% decrease), after which
$\lambda E_T$ starts decreasing. This behaviour of $\lambda E_T$ was triggered due to the concurrent changes in
$R_N$ (15 to 50% change), $D_A$ (20 to 60% change) and surface temperature ($T_R$) (5% to 14%
change), which indicates the absence of any dominant biophysical regulation on $\lambda E_T$ during
the wet season (Fig. 7a and 7b). On the contrary in the dry season, although the morning rise
in $\lambda E_T$ is steadily controlled by the integrated influence of environmental variables, but a
modest to strong biophysical control is found for both PFTs during the afternoon where $\lambda E_T$
substantially decreased with decreasing conductances (Fig. 7c and 7d). This decrease in $\lambda E_T$
is mainly caused by the reduction in $g_C$ as a result of increasing $D_A$ and $T_R$ (as seen later in
Fig. 8a and 8c). In the dry season, the area under the hysteretic relationship between $\lambda E_T$, $g_C$
and environmental variables was substantially wider in pasture (Fig. 7d) than for the
rainforest (Fig. 7c), which is attributed to greater hysteresis area between $R_N$ and $D_A$ in
pasture as a result of reduced water supply. The stronger hysteresis effects in pasture during
the dry season (Fig. 7d) ultimately led to the stronger relationship between coupling and $\lambda E_T$
(as seen in Fig. 5a).

## 4.4 Factors affecting variability of $g_C$ and $g_A$

The sensitivity of stomatal conductance to vapor pressure deficit is a key governing factor of transpiration (Ocheltree et al., 2014; Monteith, 1995). We examined if the feedback or feed-forward response hypothesis (Monteith, 1995; Farquhar, 1987) between $g_C$, $D_A$, and $\lambda E_T$ is reflected in our canopy-scale $g_C$ retrievals. Combining data of all PFTs, we found an exponential decline of $g_C$ in response to increasing $D_A$ regardless of the variations of net radiation (Fig. 8a). High $g_C$ is consistent with high humidity and low evaporative demand. Five negatively logarithmic scatters fit the data with $r$ values of 0.38 ($0< R_N <150$ W m$^{-2}$), 0.63 ($150< R_N <300$ W m$^{-2}$), 0.73 ($300< R_N <450$ W m$^{-2}$), 0.78 ($450< R_N <600$ W m$^{-2}$), and 0.87 ($R_N >600$ W m$^{-2}$). The sensitivity of $g_C$ to $D_A$ was at the maximum in the high $R_N$ range beyond 600 W m$^{-2}$ and the sensitivity progressively declined with declining magnitude of $R_N$ ($0 - 150$ W m$^{-2}$).

Scatter plots between $g_C$ and $\lambda E_T$ for different levels of $D_A$ revealed a linear pattern between them for a wide range of $D_A$ ($20> D_A >0$ hPa) (Fig. 8b). Following Monteith (1995), isopleths of $R_N$ are delineated by the solid lines passing through $\lambda E_T$ on the x-axis and through $g_C$ on the y-axis. Isobars of $D_A$ (dotted lines) pass through the origin because $\lambda E_T$ approaches zero as $g_C$ approaches zero. Figure (8b) shows substantial reduction of $g_C$ with increasing $D_A$ without any increase of $\lambda E_T$, like an inverse hyperbolic pattern to $D_A$ (Monteith 1995; Jones, 1998). For all the PFTs, an active biological (i.e., stomatal) regulation maintained almost constant $\lambda E_T$ when $D_A$ was changed from low to high values (Fig. 8b). At high $D_A$ (above 10 hPa), after an initial increase of $\lambda E_T$ with $g_C$, $g_C$ approached a maximum limit and remained nearly independent of $\lambda E_T$ (Fig. 8b). Among all the $D_A$ levels, the maximum control of $g_C$ on $\lambda E_T$ variability (62 to 80%) was found at high atmospheric water demand (i.e., 30 hPa$>D_A>20$ hPa). The scatter plots between $g_C$ and $T_R$ (Fig. 8c) for different levels of $D_A$ revealed an exponential decline in $g_C$ with increasing $T_R$ and atmospheric water demand.

When retrieved $g_A$ was plotted against the radiometric surface temperature and air
temperature difference $(T_R - T_A)$, an exponential decline in $g_A$ was found in response to
increasing $(T_R - T_A)$ (Fig. 8d). High $g_A$ is persistent with low $(T_R - T_A)$ irrespective of the
variations in $R_N$ (with the exception of very low $R_N$). Four negatively logarithmic scatters fit
$g_A$ versus $(T_R - T_A)$ relationship with $r$ values of 0.28 ($150< R_N <300$ W m$^{-2}$), 0.55 ($3000< R_N$
$<450$ W m$^{-2}$), 0.64 ($450< R_N <600$ W m$^{-2}$), and 0.77 ($R_N >600$ W m$^{-2}$).

## 5 Discussion

### 5.1 Evaluating $g_A$, $g_C$, and surface energy balance fluxes

The aerodynamic conductance retrieved with STIC1.2 showed acceptable correlation and
valid estimates of $g_A$ when compared against an empirical model that uses $u^*$ and $u$ to derive
$g_A$ (Fig. 1 and 2a) and two other inversion/hybrid-based $g_A$ estimates. The differences
between $g_{A\text{-}STIC}$ and $g_{A\text{-}BM13}$ were mainly attributed to the structural differences and empirical
nature of the parameterization for the near-surface boundary layer conductance
$((2/ku^{*2})(S_c/P_r)^{0.67})$ in $g_{A\text{-}BM13}$, which results in some discrepancies between $g_{A\text{-}STIC}$ and $g_{A\text{-}BM13}$
particularly in the pasture (Fig. 2a). The extent to which the structural discrepancies between
$g_{A\text{-}STIC}$ and $g_{A\text{-}BM13}$ relate to actual differences in the conductances for momentum vs. heat is
beyond the scope of this manuscript, and a detailed investigation using data on atmospheric
profiles of wind speed, temperature etc. are needed to actually quantify such differences.
Momentum transfer is associated with pressure forces and not identical to heat and mass
transfer (Massman, 1999). In STIC1.2, $g_A$ is directly estimated and is a robust representative
of the conductances to heat/water vapor transfer; whereas $g_{A\text{-}BM13}$ estimates based on $u^*$ and $u$
is more representative for the momentum transfer. Therefore, the difference between the two
different $g_A$ estimates (Fig. 2) can be largely attributed to the actual difference in the
conductances for momentum and heat/water vapor. The turbulent conductance equation
$(u^{*2}/u)$ in $g_{A\text{-}BM13}$ is also very sensitive to the uncertainties in the sonic anemometer
measurement (Contini et al., 2006; Richiardone et al., 2012). However, the evidence of a
weak systematic relationship between the $g_A$ residuals and $u$ (Fig. 2d) and capability of the
thermal ($T_R$), radiative ($\phi$), and meteorological ($T_A$, $D_A$) variables in capturing the variability
of $g_{A\text{-}BM13}$ (Fig. 2e and 2f) indicates the diagnostic potential $g_{A\text{-}STIC}$ estimates to explain the
wind driven $g_A$ variability. Excluding $u$ might introduce errors in cases where wind is the
only source of variations in $g_A$ and surface fluxes (Mallick et al., 2015). In general, the
accuracies in commonly used parametric $g_A$ estimates based on $u$ and surface roughness
parameters several meters distant from canopy foliage are limited due to the uncertainties
concerning the attenuation of $u$ close to the vegetation surface (Meinzer et al., 1997; Prihodko
et al., 2008). The magnitude of $u$ near the foliage can be substantially lower than that
measured considerably away at some reference location above or within the canopy (Meinzer
et al., 1997). Notwithstanding the inequalities of $g_A$ estimated with different methods, it is
challenging to infer the accuracy of the different estimates. It is imperative to mention that $g_A$
is one of the main anchors in the PM-SW model because it not only appears in the numerator
and denominator of these models, $g_A$ also provides feedback to $g_C$, $T_0$, and $D_0$ (seminal paper
of Jarvis and McNaughton, 1986). Therefore, the estimates of $\lambda E$ in the PM-SW framework
are very sensitive to parameterization of $g_A$ and stable $\lambda E$ estimates might be possible if $g_A$
estimation is unambiguous (Holwerda et al., 2012; van Dijk et al., 2015). Given the lack of
consensus in the community on the 'true' $g_A$ and from the nature of surface flux validation
results (Fig. 4) it appears that $g_{A\text{-}STIC}$ tends to be the appropriate aerodynamic conductance
that satisfies the PM-SW equation. Discrepancies between $g_{C\text{-}STIC}$ and $g_{C\text{-}INV}$ originated from
the differences in $g_A$ estimates between the two methods.
Despite the good agreement between the measured and predicted $\lambda E$ and $H$ (Fig. 4, Table 4),
the larger error in $H$ was associated with the higher sensitivity of $H$ to the errors in $T_R$ (due to
poor emissivity correction) (Mallick et al., 2015). Since the difference between $T_R$ and $T_A$ is
considered to be the primary driving force of $H$ (van der Tol et al., 2009), the modelled errors
in $H$ are expected to arise due to the uncertainties associated with $T_R$.
**5.2 Canopy coupling, $g_c$ and $g_A$ versus transpiration and evaporation**
The correlation analysis between 1-$\Omega$ and $\lambda E_T$ revealed the extent of biophysical and
radiative controls on $\lambda E_T$ (Fig. 5). The degree of biophysical control is a function of the ratio
of $g_C$ to $g_A$. Minor biophysical control on $\lambda E_T$ was apparent for forest and pasture during the
wet seasons (Fig. 5c and 5d) as a result of a high $g_C/g_A$ ratio along with increasing $\lambda E_T$. Such
conditions stimulate local humidification of air surrounding the canopy and uncoupling of the
in-canopy vapor pressure deficit ($D_0$) from that in the air above (i.e., $D_0 < D_A$) (Meinzer et al.,
1997; Motzer et al., 2005) (Fig. 9a), which implies that $\lambda E_T$ becomes largely independent of
$g_C$. On the contrary, an enhanced biophysical control on $\lambda E_T$ was apparent during the dry
season and drought year 2005 during the period of reduced water supply particularly over
PAS (Fig. 5e, 5f, and 7). Such condition leads to a relatively dry canopy surface, and
substantially high $g_A$ compared to $g_C$, thus resulting in low $g_C/g_A$ ratios regardless of their
absolute values (Meinzer et al., 1993; McNaughton and Jarvis, 1991). Here, fractional change
in $g_C$ results in an equivalent fractional change in $\lambda E_T$. This impedes transpiration from
promoting local equilibrium of $D_0$ and minimizing (or maximizing) the gradient between $D_0$
and atmospheric vapor pressure deficit ($D_A$) (i.e., $D_0 \cong D_A$ or $D_0 > D_A$) (eqn. A10) (Fig 9a),
thereby resulting in strong coupling between $D_0$ and $D_A$ (Meinzer et al., 1993; Jarvis and
McNaughton, 1986). Besides, a supplemental biophysical control on $\lambda E_T$ might have been
imposed as a consequence of a direct negative feedback of $D_A$ and $D_0$ on $g_C$ (McNaughton
and Jarvis, 1991; Jarvis, 1986). Increase in $D_A$ (or $D_0$) beyond a certain limit decreases $g_C$
(Fig. 7 and 8), resulting in a low and narrow increase of $\lambda E_T$, despite steady increase in $g_A$ and
$R_N$. The combination of negative feedback response between $D_A$ and $g_C$ with the overall
radiative-aerodynamic coupling significantly dampens the variation of transpiration in PAS
and TDF in the dry season, thus featuring increased biophysical control in these PFTs. These
results are in agreement with von Randow et al. (2012), who found enhanced biophysical
control on $\lambda E_T$ for the pasture during the dry season. For the wet season, evidence of minor
biophysical control indicates the dominance of $R_N$ driven equilibrium evaporation in these
PFTs (Hasler and Avissar, 2007; da Rocha et al., 2009; Costa et al., 2010). In the TRF and
TMF, 94% and 99% of the retrieved $g_C/g_A$ ratios fall above 0.5, and, only 1% and 6% of the
retrieved $g_C/g_A$ ratios fall below the 0.5 range (Fig. 9b). In contrast, 90% and 73% of the
$g_C/g_A$ ratios range above 0.5, and 10% to 27% of the $g_C/g_A$ ratios were below 0.5 for TDF and
PAS, respectively (Fig. 9b). This shows that, although radiation control is prevailing in all the
sites, biophysical control is relatively stronger in TDF and PAS as compared to the other
sites. For large $g_C/g_A$ ratios, the conditions within the planetary boundary layer (PBL) become
decoupled from the synoptic scale (McNaughton and Jarvis, 1991) and the net radiative
energy becomes the important regulator of transpiration. For small $g_C/g_A$ ratios (e.g., in dry
season), the conditions within the PBL are strongly coupled to the atmosphere above by rapid
entrainment of air from the capping inversion and by some ancillary effects of sensible heat
flux on the entrainment (McNaughton and Jarvis, 1991). These findings substantiate the
earlier theory of McNaughton and Jarvis (1991), who postulated that large $g_C/g_A$ ratios result
in minor biophysical control on canopy transpiration due to the negative feedback on the
canopy from the PBL. The negative relationship between 1-$\Omega$ and $\lambda E_E$ (Fig. 5b) over all the
PFTs is due to the feedback of $g_A$ on $g_C$. However, over all the PFTs, a combined control of
$g_A$ and environmental variables on $\lambda E_E$ again highlighted the impact of realistically estimated
$g_A$ on $\lambda E_E$ (Holwerda et al., 2012).
It is important to mention that forests are generally expected to be better coupled to the
atmosphere, which is related to generally higher $g_A$ (due to high surface roughness) compared
to the pastures. This implies that forests exhibit stronger biophysical control on $\lambda E_T$.
However, due to the broad leaves of the rain forests (larger leaf area index) and higher
surface wetness (due to higher rainfall amounts) the wet surface area is much larger in the
forests than in the pastures. This results in much higher $g_C$ values for forests than for pastures
during the wet season ($g_C \approx g_A$), and $g_C/g_A \rightarrow 1$. Consequently, no significant difference in
coupling was found between them during the wet season (Fig. 5c and 5d). Despite the
absolute differences in $g_A$ and $g_C$ between forest and pasture, the high surface wetness is
largely offsetting the expected $\Omega$ difference between them. Although the surface wetness is
substantially lower during the dry season, the high water availability in the forests due to the
deeper root systems help maintaining a relatively high $g_C$ compared to the pastures. Hence,
despite $g_A$ (forest) > $g_A$ (pasture) during the dry season, substantially lower $g_C$ values for the
pasture result in lower $g_C/g_A$ ratio for the pasture compared to the forest, thus causing more
biophysical control on $\lambda E_T$ during the dry season. The relatively better relationship between
coupling versus $\lambda E_T$ in PAS and TDF during the dry season was also attributed to high
surface air temperature difference ($T_R - T_A$) in these PFTs that resulted in low $g_C/g_A$ ratios
(Fig. 9c).
**5.3 Factors affecting $g_C$ and $g_A$ variability**
The stomatal feedback-response hypothesis (Monteith, 1995) also became apparent at the
canopy-scale (Fig. 8a, 8b), which states that a decrease in $g_C$ with increasing $D_A$ is caused by
a direct increase in $\lambda E_T$ (Monteith, 1995; Matzner & Comstock, 2001; Streck, 2003) and $g_C$
responds to the changes in the air humidity by sensing $\lambda E_T$, rather than $D_A$. This feedback
mechanism is found because of the influence of $D_A$ on both $g_C$ and $\lambda E_T$, which in turn
changes $D_A$ by influencing the air humidity (Monteith, 1995). The change in $g_C$ is dominated
by an increase in the net available energy, which is partially offset by an increase in $\lambda E_T$.
After the net energy input in the canopy exceeds a certain threshold, $g_C$ starts decreasing even
if $\lambda E_T$ increases. High $\lambda E_T$ increases the water potential gradient between guard cells and
other epidermal cells or reduces the bulk leaf water potential, thus causing stomatal closure
(Monteith, 1995; Jones, 1998; Streck, 2003). The control of soil water on transpiration also
became evident from the scatter plots between $g_C$ versus $\lambda E_T$ and $T_R$ for different $D_A$ levels
(Fig. 8b, 8c) (also Fig. 7). Denmead and Shaw (1962) hypothesized that reduced $g_C$ and
stomatal closure occurs at moderate to higher levels of soil moisture (high $\lambda E_T$) when the
atmospheric demand of water vapor increases (high $D_A$). The water content in the immediate
vicinity of the plant root depletes rapidly at high $D_A$, which decreases the hydraulic
conductivity of soil, and the soil is unable to efficiently supply water under these conditions.
For a given evaporative demand and available energy, transpiration is determined by the
$g_C/g_A$ ratio, which is further modulated by the soil water availability. These combined effects
tend to strengthen the biophysical control on transpiration (Leuzinger and Kirner, 2010;
Migletta et al., 2011). The complex interaction between $g_C$, $T_R$, and $D_A$ (Fig. 8c) explains why
different parametric $g_C$ models produce divergent results.
Although $\lambda E_T$ and $\lambda E_E$ estimates are interdependent on $g_C$ and $g_A$ (as shown in Fig. 6 to Fig.
8); the figures reflect the credibility of the conductances as well as transpiration estimates by
realistically capturing the hysteretic behavior between biophysical conductances and water
vapor fluxes, which is frequently observed in natural ecosystems (Zhang et al., 2014, Renner
et al., 2016) (also Zuecco et al., 2016). These results are also compliant with the theories
postulated earlier from observations that the magnitude of hysteresis depends on the
radiation-vapor pressure deficit time-lag, while the soil moisture availability is a key factor
modulating the hysteretic transpiration-vapor pressure deficit relation as soil moisture
declines (Zhang et al., 2014; O'Grady et al., 1999; Jarvis and McNaughton, 1986). This
shows that despite being independent of any predefined hysteretic function, the
interdependent conductance-transpiration hysteresis is still captured in STIC1.2.
Fig. 8d is in accordance with existing theory that under conditions of extremely high
atmospheric turbulence (i.e., high $g_A$), a close coupling exists between the surface and the
atmosphere, which causes $T_R$ and $T_A$ to converge (i.e., $T_R - T_A \rightarrow 0$). When $g_A$ is low, the
difference between $T_R$ and $T_A$ increases due to poor vertical mixing of the air.

## 6 Conclusions

By integrating the radiometric surface temperature ($T_R$) into a combined structure of PM-SW
model, we have estimated the canopy-scale biophysical conductances and quantified their
control on the terrestrial evapotranspiration components in a simplified SEB modeling
perspective that treats the vegetation canopy as 'big-leaf'. The STIC1.2 biophysical modeling
scheme is independent of any leaf-scale empirical parameterisation for stomata and
associated aerodynamic variables.
Stomata regulate the coupling between terrestrial carbon and water cycles, which implies that
their behaviour under global environmental change is critical to predict vegetation
functioning (Medlyn et al., 2011). The combination of variability in precipitation (Hilker et
al., 2014) and land cover change (Davidson et al., 2012) in the Amazon Basin is expected to
increase the canopy-atmosphere coupling of pasture or forest systems under drier conditions
by altering the ratio of the biological and aerodynamic conductances. An increase of
biophysical control will most likely be an indicator of shifting the transpiration from an
energy-limited to a water-limited regime (due to the impact of $T_R$, $T_A$, and $D_A$ on the $g_C/g_A$
ratio) with further consequences for the surface water balance and rainfall recycling. At the
same time, a transition from forest to pasture or agriculture lands will substantially reduce the
contribution of interception evaporation in the Amazon, hence, it will affect the regional
water cycle. This might change the moisture regime of the Amazonian Basin and affect the
moisture transport to other regions. In this context, STIC1.2 provides a new quantitative and
internally consistent method for interpreting the biophysical conductances and able to
quantify their controls on the water cycle components in response to a range of climatic and
ecohydrological conditions (excluding rising atmospheric $CO_2$) across a broad spectrum of
PFTs. It could also provide the basis to improve existing land surface parameterisations for
simulating vegetation water use at large spatial scales.
It should also be noted that although the case study described here provides general insights
into the biophysical controls of $\lambda E$ and associated feedback between $g_C$, $D_A$, $T_R$ and $\lambda E_T$ in the
framework of the PM-SW equation, there is a tendency for overestimation of $g_C$ due to the
embedded evaporation information in the current single-source composition of STIC1.2. For
accurate characterisation of canopy conductance, explicit partitioning of $\lambda E$ into transpiration
and evaporation (both soil and interception) is one of the further scopes for improving
STIC1.2 and this assumption needs to be tested further.

## Acknowledgements

The developed modeling framework contributes to the "Catchments As Organized Systems
(CAOS)" Phase-2 research group (FOR 1598) funded by the German Science Foundation
(DFG) and to the "HiWET (High-resolution modelling and monitoring of Water and Energy
Transfers in wetland ecosystems)" consortium funded by BELSPO and FNR. We sincerely
thank Dr. Andrew Jarvis (Lancaster University, UK), Dr. Monica Garcia (Technical
University of Denmark, Denmark), and Dr. Georg Wohlfahrt (University of Innsbruck,
Austria) for very helpful discussions and edits in the manuscript. We are grateful to all
Brazilian and international collaborators and all the funding agencies that have contributed to
the Large-scale Biosphere Atmosphere Experiment in Amazônia (LBA). The authors are
indebted to Pavel Kabat, Antônio Ocimar Manzi, David R. Fitzjarrald, Julio Tota, Humberto
Ribeiro da Rocha, Michael Goulden, Maarten J. Waterloo and Luiz Martinelli for planning,
coordinating, conducting, and evaluating the eddy covariance, meteorological and leaf gas
exchange measurements at the LBA sites. We are particularly grateful to all field technicians
whose hard work were the key ingredients to establish the quality of the datasets used in this
paper. The authors declare no conflict of interest. DTD acknowledges support of the Jet
Propulsion Laboratory, California Institute of Technology, under a contract with the National
Aeronautics and Space Administration.

## **Appendix A:**


### **A1 Derivation of 'state equations' in STIC 1.2**


Neglecting horizontal advection and energy storage, the surface energy balance equation is
written as follows:

$$\phi = \lambda E + H \tag{A1}$$

Figure (A1) shows that, while $H$ is controlled by a single aerodynamic resistance ($r_A$) (or
$1/g_A$); $\lambda E$ is controlled by two resistances in series, the surface resistance ($r_C$) (or $1/g_C$) and
the aerodynamic resistance to vapor transfer ($r_C + r_A$). For simplicity, it is implicitly assumed
that the aerodynamic resistance of water vapor and heat are equal (Raupach, 1998), and both
the fluxes are transported from the same level from near surface to the atmosphere. The
sensible and latent heat flux can be expressed in the form of aerodynamic transfer equations
(Boegh et al., 2002; Boegh and Soegaard, 2004) as follows:

$$H = \rho c_P g_A (T_o - T_A) \tag{A2}$$

$$\lambda E = \frac{\rho c_P}{\gamma} g_A (e_0 - e_A) = \frac{\rho c_P}{\gamma} g_C (e_0^* - e_0) \tag{A3}$$

Where $T_0$ and $e_0$ are the air temperature and vapor pressure at the source/sink height (i.e.,
aerodynamic temperature and vapor pressure) or at the so-called roughness length ($z_0$), where
wind speed is zero. They represent the vapor pressure and temperature of the quasi-laminar
boundary layer in the immediate vicinity of the surface level (Fig. A1), and $T_0$ can be
obtained by extrapolating the logarithmic profile of $T_A$ down to $z_0$. $e_0{}^*$ is the saturation vapor
pressure at $T_0$ (hPa).
By combining eqn. (A1), (A2), and (A3) and solving for $g_A$, we get the following equation.

$$g_A = \frac{\phi}{\rho c_P \left[ (T_o - T_A) + \left( \frac{e_0 - e_A}{\gamma} \right) \right]} \qquad \text{(A4)}$$

Combining the aerodynamic expressions of $\lambda E$ in eqn. (A3) and solving for $g_C$, we can
express $g_C$ in terms of $g_A$, $e_0{}^*$, $e_0$, and $e_A$.

$$g_C = g_A \frac{(e_0 - e_A)}{(e_0^* - e_0)} \qquad \text{(A5)}$$

While deriving the expressions for $g_A$ and $g_C$, two more unknown variables are introduced ($e_0$
and $T_0$), thus there are two equations and four unknowns. Therefore, two more equations are
needed to close the system of equations.
An expression for $T_0$ is derived from the Bowen ratio ($\beta$) (Bowen, 1926) and evaporative
fraction ($\Lambda$) (Shuttleworth et al., 1989) equation.

$$\beta = \left( \frac{1 - \Lambda}{\Lambda} \right) = \frac{\gamma(T_0 - T_A)}{(e_0 - e_A)} \qquad \text{(A6)}$$

$$T_o = T_A + \left( \frac{e_0 - e_A}{\gamma} \right) \left( \frac{1 - \Lambda}{\Lambda} \right) \qquad \text{(A7)}$$

This expression for $T_0$ introduces another new variable ($\Lambda$); therefore, one more equation that
describes the dependence of $\Lambda$ on the conductances ($g_A$ and $g_C$) is needed to close the system
of equations. In order to express $\Lambda$ in terms of $g_A$ and $g_C$, we had adopted the advection –
aridity (AA) hypothesis (Brutsaert and Stricker, 1979) with a modification introduced by
(Mallick et al., 2015). The AA hypothesis is based on a complementary connection between
the potential evaporation ($E^*$), sensible heat flux ($H$), and $E$; and leads to an assumed link
between $g_A$ and $T_0$. However, the effects of surface moisture (or water stress) were not
explicit in the AA equation and Mallick et al. (2015) implemented a moisture constraint in
the original advection-aridity hypothesis while deriving a 'state equation' of $\Lambda$ (eqn. A8
below). A detailed derivation of the 'state equation' for $\Lambda$ is described in the Supplement (S1)
(also see Mallick et al., 2014, 2015). Estimation of $e_0$, $e_0^*$, $M$, and $\alpha$ is described in the
Appendix (A2).

$$\Lambda = \frac{2\alpha s}{2s \,+\, 2\gamma \,+\, \gamma \frac{g_A}{g_C}(1 + M)} \tag{A8}$$

### A2 Iterative solution of $e_0$, $e_0^*$, M, and α in STIC 1.2
In STIC1.0 and 1.1 (Mallick et al., 2014; 2015), no distinction was made between the surface
and source/sink height vapor pressures. Therefore, $e_0^*$ was approximated as the saturation
vapor pressure at $T_R$ and $e_0$ was empirically estimated from $M$ based on the assumption that
the vapor pressure at the source/sink height ranges between extreme wet–dry surface
conditions. However, the level of $e_0$ and $e_0^*$ should be consistent with the level of the
aerodynamic temperature ($T_0$) from which the sensible heat flux is transferred (Lhomme and
Montes, 2014). The predictive use of the PM model could be hindered due to neglecting the
feedbacks between the surface layer evaporative fluxes and source/sink height mixing and
coupling (McNaughton and Jarvis, 1984), and their impact on the canopy scale conductances.
Therefore, in STIC1.2, we have used physical expressions for estimating $e_0$ and $e_0^*$ followed
by estimating $T_{SD}$ and $M$ as described below. The fundamental differences between STIC1.0,
1.1 and 1.2 modeling philosophy are described in Table A1.
An estimate of $e_0^*$ is obtained by inverting the aerodynamic transfer equation of $\lambda E$.

$$e_0^* = e_A + \left[ \frac{\gamma \lambda E (g_A + g_C)}{\rho c_P g_A g_C} \right] \tag{A9}$$

Following Shuttleworth and Wallace (1985) (SW), the vapor pressure deficit ($D_0$) (= $e_0^*$ - $e_0$)
and vapor pressure ($e_0$) at the source/sink height are expressed as follows.

$$D_0 = D_A + \left[ \frac{\{s\phi - (s + \gamma)\lambda E\}}{\rho c_P g_A} \right] \tag{A10}$$

$$e_0 = e_0^* - D_0 \tag{A11}$$

A physical equation of $\alpha$ is derived by expressing the evaporative fraction ($\Lambda$) as function of
the aerodynamic equations of $H$ $[\rho c_P g_A (T_0 - T_A)]$ and $\lambda E$ $[\frac{\rho c_P}{\gamma} \frac{g_A g_C}{g_A + g_C} (e_0^* - e_A)]$ as follows.

$$\Lambda = \frac{\lambda E}{H + \lambda E} \tag{A12}$$

$$= \frac{\frac{\rho c_P}{\gamma} \frac{g_A g_C}{g_A + g_C} (e_0^* - e_A)}{\rho c_P g_A (T_0 - T_A) + \frac{\rho c_P}{\gamma} \frac{g_A g_C}{g_A + g_C} (e_0^* - e_A)} \tag{A13}$$

$$= \frac{g_C (e_0^* - e_A)}{[\gamma (T_0 - T_A)(g_A + g_C) + g_C (e_0^* - e_A)]} \tag{A14}$$

Combining eqn. (A14) and eqn. (A8) (eliminating $\Lambda$), we can derive a physical equation of $\alpha$.

$$\alpha = \frac{g_C (e_0^* - e_A) \left[ 2s + 2\gamma + \gamma \frac{g_A}{g_C} (1 + M) \right]}{2s [\gamma (T_0 - T_A)(g_A + g_C) + g_C (e_0^* - e_A)]} \tag{A15}$$

Following Venturini et al. (2008), $M$ can be expressed as the ratio of the vapor pressure
difference to the vapor press deficit between surface to atmosphere as follows.

$$M = \frac{(e_0 - e_A)}{(e_0^* - e_A)} = \frac{(e_0 - e_A)}{\kappa (e_S^* - e_A)} = \frac{s_1 (T_{SD} - T_D)}{\kappa s_2 (T_R - T_D)} \tag{A16}$$

Where $T_{SD}$ is the dewpoint temperature at source/sink height and $T_D$ is the air dewpoint
temperature; $s_1$ and $s_2$ are the psychrometric slopes of the saturation vapor pressure and
temperature between $(T_{SD} - T_D)$ versus $(e_0 - e_A)$ and $(T_R - T_D)$ versus $(e_s^* - e_A)$ relationship
(Venturini et al., 2008); and $\kappa$ is the ratio between $(e_0^* - e_A)$ and $(e_s^* - e_A)$. Despite $T_0$ drives
the sensible heat flux, the comprehensive dry-wet signature of underlying surface due to soil
moisture variations is directly reflected in $T_R$ (Kustas and Anderson, 2009). Therefore, using
$T_R$ in the denominator of eqn. (A16) tends to give a direct signature of the surface moisture
availability ($M$). In eqn. (A16), $T_{SD}$ computation is challenging because both $e_0$ and $s_1$ are
unknown. By decomposing the aerodynamic equation of $\lambda E$, $T_{SD}$ can be expressed as follows.

$$\lambda E = \frac{\rho c_P}{\gamma} g_A (e_0 - e_A) = \frac{\rho c_P}{\gamma} g_A s_1 (T_{SD} - T_D)$$

$$T_{SD} = T_D + \frac{\gamma \lambda E}{\rho c_P g_A s_1} \tag{A17}$$

In the earlier STIC versions, $s_1$ was approximated at $T_D$, $e_0^*$ was approximated at $T_R$, $T_{SD}$ was
estimated from $s_1$, $T_D$, $T_R$, and related saturation vapor pressures (Mallick et al., 2014; 2015),
and $M$ was estimated from eqn. (A16) (estimation of $T_{SD}$ and $M$ was stand-alone earlier).
However, since $T_{SD}$ depends on $\lambda E$ and $g_A$, an iterative procedure is applied to estimate $T_{SD}$
and $M$ as described below.
In STIC1.2, an initial value of $\alpha$ is assigned as 1.26 and initial estimates of $e_0^*$ and $e_0$ are
obtained from $T_R$ and $M$ as $e_0^* = 6.13753 e^{\frac{17.27 T_R}{(T_R + 237.3)}}$ and $e_0 = e_A + M(e_0^* - e_A)$. Initial $T_{SD}$
and $M$ were estimated as described above. With the initial estimates of these variables; first
estimate of the conductances, $T_0$, $\Lambda$, and $\lambda E$ are obtained. The process is then iterated by
updating $e_0^*$ (using eqn. A9), $D_0$ (using eqn. A10), $e_0$ (using eqn. A11), $T_{SD}$ (using eqn. A17
with $s_1$ estimated at $T_D$), $M$ (using eqn. A16), and $\alpha$ (using eqn. A15), with the first estimates
of $g_C$, $g_A$, and $\lambda E$, and recomputing $g_C$, $g_A$, $T_0$, $\Lambda$, and $\lambda E$ in the subsequent iterations with the
previous estimates of $e_0^*$, $e_0$, $T_{SD}$, $M$, and $\alpha$ until the convergence $\lambda E$ is achieved. Stable
values of $\lambda E$, $e_0^*$, $e_0$, $T_{SD}$, $M$, and $\alpha$ are obtained within ~25 iterations. Illustrative examples
of the convergence of $e_0^*$, $e_0$, $T_{SD}$, $M$, and $\alpha$ are shown in Fig. (A3).
To summarize, the computational steps of the conductances and evaporative fluxes in STIC
are:
*Step 1: Analytical solution of the conductances, $T_0$ and $\Lambda$ by solving the 'state equations'*
*(eqn. 2, 3, 4, and 5). Step 2: Initial estimates of the conductances ($g_C$ and $g_A$), $T_0$, $\Lambda$, $\lambda E$ and*
*H. Step 3: Simultaneous iteration of $\lambda E$, $e_0^*$, $e_0$, $T_{SD}$, M, and $\alpha$; and final estimation of the*
*conductances ($g_C$ and $g_A$), $T_0$, $\Lambda$, $\lambda E$ and H. Step 4: Partitioning $\lambda E$ into $\lambda E_T$ and $\lambda E_E$.*

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

 **Table 1:** Variables and symbols and their description used in the present study.

| Variables and symbol | Description |
| --- | --- |
| $\lambda E$ | Evapotranspiration (evaporation + transpiration) as latent heat flux (W m$^{-2}$) |
| $H$ | Sensible heat flux (W m$^{-2}$) |
| $R_N$ | Net radiation (W m$^{-2}$) |
| $G$ | Ground heat flux (W m$^{-2}$) |
| $\phi$ | Net available energy (W m$^{-2}$) |
| $T_A$ | Air temperature (ºC) |
| $T_D$ | Dewpoint temperature (ºC) |
| $T_R$ | Radiometric surface temperature (ºC) |
| $R_H$ | Relative humidity (%) |
| $e_A$ | Atmospheric vapor pressure at the level of $T_A$ measurement (hPa) |
| $D_A$ | Atmospheric vapor pressure deficit at the level of $T_A$ measurement (hPa) |
| $u$ | Wind speed (m s$^{-1}$) |
| $u^*$ | Friction velocity (m s$^{-1}$) |
| $T_{SD}$ | Dew-point temperature at the source/sink height (ºC) |
| $T_0$ | Aerodynamic temperature or source/sink height temperature (ºC) |
| $e_S$ | 'effective' vapor pressure of evaporating front near the surface (hPa) |
| $e_S^*$ | Saturation vapor pressure of surface (hPa) |
| $e_0^*$ | Saturation vapor pressure at the source/sink height (hPa) |
| $e_0$ | Atmospheric vapor pressure at the source/sink height (hPa) |
| $D_0$ | Atmospheric vapor pressure deficit at the source/sink height (hPa) |
| $\lambda E_{eq}$ | Equilibrium latent heat flux (W m$^{-2}$) |
| $\lambda E_{imp}$ | Imposed latent heat flux (W m$^{-2}$) |
| $\lambda E_E$ | Evaporation as flux (W m$^{-2}$) |
| $\lambda E_T$ | Transpiration flux (W m$^{-2}$) |
| $E$ | Evapotranspiration (evaporation + transpiration) as depth of water (mm) |
| $\lambda E^*$ | Potential evaporation as flux (W m$^{-2}$) |
| $\lambda E_T^*$ | Potential transpiration as flux (W m$^{-2}$) |
| $\lambda E_W$ | Wet environment evaporation as flux (W m$^{-2}$) |
| $\lambda E_P^*$ | Potential evaporation as flux according to Penman (W m$^{-2}$) |
| $\lambda E_{PM}^*$ | Potential evaporation as flux according to Penman-Monteith (W m$^{-2}$) |
| $\lambda E_{PT}^*$ | Potential evaporation as flux according to Priestley-Taylor (W m$^{-2}$) |
| $E^*$ | Potential evaporation as depth of water (mm) |
| $E_P^*$ | Potential evaporation as depth of water according to Penman (mm) |
| $E_{PM}^*$ | Potential evaporation as depth of water according to Penman-Monteith (mm) |
| $E_{PT}^*$ | Potential evaporation as depth of water according to Priestley-Taylor (mm) |
| $E_W$ | Wet environment evaporation as depth of water (mm) |
| $g_A$ | Aerodynamic conductance (m s$^{-1}$) |
| $g_C$ | Stomatal / surface conductance (m s$^{-1}$) |
| $g_M$ | Momentum conductance (m s$^{-1}$) |
| $g_B$ | Quasi-laminar boundary layer conductance (m s$^{-1}$) |
| $g_{Cmax}$ | Maximum stomatal / surface conductance (m s$^{-1}$) (= $g_C/M$) |

| | |
|---|---|
| $M$ | Surface moisture availability $(0 - 1)$ |
| $s$ | Slope of saturation vapor pressure versus temperature curve (hPa K$^{-1}$) (estimated at $T_A$) |
| $s_1$ | Slope of the saturation vapor pressure and temperature between $(T_{SD} - T_D)$ versus $(e_0 - e_A)$ (approximated at $T_D$) (hPa K$^{-1}$) |
| $s_2$ | Slope of the saturation vapor pressure and temperature between $(T_R - T_D)$ versus $(e_S^* - e_A)$ (hPa K$^{-1}$), estimated according to Mallick et al. (2015). |
| $s_3$ | Slope of the saturation vapor pressure and temperature between $(T_R - T_{SD})$ versus $(e_S^* - e_S)$ (approximated at $T_R$) (hPa K$^{-1}$) |
| $\kappa$ | Ratio between $(e_0^* - e_A)$ and $(e_S^* - e_A)$ |
| $\lambda$ | Latent heat of vaporization of water (j kg$^{-1}$K$^{-1}$) |
| $z_R$ | Reference height (m) |
| $z_M$ | Effective source-sink height of momentum (m) |
| $z_0$ | Roughness length (m) |
| $d$ | Displacement height (m) |
| $\gamma$ | Psychrometric constant (hPa K$^{-1}$) |
| $\rho$ | Density of air (kg m$^{-3}$) |
| $c_p$ | Specific heat of dry air (MJ kg$^{-1}$ K$^{-1}$) |
| $\Lambda$ | Evaporative fraction (unitless) |
| $\beta$ | Bowen ratio (unitless) |
| $\alpha$ | Priestley-Taylor parameter (unitless) |
| $\Omega$ | Decoupling coefficient (unitless) |
| $S_c$ | Schmidt number (unitless) |
| $P_r$ | Prandtl number (unitless) |
| $k$ | Von Karman's constant (0.4) |












**Table 2**: Overview of the LBA tower sites. Here, (-) refers to (S) and (W) for latitude and longitude,
respectively.

| Biome | PFT | Site | LBA Code | Data availability period | Latitude (°) | Longitude (°) | Tower height (m) | Annual rainfall (mm) |
|---|---|---|---|---|---|---|---|---|
| Forest | Tropical rainforest (TRF) | Manaus KM34 | K34 | 06/1999 to 09/2006 | -2.609 | -60.209 | 50 | 2329 |
| Forest | Tropical moist forest (TMF) | Santarem KM67 | K67 | 01/2002 to 01/2006 | -2.857 | -54.959 | 63 | 1597 |
| Forest | Tropical moist forest (TMF) | Santarem KM83 | K83 | 07/2000 to 12/2004 | -3.018 | -54.971 | 64 | 1656 |
| Forest | Tropical dry forest (TDF) | Reserva Biológica Jarú | RJA | 03/1999 to 10/2002 | -10.083 | -61.931 | 60 | 2354 |
| Pasture | Pasture (PAS) | Santarem KM77 | K77 | 01/2000 to 12/2001 | -3.012 | -54.536 | 18 | 1597 |
| Pasture | Pasture (PAS) | Fazenda Nossa Senhora | FNS | 03/1999 to 10/2002 | -10.762 | -62.357 | 8.5 | 1743 |
















**Table 3**: Comparative statistics for the STIC and tower-derived hourly $g_A$ and $g_C$ for a range of PFTs
in the Amazon Basin (LBA tower sites). Values in parenthesis are ± one standard deviation (standard
error for correlation).

| PFTs | $g_{A\text{-}STIC}$ vs. $g_{A\text{-}BM13}$ | | | | | $g_{C\text{-}STIC}$ vs. $g_{C\text{-}INV}$ | | | |
|---|---|---|---|---|---|---|---|---|---|
| | RMSD (m s$^{-1}$) | $R^2$ | Slope | Offset (m s$^{-1}$) | N | RMSD (m s$^{-1}$) | $R^2$ | Slope | Offset (m s$^{-1}$) |
| TRF | 0.013 | 0.41 (±0.03) | 1.07 (±0.047) | 0.0031 (±0.0008) | 1159 | 0.012 | 0.14 (±0.04) | 0.39 (±0.039) | 0.0097 (±0.0007) |
| TMF | 0.012 | 0.55 (±0.12) | 0.81 (±0.023) | 0.0006 (±0.0006) | 1927 | 0.009 | 0.55 (±0.12) | 0.85 (±0.025) | 0.0032 (±0.0005) |
| TDF | 0.007 | 0.49 (±0.15) | 0.89 (±0.041) | 0.0019 (±0.0006) | 787 | 0.012 | 0.33 (±0.19) | 0.30 (±0.022) | 0.0050 (±0.0005) |
| PAS | 0.012 | 0.22 (±0.18) | 1.03 (±0.083) | 0.0059 (±0.0007) | 288 | 0.007 | 0.58 (±0.12) | 0.65 (±0.025) | 0.0024 (±0.0003) |
| **Mean** | **0.012** | **0.44 (±0.10)** | **0.76 (±0.016)** | **0.0047 (±0.003)** | **4161** | **0.010** | **0.39 (±0.08)** | **0.63 (±0.016)** | **0.0046 (±0.0003)** |

N = number of data points; RMSD = root mean square deviation between predicted (P) and observed (O)
variables $=\left[\frac{1}{N}\sum_{i=0}^{N}(P_i - O_i)^2\right]^2$.


**Table 4**: Comparative statistics for the STIC and tower-derived hourly λE and H for a range of PFTs
in the Amazon Basin (LBA tower sites). Values in parenthesis are ±one standard deviation (standard
error for correlation).

| PFTs | λE | | | | H | | | | N |
|---|---|---|---|---|---|---|---|---|---|
| | RMSD (W m$^{-2}$) | $R^2$ | Slope | Offset (W m$^{-2}$) | RMSD (W m$^{-2}$) | $R^2$ | Slope | Offset (W m$^{-2}$) | |
| TRF | 28 | 0.96 (±0.007) | 1.10 (±0.008) | -16 (±2) | 34 | 0.52 (±0.030) | 0.60 (±0.025) | 29 (±2) | 1159 |
| TMF | 20 | 0.98 (±0.004) | 1.08 (±0.004) | -11 (±1) | 23 | 0.71 (±0.019) | 0.61 (±0.014) | 20 (±1) | 1927 |
| TDF | 26 | 0.96 (±0.009) | 0.96 (±0.008) | -7 (±2) | 30 | 0.66 (±0.032) | 0.89 (±0.035) | 20 (±3) | 787 |
| PAS | 31 | 0.96 (±0.009) | 1.14 (±0.010) | -2 (±2) | 33 | 0.88 (±0.016) | 0.67 (±0.011) | 9 (±1) | 288 |
| **Mean** | **33** | **0.94 (±0.005)** | **1.04 (±0.005)** | **-1 (±1)** | **37** | **0.61 (±0.021)** | **0.58 (±0.009)** | **24 (±2)** | **4161** |






**Figure 1**. Examples of monthly averages of the diurnal time series of canopy-scale (a) $g_A$ and (b) $g_C$
estimated for two different biomes (forest and pasture) in the Amazon Basin (LBA sites K34 and
FNS). The time series of four different $g_A$ estimates and their corresponding $g_C$ estimates are shown
here.

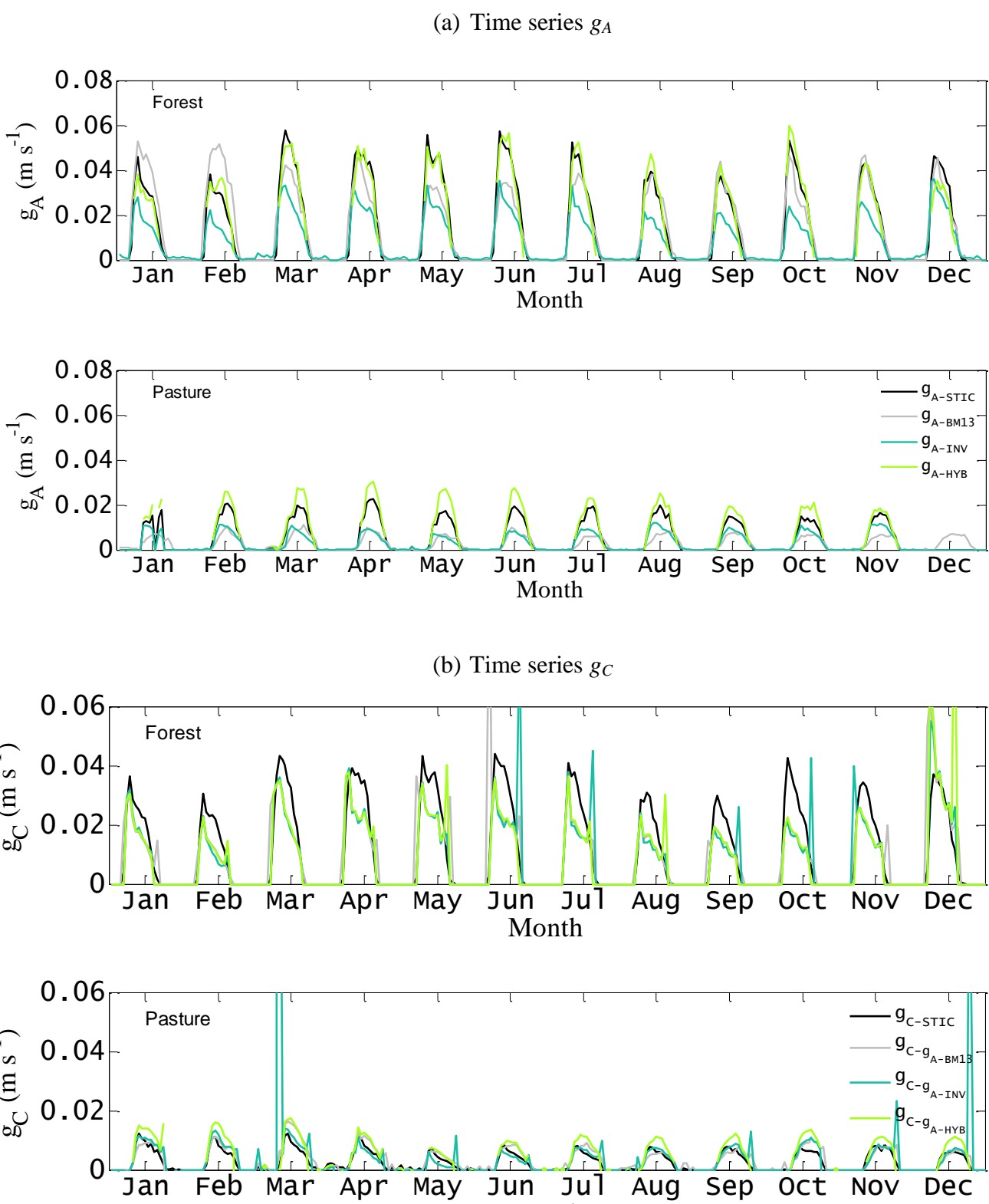

(a) Time series $g_A$

(b) Time series $g_C$


**Figure 2**. (a) Comparison between STIC derived $g_A$ ($g_{A-STIC}$) with an estimated aerodynamic
conductance based on friction velocity ($u^*$) and wind speed ($u$) according to Baldocchi and Ma (2013)
($g_{A-BM13}$), (b) Comparison between $g_{A-STIC}$ with an inverted $g_A$ ($g_{A-INV}$) based on EC observations of $\lambda E$
and $D_A$, (c) Comparison between $g_{A-STIC}$ with a hybrid $g_A$ ($g_{A-HYB}$) based on EC observations of $H$ and
estimated $T_0$ over the LBA EC sites, (d) Comparison between residual $g_A$ differences versus $u$ and $u^*$,
(e) and (f) Relationship between wind and shear derived $g_A$ versus $\phi$, $T_R$, and $D_A$ over the LBA EC
sites.

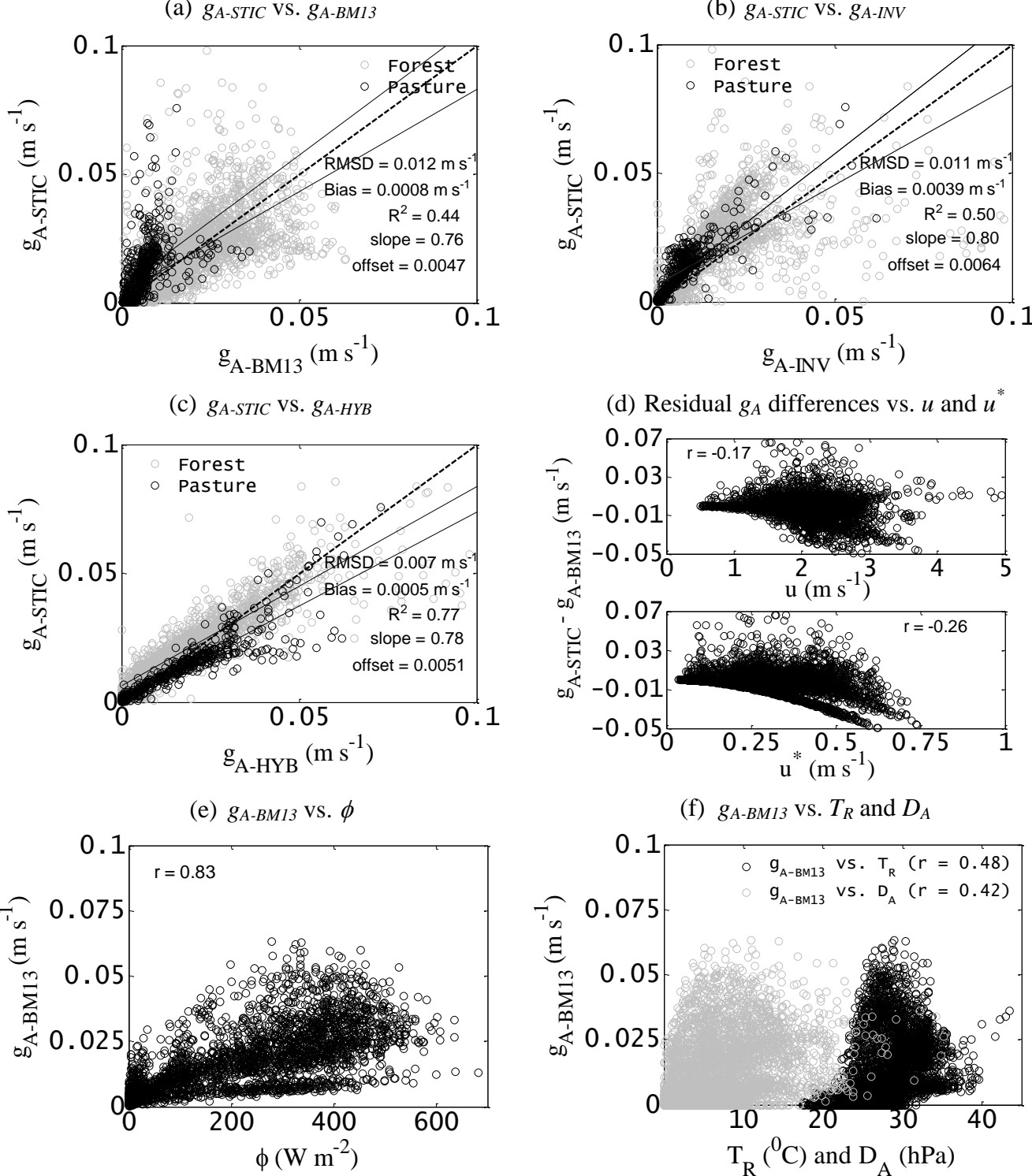


**Figure 3**. (a) Comparison between STIC derived $g_C$ ($g_{C-STIC}$) and $g_C$ computed by inverting the PM
model ($g_{C-INV}$) over the LBA EC sites, where $g_{A-BM13}$ was used as aerodynamic input in conjunction
with tower measurements of $\lambda E$, radiation and meteorological variables, (b) Residual $g_C$ differences
versus wind speed ($u$) and friction velocity ($u^*$) over the LBA EC sites.

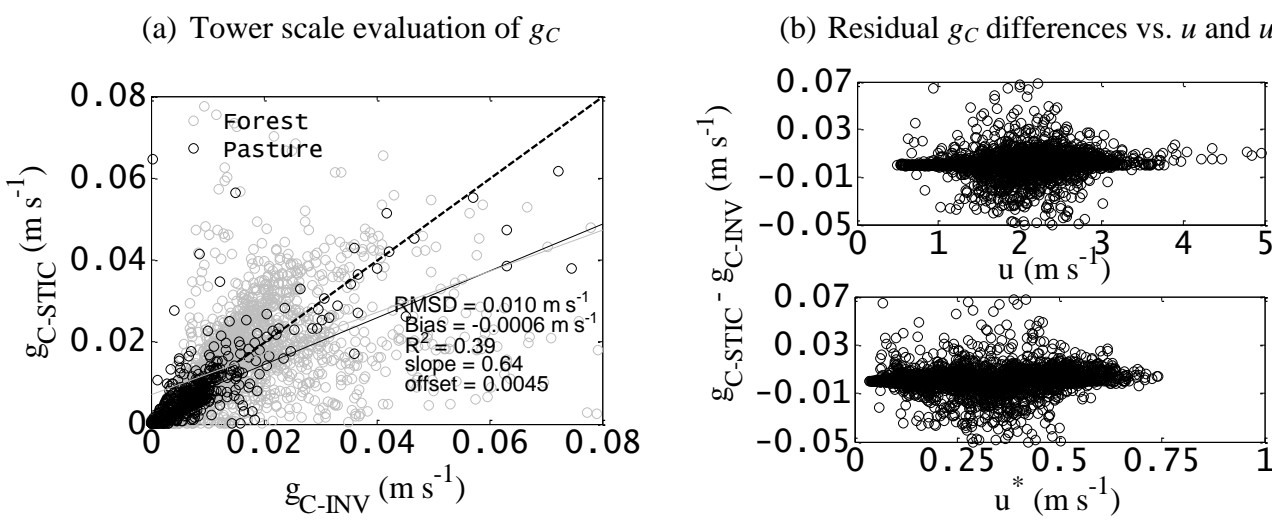


















**Figure 4**. Comparison between STIC derived (a) $\lambda E$ and (b) $H$ over four different PFTs in the
Amazon Basin (LBA tower sites). MAPD is the percent error defined as the mean absolute deviation
between predicted and observed variable divided by mean observed variable.

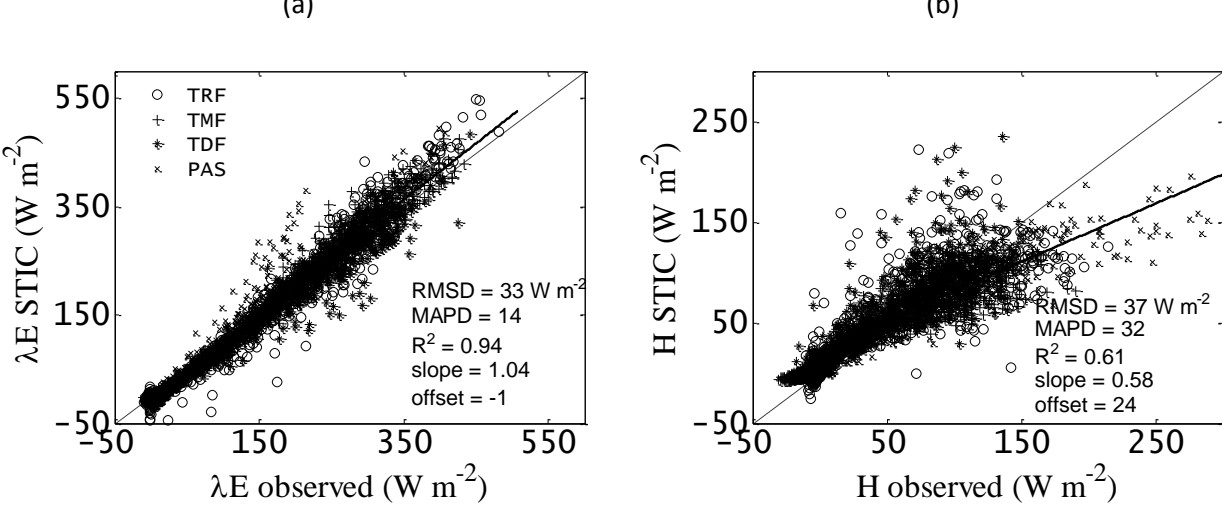


















**Figure 5**. Correlation of coupling (1-Ω) with (a) transpiration ($\lambda E_T$) and (b) evaporation ($\lambda E_E$) and over four different PFTs by combining data for all the years, only during dry seasons for all the years, and during drought year 2005. Data for 2005 was not available for TDF and PAS. (c) to (e) Examples of diurnal pattern of Ω (black lines), $\lambda E_E$ (grey dotted lines) and $\lambda E_T$ (grey solid lines) estimated over two ecohydrologically contrasting biomes (K34 for forest and FNS for pasture) in the Amazon Basin (LBA tower sites) during wet and dry seasons.

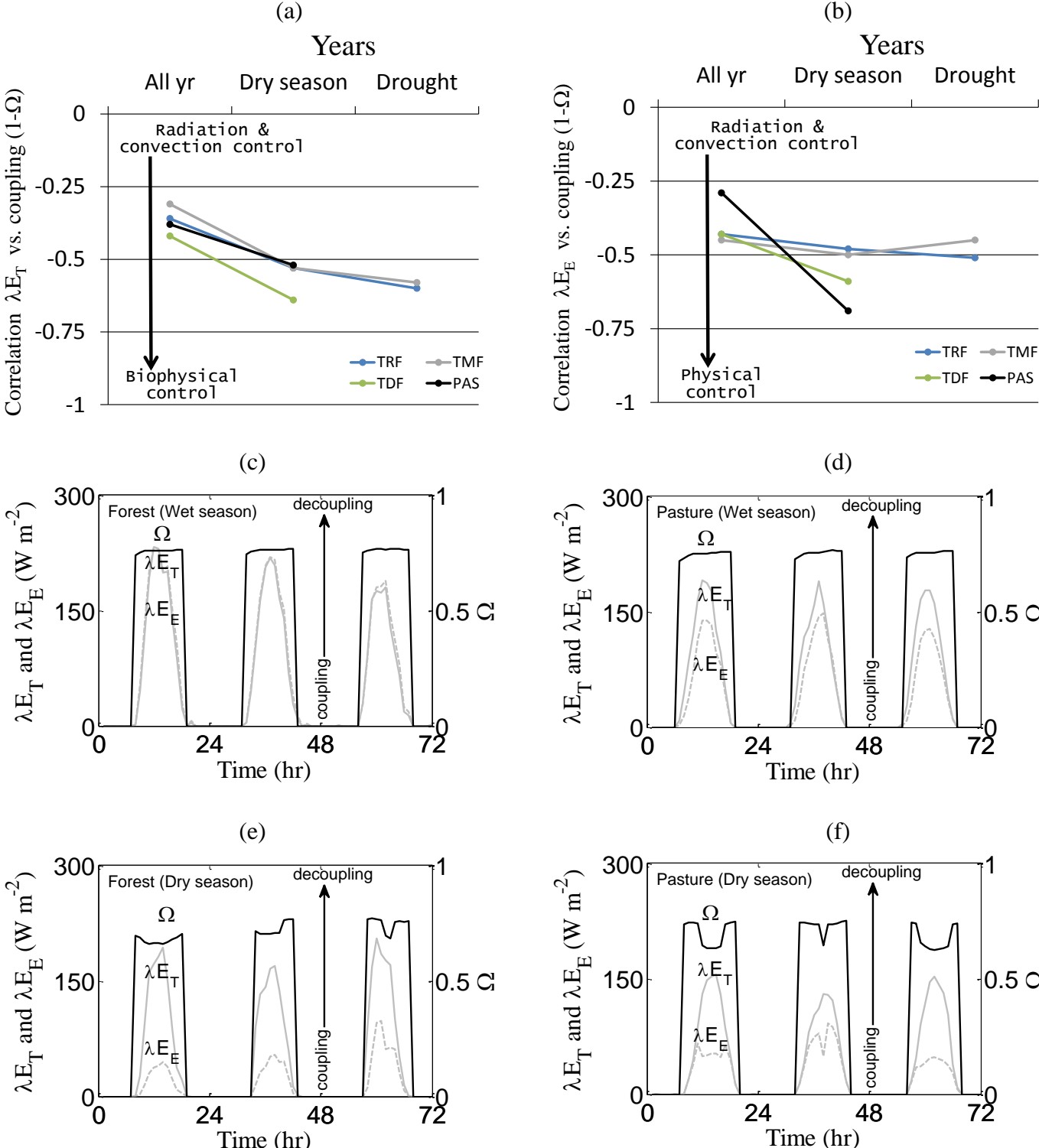

**Figure 6**. Scatter plots of transpiration ($\lambda E_T$) and evaporation ($\lambda E_E$) versus $g_C$ and $g_A$ over four
different PFTs in the Amazon Basin (LBA tower sites).

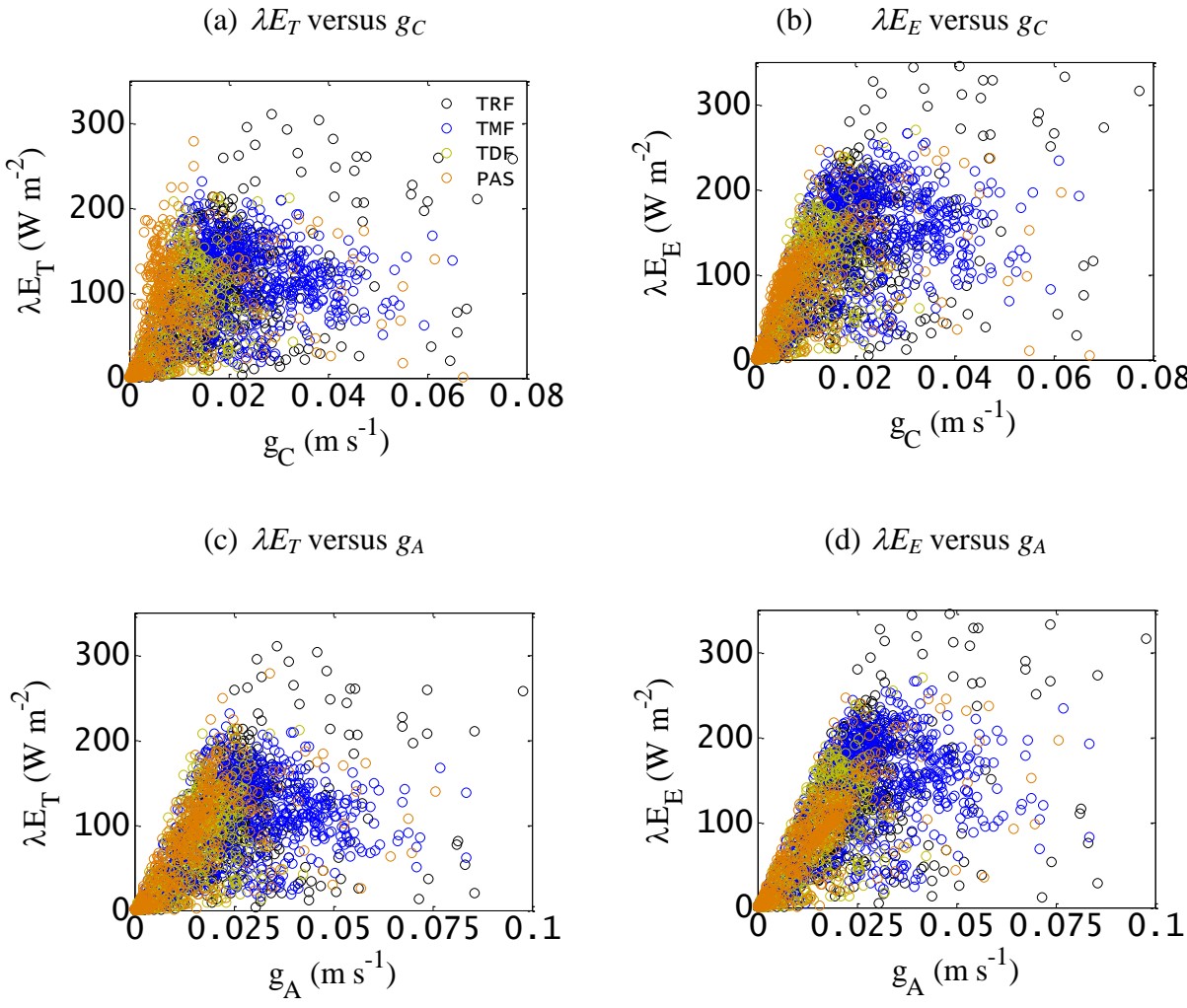











**Figure 7**. Illustrative examples of the occurrence of diurnal hysteresis of transpiration ($\lambda E_T$) during
wet and dry seasons with canopy and environmental controls over two different sites with different
annual rainfall (2329 mm and 1597 mm, respectively) in the Amazon Basin (LBA tower sites K34
and FNS).

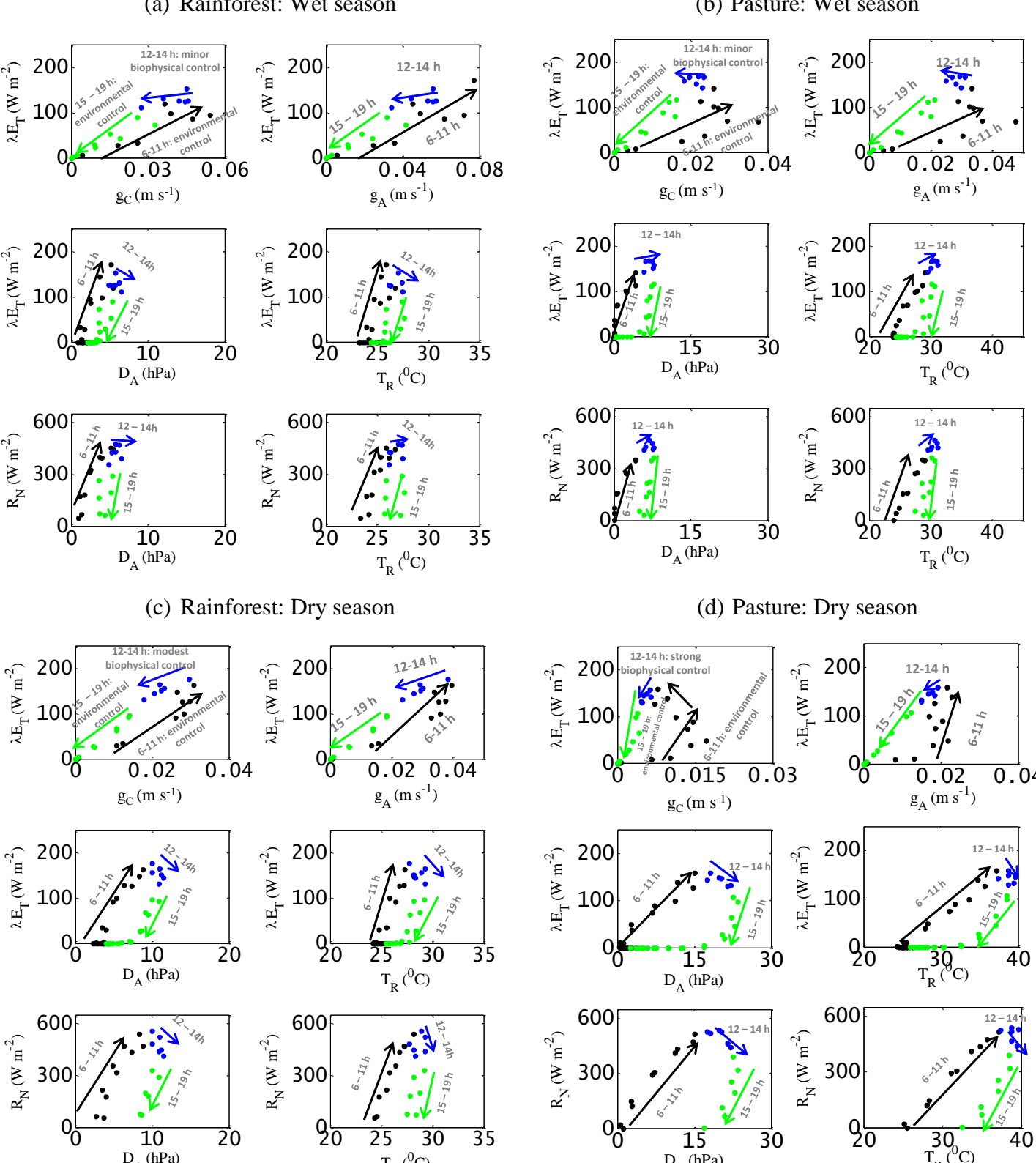

**Figure 8**. (a) Response of retrieved $g_C$ to atmospheric vapor pressure deficit ($D_A$) for different classes of net radiation ($R_N$), (b) Response of retrieved $g_C$ to transpiration for different classes of $D_A$, (c) Response of retrieved $g_C$ to radiometric surface temperature ($T_R$) for different classes $D_A$, (d) Relationship between retrieved $g_A$ and radiometric surface temperature and air temperature difference ($T_R - T_A$) in the Amazon Basin (LBA tower sites).

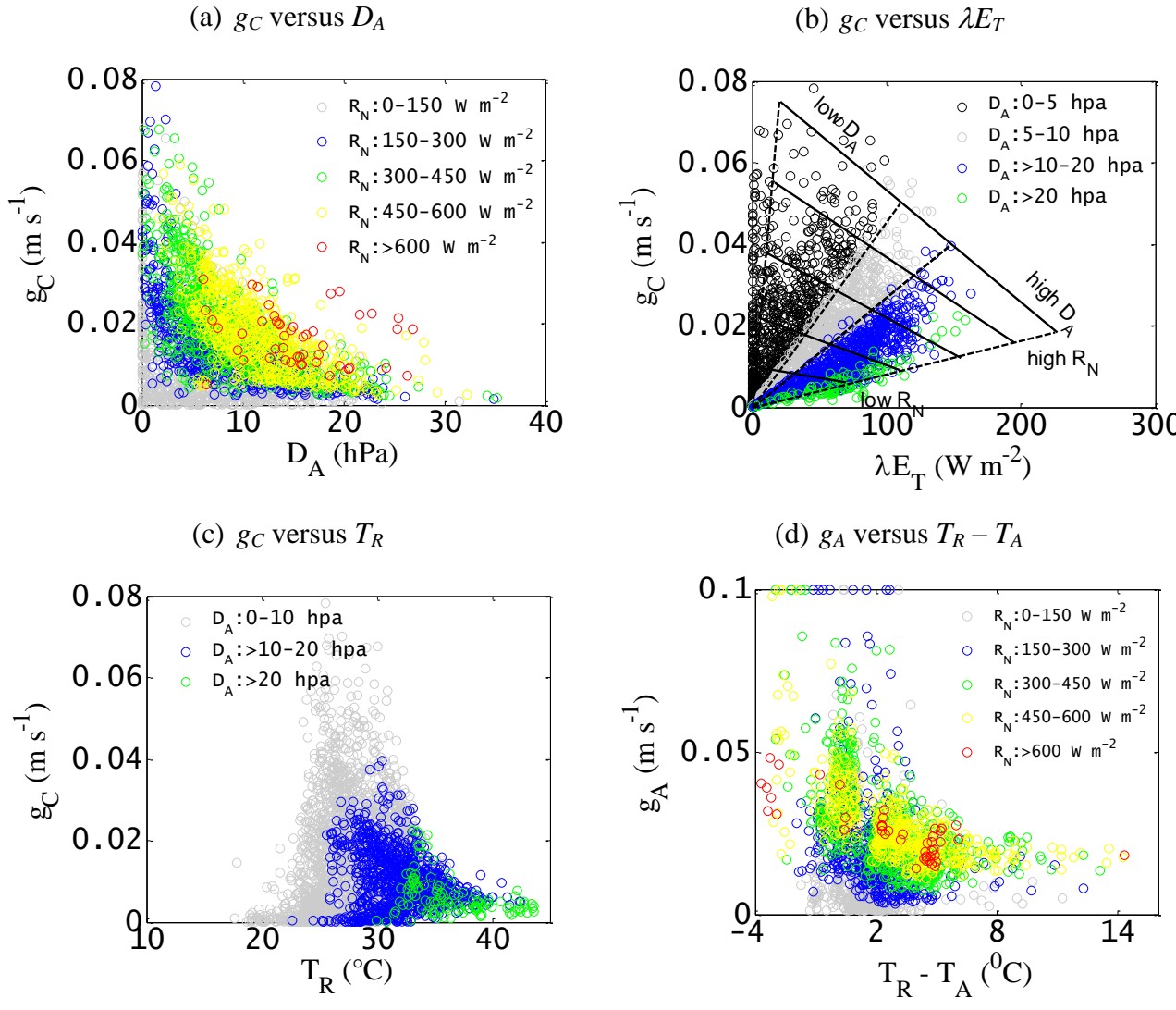

**Figure 9**. (a) Scatter plots between source/sink height (or in-canopy) vapor pressure deficit ($D_0$) and
atmospheric vapor pressure deficit ($D_A$) for two different classes of $g_C/g_A$ ratios over four PFTs, which
clearly depicts a strong coupling between $D_0$ and $D_A$ for low $g_C/g_A$ ratios. (b) Histogram distribution of
$g_C/g_A$ ratios over the four PFTs in the Amazon Basin (LBA tower sites). (c) Scatter plots between
$g_C/g_A$ ratio versus surface air temperature difference ($T_R - T_A$) for the four PFT during wet season and
dry season in the Amazon Basin (LBA tower sites).

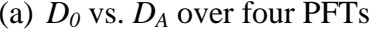

(a) $D_0$ vs. $D_A$ over four PFTs          (b) Distribution of $g_C/g_A$ ratio over four PFTs

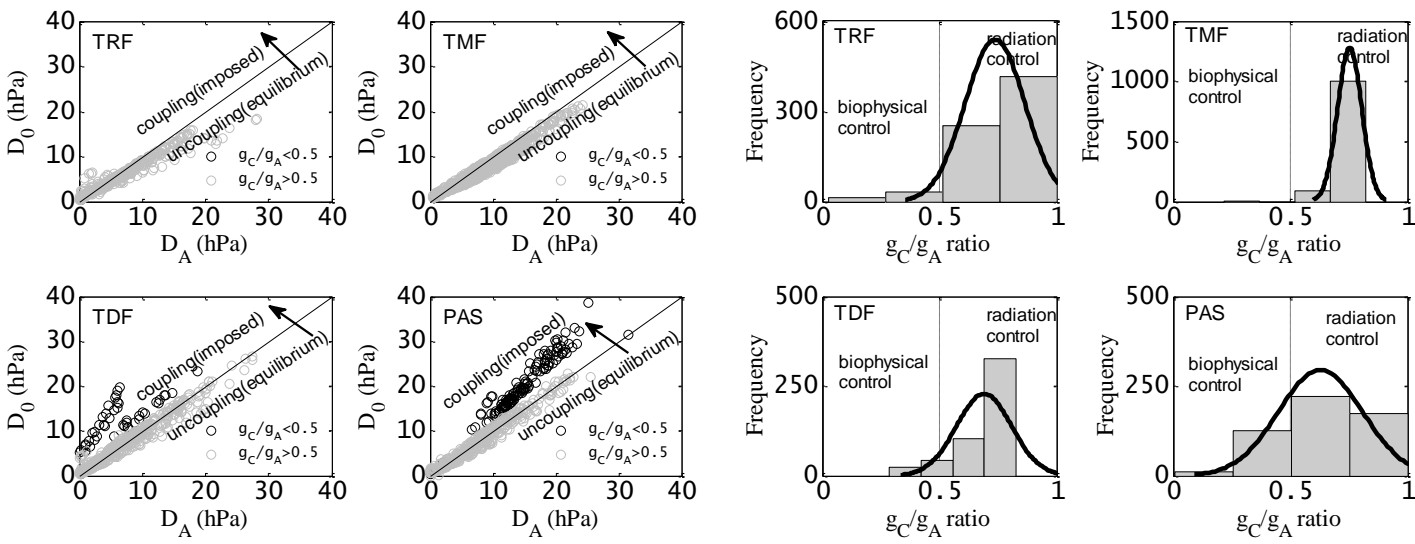

(c) $g_C/g_A$ vs. $T_R$-$T_A$ over four PFTs

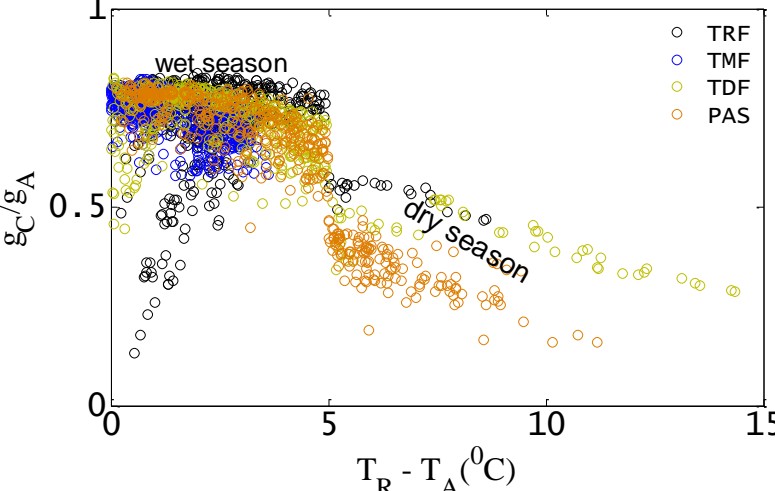






**Table A1**: Differences in the modeling philosophy of source/sink height vapor pressures ($e_0$, $e_0^*$) and

dewpoint temperature ($T_{SD}$), surface wetness ($M$), and $\alpha$ between STIC1.0, STIC1.1 and STIC1.2.

| Variable estimation | Principles | | |
|---|---|---|---|
| | **STIC1.0 (Mallick et al., 2014)** | **STIC1.1 (Mallick et al., 2015)** | **STIC1.2 (This study [Mallick et al., 2016])** |
| Saturation vapor pressure at source/sink height ($e_0^*$) | $e_0^*$ was approximated as the saturation vapor pressure at $T_R$. | Same as STIC1.0 | $e_0^*$ is estimated through numerical iteration by inverting the aerodynamic equation of $\lambda E$ (as described in appendix A2). $$e_0^* = e_A + \left[ \frac{\gamma \lambda E (g_A + g_C)}{\rho c_P g_A g_C} \right]$$ |
| Actual vapor pressure at source/sink height ($e_0$) | $e_0$ was empirically estimated from $M$ based on the assumption that the vapor pressure at the source/sink height ranges between extreme wet–dry surface conditions. | Same as STIC1.0 | $e_0$ is estimated as $e_0 = e_0^* - D_0$, where $D_0$ was iteratively estimated by combining PM with Shuttleworth-Wallace approximation (as described in appendix A2). $$D_0 = D_A + \left[ \frac{\{s\phi - (s + \gamma)\lambda E\}}{\rho c_P g_A} \right]$$ |
| Dewpoint temperature at source/sink height ($T_{SD}$) | $$T_{SD} = \frac{(e_S^* - e_A) - s_3 T_R + s_1 T_D}{(s_1 - s_3)}$$ $s_1$ and $s_3$ are the slopes of saturation vapor pressures at temperatures, approximated at $T_D$ and $T_R$, respectively. | Same as STIC1.0 | $T_{SD}$ is estimated through numerical iteration by inverting the aerodynamic equation of $\lambda E$ (as described in appendix A2). $$T_{SD} = T_D + \frac{\gamma \lambda E}{\rho c_P g_A s_1}$$ |
| Surface moisture availability ($M$) | As a stand-alone equation, without any feedback to $\lambda E$. | Same as STIC1.0 | A feedback of $M$ into $\lambda E$ is introduced and $M$ is iteratively estimated after estimating $T_{SD}$ (as described in appendix A2). |
| Priestley-Taylor parameter ($\alpha$) | As fixed parameter (1.26). | A physical equation of $\alpha$ is derived as a function of the conductances and $\alpha$ is numerically estimated as a variable. | A physical equation of $\alpha$ is derived as a function of the conductances and $\alpha$ is numerically estimated as a variable (eqn. A15) (as described in appendix A2). |

**Table A2**: Fundamental differences in the modeling principles between STIC1.2 and previous
approaches for characterising the biophysical controls on $\lambda E$ components.

| Biophysical states | Modeling principles | |
|---|---|---|
| | Parametric modeling (Ma et al., 2015; Chen et al., 2011; Kumagai et al., 2004) | STIC1.2 |
| $g_A$ | Either $g_A$ is assumed to be the momentum conductance ($g_M$) or estimated as a sum of $g_M$ and quasilaminar boundary-layer conductance ($g_B$).<br><br>$1/g_A = 1/g_M + 1/g_B$<br>$g_M = u^*/u$<br>$g_B = f\{Nusselt\ number,\ leaf\ dimension,\ thermal\ conductivity\ of\ air\ in\ boundary\ layer,\ u,\ kinematic\ viscosity,\ Reynolds\ number\}$<br><br>If $u^*$ is available from EC tower, it is directly used, otherwise $u^*$ is parametrized using Monin-Obukhov Similarity Theory (MOST).<br><br>Disadvantages: (1) MOST is only valid for an extended, uniform, and flat surface (Foken, 2006). MOST tends to fail over rough surfaces due to breakdown of the similarity relationships for heat and water vapor transfer in the roughness sub-layer, which results in an underestimation of the 'true' $g_A$ by a factor 1-3 (Thom et al., 1975; Chen and Schwerdtfeger, 1988; Simpson et al., 1998; Holwerda et al., 2012). (2) In the state-of-art $\lambda E$ modeling, the parametric $g_A$ sub-models are stand alone and empirical, and do not provide any feedback to $g_C$, aerodynamic temperature ($T_0$), and aerodynamic vapor pressures ($e_0$ and $D_0$). (3) Additional challenges in grid-scale or spatial-scale $g_A$ estimation are the requirements of numerous site specific parameters (e.g., vegetation height, measurement height, vegetation roughness, leaf size, soil roughness) and coefficients needed to correct the atmospheric stability conditions (Raupach, 1998). | Analytically retrieved by solving 'n' state equations and 'n' unknowns, with explicit convective feedback and without any wind speed (u) information.<br>In a hallmark paper by Choudhury and Monteith (1986), it is clearly stated that 'aerodynamic conductance determined by wind speed and roughness is assumed to be unaffected by buoyancy. Strictly, the aerodynamic conductance should be replaced by a term which accounts for radiative as well as convective heat transfer'. The role of $g_A$ is associated with the role of convection (Choudhury and Monteith, 1986) according to the surface energy balance principle as reflected in the derivation of eqn. (A4). Wind is generated as a result of the differences in atmospheric pressure which is a result of uneven surface radiative heating. Therefore, the aerodynamic conductance (and wind as well) is an effect of net radiative heating and there should be a physical relationship between these two.<br><br>Advantages: (1) STIC1.2 consists of a feedback describing the relationship between $T_R$ and $\lambda E$, coupled with canopy-atmosphere components relating $\lambda E$ to $T_0$ and $e_0$. (2) Supports the findings of Villani et al. (2003) which stated that during unstable surface layer conditions the major source of net available energy is located at the canopy top and drives the convective motion in the layers above. |
| $g_C$ | (a) If $\lambda E$ measurements are available from the EC towers, $g_C$ is estimated by inverting the PM equation. None of these approaches allow independent quantification of biophysical controls of $\lambda E$ as $g_C$ is constrained by $\lambda E$ itself.<br>(b) Sometimes $g_C$ is modelled either by coupled leaf-scale photosynthesis models (Ball et al., 1987; Leuning, 1995) or $g_C$ is estimated from standalone empirical models (Jarvis, 1976) | Analytically retrieved by solving 'n' state equations and 'n' unknowns where physical feedbacks of $g_A$, soil moisture, and vapor pressure deficit are embedded (as explained in STIC1.2 equations in Appendix). |



**Figure A1.** Schematic representation of one-dimensional description of STIC1.2. In STIC1.2, a feedback is established between the surface layer evaporative fluxes and source/sink height mixing and coupling, and the connection is shown in dotted arrows between $e_0$, $e_0^*$, $g_A$, $g_C$, and $\lambda E$. Here, $r_A$ and $r_C$ are the aerodynamic and canopy (or surface in case of partial vegetation cover) resistances, $g_A$ and $g_C$ are the aerodynamic and canopy conductances (reciprocal of resistances), $e_S^*$ is the saturation vapor pressure of the surface, $e_0^*$ is the saturation vapor pressure at the source/sink height, $T_0$ is the source/sink height temperature (i.e. aerodynamic temperature) that is responsible for transferring the sensible heat ($H$), $e_0$ is the source/sink height vapor pressure, $e_S$ is the vapor pressure at the surface, $z_0$ is the roughness length, $T_R$ is the radiometric surface temperature, $T_{SD}$ is the source/sink height dewpoint temperature, $M$ is the surface moisture availability or evaporation coefficient, $R_N$ and $G$ are net radiation and ground heat flux, $T_A$, $e_A$, and $D_A$ are temperature, vapor pressure, and vapor pressure deficit at the reference height ($z_R$), $\lambda E$ is the latent heat flux, $H$ is the sensible heat flux, respectively.

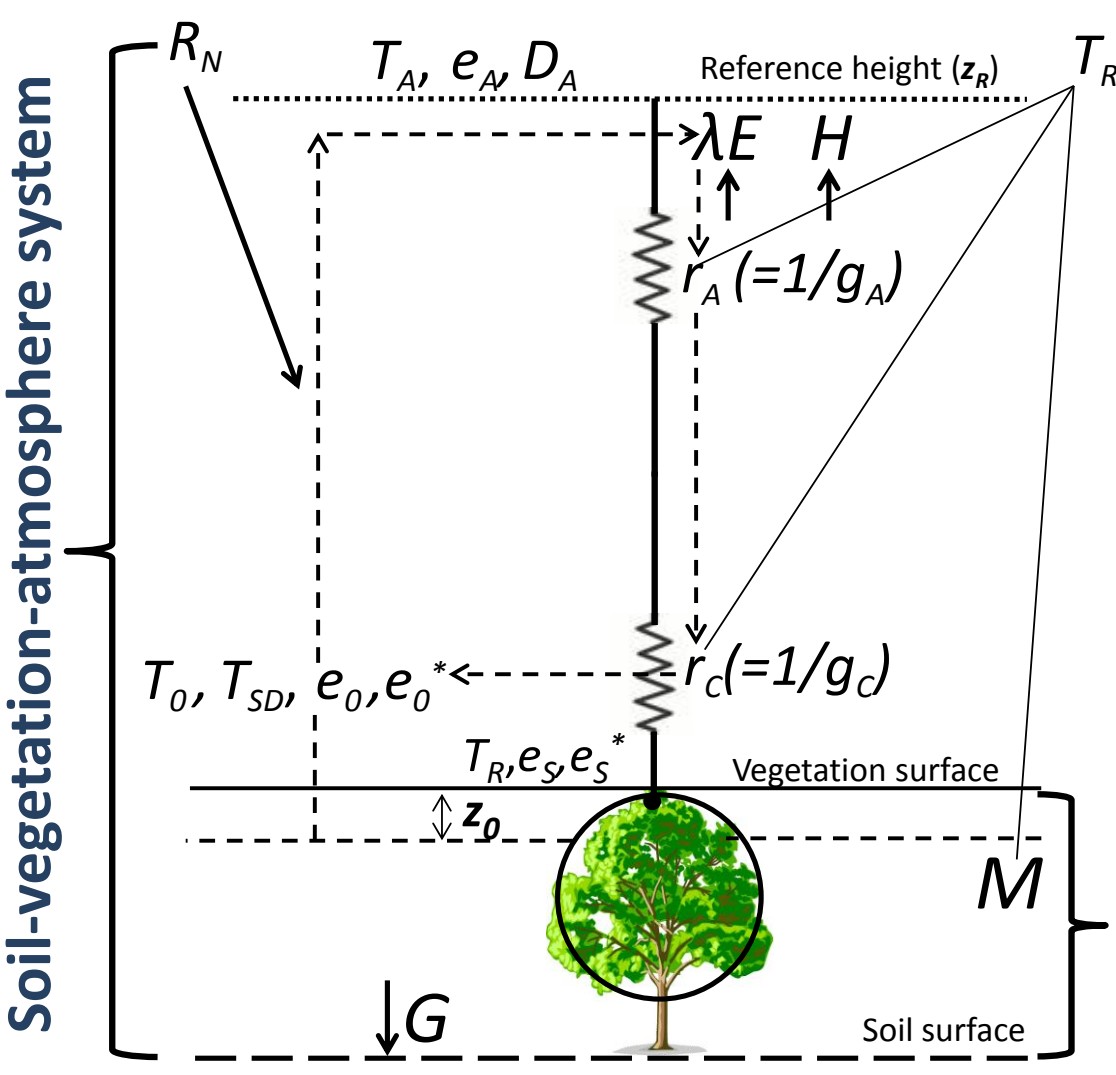

**Figure A2.** Aerodynamic temperature obtained from STIC1.2 ($T_{0\text{-}STIC}$) versus radiometric surface
temperature ($T_R$) over two different biomes in the Amazon basin. The regression equation of line of
best fit is $T_{0\text{-}STIC} = 0.67(\pm0.10)T_R + 10.59\ (\pm2.79)$ with $r = 0.65$.

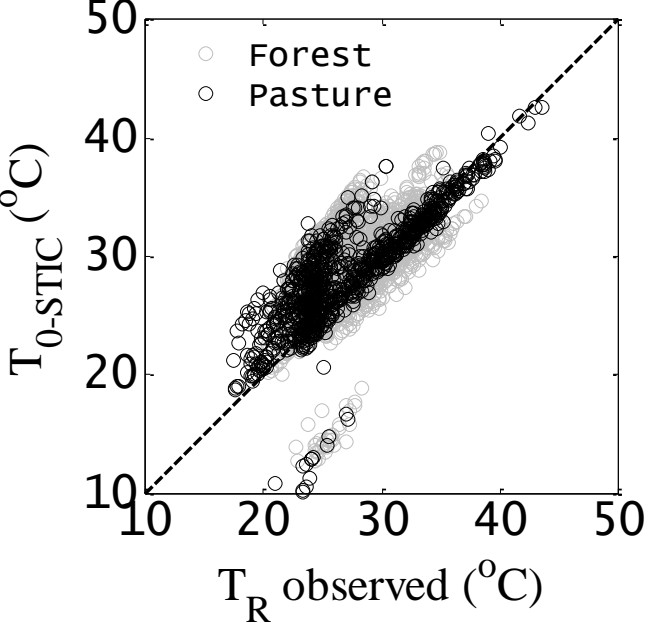
















**Figure A3.** (a) Convergence of the iteration method for retrieving the source/sink height (or in-
canopy) vapor pressures ($e_0$ and $D_0$) and Priestley-Taylor coefficient ($\alpha$). (b) Convergence of the
iteration method for retrieving the surface wetness ($M$) and source/sink height dewpoint temperature
($T_{SD}$). The initial values of $\lambda E$, $g_A$, $g_C$, and $T_0$ were determined with $\alpha = 1.26$. The process is then
iterated by updating $\lambda E$, $e_0$, $D_0$, $M$, $T_{SD}$, and $\alpha$ in subsequent iterations with the previous estimates of
$g_A$, $g_C$, and $T_0$.

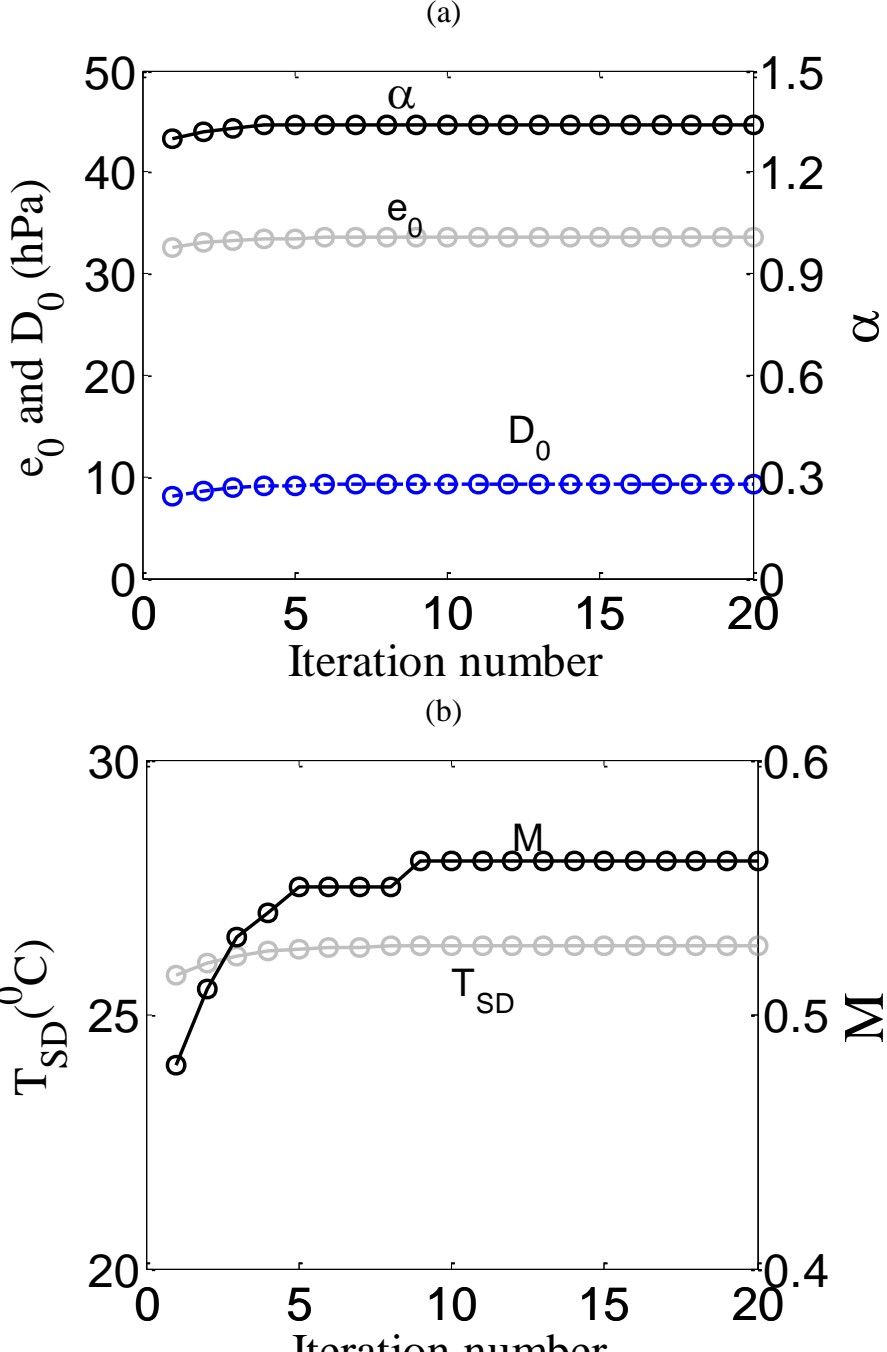
