# Peer review of "Canopy-scale biophysical controls of transpiration and"

_Hydrology and Earth System Sciences, 2015_

## Referee Comment (RC1) · Anonymous Referee #1 · 1 Apr 2016

Review Mallick HESS Major comments My main concern is that the manuscript does not present any new theory (and on top of that uses an approach (STICS) that in my opinion is misguided, despite the fact that it has been published). STICs is misguided because it ends up with an aerodynamic conductance that does not depend on wind-speed and introduces a soil moisture stress term that only depends on atmospheric variables. The paper then uses Amazonian micrometeorological data to compare a range of gA and gC terms. No measurements of gC are used to provide verification. A large number of plots are then presented where STICS variables are plotted against meteorological variables in a host of different ways. I am not surprised to see that these dependencies exist as all of them are intrinsic to the model. Also, because all of them

are interdependent I am not sure how much realism there ultimately is in the findings. Furthermore STICS assumes that T0=TR and yet the manuscript does not mention the potential implications of this assumption, nor the fact that considerable errors can be made when measuring TR. In the end I feel I have learned little new and what has been presented is tentative and therefore potentially misleading. This is underlined by sentences (line 325-329) such as "The evaluation of the conductances and surface energy fluxes indicates some efficacy for the STIC derived fluxes and conductance estimates ..... As a result we feel some justification for exploring the canopy-scale biophysical controls on $\lambda$ET and $\lambda$EE generated through the STIC framework".

Detailed comments Line 81-82: "An intensification of the Amazon hydrological cycle was observed in the past two decades characterised by increased temperatures and more frequent droughts and floods" How are increased air (?) temperatures directly linked to hydrological cycle? If it is surface temperatures then say this, but this would mean a decreased ET (hence the floods?) Line 86: "the Amazon forest may become an increasing carbon source". Should this be " increasingly become a net source of carbon? Line 97-104: I disagree with the final point made in this section: GC does not include the conductance relating to bare soil. If you would have called it the surface conductance instead and defined it via the PM Big leaf equation I would have agreed. Line 111-126 are stating the obvious. Where is this going? Line 136: Why is the partitioning between soil evaporation and transpiration deemed so important in the Amazon? Soil evaporation must only make up a small part of total ET. Will this soil term affect flooding, atmospheric circulation etc. I highly doubt this. Line 141-143: "Given the persistent risk of deforestation, the ecophysiological changes of different plant functional types (PFTs) are expected to be reflected in gA and gC and $\lambda$EE and $\lambda$ET". I really do not understand what is meant by this sentence. Line 154-157: The surface temperature is already implicit in the PM equation as it combines the energy balance with bulk transfer equations. Line 179-181: "The retrieval of gA, gC, and $\lambda$E are based on finding a 'closure' of the PM equation using the STIC framework". In my opinion, the PM is already closed, see my point above. It calculates ET from Rn-G, and

H is implicitly in there. Please study books such as those by Hamlyn Jones to see how PM equation is derived. Line 184: This should be 'radiative temperature'. Line 203: You have now tacitly assumed that T0 = TR. There is a host of literature references that will tell you otherwise. Line 204-205: PM equation is already closed. This assumption of energy balance closure is implicit in its derivation. But maybe I do not understand what you mean by this statement. Line 225-227: "The estimates of $\lambda$EE in the current method consists of aggregated contribution from both interception and soil evaporation, and no further attempt is made to separate these two components". This is a considerable weakness in the approach seeing leaf area index and hence interception is so large for large parts of the Amazon and soil evaporation will be negligible. You are making this point yourself a few sentences later (line 232) Also: these two types of evaporation fluxes take place at very different source heights, so their GA will be very different, further weakening your approach. Line 285: "The conductances showed a marked diurnal variation expressing their overall dependence on net radiation, vapor pressure deficit, and surface temperature". What conductance are you referring to here? gA or gC? Or both? Note that gA generally does not depend on net radiation or VPD etc., although it does in STICS.

---

## Referee Comment (RC2) · Anonymous Referee #2 · 12 Apr 2016

This manuscript describes a study that infers stomatal and aerodynamic conductances from eddy flux observations. I think in general, this study is innovative and presents novel material, so that in principle it should be published. I hesitate recommendation for publication mostly because I am not entirely convinced by the approach and I feel that this needs revision. Hence, I recommend major revisions, although I do not think that it necessarily involves a lot of work to address the points below.

Major points:

My major problem with the manuscript is that I do not understand the approach, so that it is difficult to assess its plausibility. While the main equations are provided in the manuscript (eqn. 2-5), there is no more description on where these equations come

from, except for references to prior papers by the authors. I think it is necessary to at least provide a description at a qualitative level where these equations come from. The point where I really got confused is that eqn. 5 uses the Priestley-Taylor coefficient, which is an empirical coefficient in an evaporation equation that is rather different from the Penman Monteith equation. Where does this coefficient suddenly come from? I find this quite confusing, and it needs at least a minimum of explanation as it is not obvious.

What I also do not understand is why an iterative scheme is needed. Can't one simply use the observations and use a simple partitioning based on the Bowen ratio? It would be good to describe what the differences and similarities are to previous approaches. As the authors propose a new approach, they should provide a better description that is easier to follow of what is being done.

Minor points:

- The authors refer to $\lambda E$ as evaporation, which, technically speaking, is the latent heat flux, not evaporation.

- Abstract: dry and wet conditions — do you mean conditions in which water is not limiting vs. limiting, or precipitation vs. radiation driven conditions?

- Biophysical control — should be briefly explained by what this means.

- Line 145: I wonder why approaches hat directly link stomatal conductance to photosynthesis are not mentioned, such as Ball-Berry?

- Line 194: Where do these "state equations" come from? Referring to previously published work is fine for derivations, but the description should still mention what the concepts are that are behind these equations.

- A table of variables would help.

- Line 238: I think the authors assume that the conductances to momentum, sensible

and latent heat are identical. If this is the case, it should be mentioned, as there are also approaches to surface exchange that do not treat them as being identical.

- Line 331: As the typical readers of HESS are not micrometeorologists, it would be useful to explain the decoupling coefficient in some more detail. This will help to interpret the following results.

- Line 422: To what extent could these discrepancies between how conductances are derived also relate to actual differences in the conductances for momentum vs. heat?

- Line 498: The authors should stick to the same ratio gA/gC for easier interpretation.

---

## Author Comment (AC1) · 6 May 2016

**Reviewer 1 (R1):**

We thank R1 for the comments and firmly object the statement on 'misguidance' (below) which appears to be potentially misleading particularly when a large part of the community have outright enthusiasm in developing analytical approaches for estimating terrestrial evapotranspiration (E) (or latent heat flux,  $\lambda E$ ) and sensible heat fluxes (H) to overcome the ambiguities associated with parameterizations of aerodynamic (gA) and canopy surface conductances (gC) (**Kleidon et al., 2014; Matheny et al., 2014; Ershadi et al., 2015**). Besides, we would like to clarify that the abbreviation for our model framework is "STIC" and not "STICS" as referred to by R1.

**Major comments:**

(1) My main concern is that the manuscript does not present any new theory (and on top of that uses an approach (STICS) that in my opinion is misguided, despite the fact that it has been published).

Summary response: R1 claims that the manuscript does not present any new theory. To the opinion of the authors, this claim is flawed because STIC (Surface Temperature Initiated Closure) introduced a novel analytical method to integrate radiometric surface temperature ( $T_R$ ) into the Penman-Monteith model to overcome the limitations associated with empirical parameterisations of the aerodynamic and canopy surface conductances ( $g_A$  and  $g_C$ ) which are not directly measurable either at the canopy-scale or at the large spatial grid-scale. To our knowledge, this research objective is unquestionably novel and the behavior of the analytically retrieved canopy-scale conductances as well as transpiration are compliant with the theory earlier postulated in the literatures (Jarvis and McNaughton, 1986; Monteith, 1995, Raupach, 1998). In addition to its simplicity, STIC has the capabilities for generating spatially explicit surface energy fluxes and independent of submodels for boundary layer developments.

Detailed response: The most tangible accomplishment and uniqueness of STIC (STIC1.2) is the physical integration of land surface temperature (i.e., radiometric surface temperature,  $T_R$ ) into a combined framework of the Penman-Monteith (PM) and Shuttleworth-Wallace (SW) model for simultaneously estimating E, H,  $g_A$ ,  $g_c$ , surface moisture status, and E components (evaporation,  $E_E$  and transpiration,  $E_T$ ). The intrinsic link between the PM-SW model and  $T_R$  emanates through the first-order dependence of the biophysical conductances ( $g_A$  and  $g_C$ ) on the aerodynamic temperature ( $T_0$ ) (through  $T_R$ ) and soil moisture (through  $T_R$ ). However, until now the explicit use of  $T_R$  in the PM-SW model was hindered due to the unavailability of any direct method to integrate  $T_R$  into these models, and, furthermore, due to the lack of physical models expressing biophysical states of vegetation as a function of  $T_R$ . Therefore, the majority of the E modeling approaches strongly rely on surface reflectance and meteorology; and thermal approaches require significant parameterization of land surface properties (e.g.,  $g_A$  and  $g_C$ ) which are very empirical in nature (Schulz and Beven, 2003; Prihodko et al., 2008; Bonan et al., 2014; Ershadi et al., 2015).

To bridge this gap, the STIC methodology was developed as a novel thermal-based biophysical scheme for directly estimating E over terrestrial ecosystems by leveraging the combined strength of  $T_R$  observations and physically-based models

(Mallick et al., 2014; 2015). In addition to physically integrating  $T_R$  observations into a combined PM-SW framework, STIC1.2 also establishes of a feedback loop describing the relationship between  $T_R$  and E, coupled with canopy-atmosphere components relating E to aerodynamic temperature ( $T_0$ ) and vapor pressure ( $e_0$ ) (in STIC1.2). By blending  $T_R$  with standard SEB principles and vegetation-atmosphere exchange biophysics, STIC formulates multiple state equations in order to eliminate the need of exogenous parametric submodels for the surface and aerodynamic conductances, aerodynamic temperatures, and land-atmosphere coupling. Instead these 'internal states' are numerically retrieved. Originally designed for application to thermal remote sensing data from Earth observation sensors, the STIC framework exploits observations of  $T_R$ , radiative, and meteorological variables including net radiation ( $R_N$ ), ground heat flux (G), air temperature ( $T_A$ ), relative humidity ( $R_H$ ) or vapor pressure ( $e_A$ ) at a reference level above the surface, and can be applied over any ecosystem, provided the necessary input variables are available.

**We hope this extended summary will help expanding R1's constrained judgement on STIC.**

(2) STICs is misguided because it ends up with an aerodynamic conductance that does not depend on wind speed and introduces a soil moisture stress term that only depends on atmospheric variables.

Summary response: R1 claims that STIC is misguided due to two reasons. According to R1, the first reason should be that aerodynamic conductance does not depend on wind speed  $(W_s)$ . It should be noted that, in one of the hallmark papers by Choudhury and Monteith (1986), it is clearly stated that 'aerodynamic conductance determined by wind speed and roughness is assumed to be unaffected by buoyancy'. Strictly, the aerodynamic conductance should be replaced by a term which accounts for radiative as well as convective heat transfer'. Although incorporation of  $W_s$  data has almost become a dogma (Foken, 2006) in the field of land surface energy balance modelling, there are several widely accepted evapotranspiration estimation approaches that do not incorporate  $W_s$ , for example, maximum entropy production approach (Kleidon et al., 2014), evaporative fraction approach (Jiang and Islam, 2001; Batra et al., 2006), complementary relationship approach (Venturini et al., 2008) etc.

On the second claim of R1 regarding the estimation of soil moisture stress that only depends on atmospheric variables, the claim is not substantiated because the water stress factor was estimated by combining the radiometric surface temperature ( $T_R$ ) with air temperature ( $T_A$ ), dewpoint temperature ( $T_D$ ), and near surface dewpoint temperature ( $T_{SD}$ ) as explained in Mallick et al. (2015). The procedure is also briefly explained in the appendix of the current manuscript

Detailed response: Given the importance of  $g_A$  for evapotranspiration (E) estimates there are overriding cases for getting this 'right' in the surface energy balance models (Prihodko et al., 2008; Hong et al., 2010; Gibson et al., 2011; Holwerda et al., 2012; Gokmen et al., 2012; Morillas et al., 2013). However, if the empirical  $g_A$  models currently provide accurate estimates of E for the wrong reasons then this status quo has to be questioned, especially as errors like this might become important when predicting E under future boundary conditions. Furthermore, it is not obvious that  $W_s$ -based models currently provide accurate estimates of  $g_A$ , in particular at the grid-scale (e.g., 1 km and above) where bundles of site specific parameters are required (which cannot be measured). We would like to bring forward the following arguments concerning WS-based gA estimation.

- (a) As highlighted in several studies (Monteith and Unsworth, 2008; Holwerda et al., 2012), the momentum transfer equation for  $q_A$  estimation based on the Monin-Obukhov Similarity Theory (MOST) only holds for an extended, uniform, and flat surface (Foken, 2006). MOST tends to fail over rough surfaces due to breakdown of the similarity relationships for heat and water vapour transfer in the roughness sub-layer, which results in an underestimation of the 'true'  $g_A$  by a factor 1-3 (Thom et al., 1975; Chen and Schwerdtfeger, 1988; Simpson et al., 1998; Holwerda et al., 2012). Despite some of the boundary layer studies based on parameterized friction velocity (u\*) demonstrated the validity of MOST (subjected to tuning and calibration) (Harman and Finnigan, 2007; 2008), a considerable number of studies have casted scepticism on the validity of u\* parameterization in the framework of MOST (Foken, 2006; Holwerda et al., 2012; van Dijk et al., 2015). It is imperative to mention that gA is one of the main anchors in the PM-SW model because it not only appears in the numerator and denominator of these models,  $g_A$  also provides feedback to  $g_c$ , aerodynamic temperature, and vapor pressure (seminal paper of Jarvis and McNaughton, 1986). Therefore, the estimates of E and interception evaporation (Ei) in the PM-SW framework are robustly sensitive to parameterization of gA and stable E estimates might be possible if gA estimation is unambiguous (Holwerda et al., 2012; van Dijk et al., 2015). Consequently, our aim was to find analytical solution of  $g_A$ , and through algebraic reorganisation of surface energy balance equation we are able to do so. Given the lack of consensus in the community on the 'true' gA, we treat STIC1.2 derived nonparametric gA to be the aerodynamic conductance that satisfies the PM-SW equation for estimating evaporative fluxes.
- (b) In the state-of-art E modeling, the parametric gA sub-models are stand alone and empirical, and do not provide any feedback to the canopy (or surface) conductances (gc), aerodynamic temperature (T0), and aerodynamic vapor pressure deficit (D0). However, gA is an internal state that provides physical feedback to E and H by influencing T0, D0, and gc. Large gA indicates small gradients of vapor pressure deficit between the air and canopy boundary layer and hence strong coupling between canopy and atmosphere (Jarvis and McNaughton, 1986). These biophysical interactions are entirely overlooked in the land surface parameterizations of gA (but are included in STIC). STIC1.2 consists of a feedback describing the relationship between TR and E, coupled with canopy-atmosphere components relating E to T0 and e0. The equations are explicitly stated in the Appendix (eq. A2 to A8) of the manuscript and the detailed descriptions are in L595 to L627.
- (c) Additional challenges in grid-scale or spatial-scale gA estimation are the requirements of numerous site specific parameters (e.g., vegetation height, measurement height, vegetation roughness, leaf size, soil roughness) and coefficients needed to correct the atmospheric stability conditions (Raupach, 1998). These informations are required to fulfil the set of assumption established around 1960's, that can and should be questioned by the community if we want to make science advance in the field of surface energy balance modeling.
- (d) The enhanced errors in E estimates in water-limited regions due to uncertain gA parameterisation (Gibson et al., 2011; Timmermans et al., 2013; Morillas et al., 2013; Castellvi et al., 2016) and repeated adjustment of different vegetation as well as soil parameters in the conductance equations to obtain a better E validation (Gokmen et al., 2012) questions the validity of wind driven non-stationary gA parameterisations. It solicits

for revisiting the state-of-art  $g_A$  parameterisations and rethinking to develop a calibration independent  $g_A$  modelling framework.

(e) The credibility of STIC1.2  $g_A$  estimates is shown in the figures (Fig. 1 and Fig. 2). While Fig. 1a in the manuscript illustrates the differences in the  $g_A$  magnitude between forest and pasture, Fig. 2a displays an independent comparison of STIC- $g_A$  versus u\*-based  $g_A$ . Fig. 2e and 2f showed that  $T_R$ , vapor pressure deficit ( $D_A$ ), and net available energy ( $\phi$ ) (difference between net radiation,  $R_N$  and ground heat flux, G) can explain 42% to 83% variability of the u\*-based  $g_A$ . These correlations and scatterplots between u\*-based  $g_A$ with radiative and meteorological variables clearly emphasize the explanatory power of these variables to characterise wind-driven  $g_A$  and the appropriateness of deriving an analytical  $g_A$  without wind speed. This also supports the findings of Villani et al. (2003) which stated that during unstable surface layer conditions the major source of net available energy is located at the canopy top and drives the convective motion in the layers above. Hence, the value of  $g_A$  is not only controlled by wind speed as advocated by R1.

Regarding R1's claim on characterising the water stress (M) as a function of atmospheric variables, we would like to draw further attention that the stress function was estimated as a function of  $T_R$ , air temperature ( $T_A$ ), dewpoint temperature ( $T_D$ ), and near surface dewpoint temperature ( $T_{SD}$ ) (Mallick et al., 2015) which is clearly stated in the Appendix (L617). In STIC1.2,  $T_{SD}$  was estimated in an iterative mode to establish a feedback between the water stress,  $T_R$ ,  $T_{SD}$ , and evapotranspiration. Since R1 acknowledges to be aware of the published STIC methodology, the immediate fact supposed to be reflected in R1's understanding are the titles of the two previous manuscripts 'A surface temperature initiated closure of the surface energy balance fluxes' and 'Reintroducing radiometric surface temperature into the Penman-Monteith formulation' published in Remote Sensing of Environment and Water Resources Research, respectively.

(3) The paper then uses Amazonian micrometeorological data to compare a range of gA and gC terms. No measurements of gC are used to provide verification.

Response: This exercise could not be performed as direct canopy-scale  $g_C$  observations are not possible with current measurement techniques. Although leaf-scale measurements of  $g_C$ are relatively straightforward, these values are not comparable to values retrieved at the canopy-scale. However, assuming u\*-based  $g_A$  as baseline aerodynamic conductance, we had estimated canopy-scale  $g_C$  by inverting the PM equation ( $g_{C-INV}$ ) to evaluate  $g_{C-STIC}$ . The comparison between  $g_{C-STIC}$  and  $g_{C-INV}$  over forest and pasture is illustrated in Fig. 3a, and the results are discussed in L305 to L315 of the manuscript.

We shall make this point explicit in revised version of the manuscript.

(4) A large number of plots are then presented where STICS variables are plotted against meteorological variables in a host of different ways. I am not surprised to see that these dependencies exist as all of them are intrinsic to the model. Also, because all of them are interdependent I am not sure how much realism there ultimately is in the findings.

Summary response: We do not agree with this statement of R1. Firstly, comparing different  $g_A$  estimates, linking the wind driven  $g_A$  estimates with some independent

variables (Fig. 2), and STIC driven  $g_A$  estimates with some interdependent variables (Fig. 6 to Fig. 8) is not a matter of choice, but a necessity, as it evident to any reader. The same is applicable to Fig. 6 to Fig. 8 for  $g_C$ . Secondly, despite the transpiration and evaporation estimates are interdependent with  $g_C$  and  $g_A$  (as shown in Fig. 6 to Fig. 8); the figures reflect the credibility of the conductances as well as transpiration estimates by realistically capturing the hysteretic behavior between biophysical conductances and water vapor fluxes which is frequently observed in natural ecosystems (Zhang et al., 2014, Renner et al., 2016). Fig. 8 (a, b) also affirms that the conductance-transpiration-vapor pressure deficit relationships are compliant with the stomatal feedback-response theory earlier postulated from observational evidences (Monteith, 1995).

**Detailed response**: Fig. 2 illustrates the diagnostic potential of thermal ( $T_R$ ), radiative ( $\phi$ ), and meteorological ( $D_A$ ) variables to explain the wind driven  $g_A$  variability (wind driven  $g_A$  is independently estimated).

**Fig. 6 and 7** explains the 'hysteresis' between transpiration and conductances which shows the degree of hysteresis was larger in the dry season than in the wet season. These results are compliant with the theories earlier postulated from observations that the magnitude of hysteresis depends on the radiation-vapor pressure deficit lag, while the soil moisture availability is a key factor modulating the hysteretic transpiration-vapor pressure deficit relation as soil moisture declines (Zhang et al., 2014; O'Grady et al., 1999; Jarvis and McNaughton, 1986). This shows that despite independent of any predefined hysteretic function, the interdependent conductance-transpiration hysteresis is still captured in STIC1.2 (which are generally observed in natural ecosystems).

**Fig. 8 (a and b) confirms the 'stomatal feedback-response' hypothesis as postulated by Lt. John Monteith (Monteith, 1995)**, which states that a decrease in stomatal conductance with increasing vapor pressure deficit is caused by a direct increase in transpiration (Monteith, 1995) and stomata responds to the changes in the air humidity by sensing transpiration, rather than vapor pressure deficit. This feedback mechanism is found because of the influence of vapor pressure deficit on both stomatal conductance and transpiration, which in turn changes vapor pressure deficit by influencing the air humidity (Monteith, 1995).

**Fig. 8c** shows the complex interaction between  $g_c$ , radiometric surface temperature ( $T_R$ ) and vapor pressure deficit ( $D_A$ ). This also answers why different parametric  $g_c$  models produce divergent results.

**Fig. 8d** emphasizes the behaviour of  $g_A$  according to existing theory that under extremely high atmospheric turbulence (i.e., high  $g_A$ ), a close coupling exists between the surface and the atmosphere, which causes  $T_R$  and  $T_A$  to converge (i.e.,  $T_R - T_A \rightarrow 0$ ).

(5) Furthermore STICS assumes that T0 = TR and yet the manuscript does not mention the potential implications of this assumption, nor the fact that considerable errors can be made when measuring TR.

Summary response: This comment by R1 is incorrect. In STIC1.2,  $T_0$  is analytically estimated by integrating  $T_R$  into a combined PM-SW framework. The analytical

expression of  $T_0$  is dependent on M and the estimation of M is based on  $T_R$  as described in the Appendix of the current manuscript. T0 is a nonlinear function of TR in STIC and they are not assumed equal. To further address R1's point on the assumption that  $T_0=T_R$ , we show here an intercomparison of retrieved  $T_0$  versus  $T_R$  for forest and pasture (figure below). This indicates the distinct difference of the retrieved  $T_0$  from  $T_R$  for the two different biomes, which proves that R1's claim to be invalid.

We will include the figure of  $T_0$  versus  $T_R$  and necessary descriptions in the Appendix of revised manuscript. We will also address this point explicitly (i.e.,  $T_0 \neq T_R$ ) in the theoretical section (section 2.1) of the manuscript.

**Figure: Aerodynamic** temperature obtained from STIC1.2 versus (**T**0-STIC) radiometric surface temperature (TR) over two different biomes in the Amazon basin. The regression equation of line of best fit is  $T_{0-STIC} = 0.67(\pm 0.10)T_{R} +$ 10.59 (±2.79) with r = 0.65

Detailed response: One of the core objectives of the original STIC formulation was to physically integrate  $T_R$  into the PM model to constrain the conductances. This is done by estimating an aggregated surface moisture availability (or water stress factor) which is an emphatic function of  $T_R$ . A detailed description of the STIC state equations is given in Mallick et al. (2015) and novel part of STIC1.2 is described in the Appendix of the current manuscript.

(6) In the end I feel I have learned little new and what has been presented is tentative and therefore potentially misleading. This is underlined by sentences (line 325-329) such as "The evaluation of the conductances and surface energy fluxes indicates some efficacy for the STIC derived fluxes and conductance estimates ..... As a result we feel some justification for exploring the canopy-scale biophysical controls on ET and EE generated through the STIC framework".

Response: We do not agree with the reviewer's impression. The major novelties of the present manuscript are as follows:

The bafflement of evapotranspiration originates from a supply-demand chain reaction where net radiation and soil moisture represents the supply side and the atmospheric vapor pressure deficit represents the demand side. This supply-demand chain reaction accelerates the biophysical feedbacks in evapotranspiration and understanding these biophysical feedbacks is necessary to assess the terrestrial biosphere response to water availability. In this context, the entire manuscript is about understanding the canopy-scale biophysical controls on transpiration and **evaporation over the Amazon basin.** The two critical biophysical state variables (i.e.,  $g_A$  and  $g_C$ ) in the PM equation are the unobserved components which cannot be measured directly. Therefore, we explored the radiative (net radiation and ground heat flux), meteorological (air temperature and relative humidity), and thermal (radiometric surface temperature) information, and developed the STIC framework to analytically estimate these variables in an internally consistent manner (as described in the manuscript). However, before understanding the controls of  $g_A$  and  $g_C$  on transpiration and evaporation, some indirect evaluation of these two biophysical states was necessary, and hence the sentences in line 325 to 329 are justified.

**Detailed comments:**

Line 81-82: "An intensification of the Amazon hydrological cycle was observed in the past two decades characterised by increased temperatures and more frequent droughts and floods" How are increased air (?) temperatures directly linked to hydrological cycle? If it is surface temperatures then say this, but this would mean a decreased ET (hence the floods?)

Response: We agree. We will make the necessary correction in the revised manuscript as "An intensification of the Amazon hydrological cycle was observed in the past two decades characterised by increased air and land surface temperatures and more frequent droughts"

Line 86: "the Amazon forest may become an increasing carbon source". Should this be "increasingly become a net source of carbon?

**Response: We will clarify this in the revised version on the manuscript.**

Line 97-104: I disagree with the final point made in this section: GC does not include the conductance relating to bare soil. If you would have called it the surface conductance instead and defined it via the PM Big leaf equation I would have agreed.

Response: We do not agree. For a dense canopy,  $g_{C}$  in the PM equation represents the canopy surface conductance. Although it is not equal to the canopy stomatal conductance, it contains integrated information of the stomata. For a heterogeneous landscape,  $g_{C}$  in the PM equation is an aggregated surface conductance containing information of canopy and soil.

**Lines 111-126 are stating the obvious. Where is this going?**

Response: Line 111 – 126 explained the unresolved challenges and problems associated with  $g_A$  and  $g_C$  parameterisations. If these are obvious, R1's previous claims on  $g_A$  appear to be unfounded.

These statements are needed to recognize the need of a non-parametric  $\mathbf{g}_A$  and  $\mathbf{g}_C$  modeling framework.

Line 136: Why is the partitioning between soil evaporation and transpiration deemed so important in the Amazon? Soil evaporation must only make up a small part of total ET. Will this soil term affect flooding, atmospheric circulation etc. I highly doubt this. Response: We intended to address 'evaporation', not 'soil evaporation'. In the Amazon forest, although the soil evaporation has negligible contribution, it is the interception evaporation that has substantial contribution in the total evaporative fluxes, and, therefore the partitioning of 'evaporation ( $\lambda E_E$ )' and 'transpiration ( $\lambda E_T$ )' is significant.

**Line 141-143: "Given the persistent risk of deforestation, the ecophysiological changes of different plant functional types (PFTs) are expected to be reflected in gA and gC and EE and ET". I really do not understand what is meant by this sentence.**

Response: Deforestation alters the radiation interception, surface temperature, surface moisture, associated meteorological conditions, and vegetation biophysical states. Conversion from forest to pasture will change the  $g_A/g_C$  ratio of the ecosystem and dry-wet evapotranspiration partitioning.

Necessary changes will be incorporated in the revised version of the manuscript.

**Line 154-157: The surface temperature is already implicit in the PM equation as it combines the energy balance with bulk transfer equations.**

Response: The surface temperature ( $T_R$ ) was eliminated from the derivation of the PM equation by expressing the slope of the saturation vapor pressure at ambient air temperature. However, in the seminal paper titled 'Evaporation and Surface Temperature' (Monteith, 1981), Lt. John Monteith described the role of leaf temperature in constraining the biophysical conductances. Although  $T_R$  is implicit in the net radiation (Rn), which appears in the numerator of the PM equation, it may be noted that Rn has a relatively weak dependence on  $T_R$  (compared to  $T_R$  sensitivities of soil moisture and E). No universally agreed formulation is available that physically constrains  $g_A$  and  $g_C$  by using  $T_R$ . Development of STIC is based on the assumption that the intrinsic link between the PM-SW model and  $T_R$  emanates through the first-order dependence of the biophysical conductances on aerodynamic temperature ( $T_0$ ) and soil moisture (through  $T_R$ ). Hence, the conductances are explicitly constrained by using  $T_R$  information as described in the manuscript.

**Line 179-181: "The retrieval of gA, gC, and E are based on finding a 'closure' of the PM equation using the STIC framework". In my opinion, the PM is already closed, see my point above. It calculates ET from Rn-G, and H is implicitly in there. Please study books such as those by Hamlyn Jones to see how PM equation is derived.**

Response: The PM equation is 'closed' upon the availability of canopy-scale measurements of the two unobserved biophysical conductances ( $g_A$  and  $g_C$ ) and if we assume the empirical models of  $g_A$  and  $g_C$  to be authentic. However, neither  $g_A$  nor  $g_C$  can be measured at the canopy-scale or at larger spatial scales. Furthermore, as shown by several recent studies (Matheny et al., 2014; van Dijk et al., 2015) a most appropriate or correct  $g_A$ - $g_C$  model is currently not available. This implies that a **true 'closure' of the PM equation is only possible upon analytical estimation of the conductances.**

Line 184: This should be 'radiative temperature'.

Response: Necessary changes will be incorporated.

Line 203: You have now tacitly assumed that T0 = TR. There is a host of literature references that will tell you otherwise.

Response: The explanation is already provided above; there is no assumption on the equality between  $T_R$  and  $T_0$ . We will address this point explicitly (i.e.,  $T_0 \neq T_R$ ) in section 2.1 of the revised manuscript.

Line 204-205: PM equation is already closed. This assumption of energy balance closure is implicit in its derivation. But maybe I do not understand what you mean by this statement.

Response: As mentioned earlier, the PM equation is closed if measurements of the two unobserved biophysical conductances ( $g_A$  and  $g_C$ ) are available. However,  $g_A$  and  $g_C$  cannot be directly measured at the canopy-scale and there is no universally agreed  $g_A$  and  $g_C$  model. By the term 'closure', we mean actual 'closure' of the PM equation by finding analytical solutions of  $g_A$  and  $g_C$ . This was done by solving 'n' equations and 'n' unknowns as described in equation 2 to 5 in the manuscript. The derivation of these equations is already explained in Mallick et al. (2014; 2015). We will briefly explain their derivation in the Appendix of the revised manuscript.

Line 225-227: "The estimates of EE in the current method consists of aggregated contribution from both interception and soil evaporation, and no further attempt is made to separate these two components". This is a considerable weakness in the approach seeing leaf area index and hence interception is so large for large parts of the Amazon and soil evaporation will be negligible. You are making this point yourself a few sentences later (line 232) Also: these two types of evaporation fluxes take place at very different source heights, so their GA will be very different, further weakening your approach.

**Response:** We do not agree. This is not a considerable weakness, but a fact which is clearly stated instead of withholding it. At the outset, the biophysical controls on evaporation and transpiration are mentioned, and no claim is made on understanding soil evaporation, interception evaporation etc.

We agree that different  $g_A$  exists for soil-canopy, sun-shade, and dry-wet conditions; which is currently integrated into a lumped  $g_A$  (given the big-leaf nature of STIC). From the big-leaf perspective, it is generally assumed that the aerodynamic conductance of water vapor and heat are equal (Raupach, 1998). However, for obtaining partitioned aerodynamic conductances, explicit partitioning of evapotranspiration is needed, which is beyond the scope of the current manuscript. We will mention this fact in the revised version of the manuscript.

Line 285: "The conductances showed a marked diurnal variation expressing their overall dependence on net radiation, vapor pressure deficit, and surface temperature". What conductance are you referring to here? gA or gC? Or both? Note that gA generally does not depend on net radiation or VPD etc., although it does in STICS.

Response: Here, we are referring to both  $g_A$  and  $g_C$  as clearly stated in Fig. 1 and the related descriptions as stated in line 279 to 288. The role of  $g_A$  is associated with the role of convection (Choudhury and Monteith, 1986) according to the surface energy balance principle as follows.

Neglecting horizontal advection and energy storage, the surface energy balance equation is written as follows:

$$\phi = \lambda E + H \tag{1}$$

Where  $\phi \cong R_N - G$ , with  $R_N$  being net radiation, and G being the conductive surface heat flux or ground heat flux, H is the sensible heat flux and  $\lambda E$  is the latent heat flux.

The sensible and latent heat flux can be expressed in the form of aerodynamic transfer equations (Boegh et al., 2002; Boegh and Soegaard, 2004) as follows:

$$H = \rho c_P g_A (T_o - T_A) \tag{2}$$

$$\lambda E = \frac{\rho c_P}{\gamma} g_A(e_0 - e_A) = \frac{\rho c_P}{\gamma} g_C(e_0^* - e_0)$$
(3)

Where  $T_A$  is the air temperature at the reference height ( $z_R$ ),  $e_A$  is the atmospheric vapor pressure (hPa) at the level at which  $T_A$  is measured,  $e_0$  and  $T_0$  are the atmospheric vapor pressure and air temperature at the source/sink height, or at the so-called roughness length ( $z_0$ ), where wind speed is zero. They represent the vapor pressure and temperature of the quasi-laminar boundary layer in the immediate vicinity of the surface level (Figure A1), and  $T_0$  can be obtained by extrapolating the logarithmic profile of  $T_A$  down to  $z_0$ .  $e_0^*$  is the saturation vapor pressure at  $T_0$  (hPa).

By combining eq. 1, 2, and 3 and solving for  $g_A$ , we get

$$g_A = \frac{\phi}{\rho c_P \left[ (T_o - T_A) + \left(\frac{e_0 - e_A}{\gamma}\right) \right]}$$
(4)

**Equation 4 clearly portrays the dependency of $g_A$ on net available energy and vapor pressure.**

Given R1's disposition on the wind speed dependent empirical  $g_A$  models based on the Monin-Obukhov Similarity Theory (MOST), it is important to mention that the Monin-Obukhov Length (L) is a function of evapotranspiration (E) (Brutsaert, 1982), and E is strongly dependent on the net available energy as well as vapor pressure deficit. The functions below describes the dependence of  $g_A$  on net available energy ( $\phi$ ) (= net radiation – ground heat flux) and vapor pressure deficit in addition to  $T_0$ - $T_A$ , despite  $g_A$  being generally estimated from wind speed information.

$$g_A = f\{L\} \tag{5}$$

$$L = \frac{u^* \rho C_P T_A}{g(H + 0.61 C_P T_A E)}$$
(6)

$$u^* = f\{L, E, specific humidity gradieant, wind speed\}$$
 (7)

$$E = f\{R_N, D_A, soil\ moisture, T_R\}$$
(8)

Here  $u^*$  is the friction velocity (m s-1),  $\rho$  is the air density (kg m-3),  $c_P$  is the specific heat of air (1004 j kg-1 K-1),  $T_A$  is the air temperature (K),  $D_A$  is the vapor pressure deficit (hPa). Rest all the variables are explained earlier.

According to equations 5 to 8, the dependence of  $g_A$  on net radiation and  $D_A$  is obvious. Wind is generated as a result of the differences in atmospheric pressure which is a result of uneven surface radiative heating. Therefore, the aerodynamic conductance (and wind as well) is an effect of net radiative heating and therefore, there should be a physical relationship between these two.

References:

- Anderson, M.C., & Kustas, W.P. (2008), Thermal remote sensing of drought and evapotranspiration. EOS, 89 (26), 233 240.
- Batra, N., Islam, S., Venturini, V., Bisht, G., and Jiang, L.: Estimation and comparison of evapotranspiration from MODIS and AVHRR sensors for clear sky days over the southern great plains, Remote Sens. Environ., 103, 1–15, 2006.
- Boegh, E., and Soegaard, H.: Remote sensing based estimation of evapotranspiration rates, Int. J. Remote Sens., 25(13), 2535–2551, 2004.
- Boegh, E., Soegaard, H., and Thomsen, A.: Evaluating evapotranspiration rates and surface conditions using Landsat TM to estimate atmospheric resistance and surface resistance, Remote Sens. Environ., 79, 329–343, 2002.
- Bonan, G. B., Williams, M., Fisher, R. A., and Oleson, K. W.: Modeling stomatal conductance in the earth system: linking leaf water-use efficiency and water transport along the soil-plant-atmosphere continuum, Geosci. Model Dev., 7, 2193-2222, doi:10.5194/gmd-7-2193-2014, 2014.
- Brutsaert, W.: Evaporation Into the Atmosphere, Reidel Pub. Comp., Dordrecht, Holland, 299 pp, 1982.
- Castellví, F., Cammalleri, C., Ciraolo, G., Maltese, A. and Rossi, F.: Daytime sensible heat flux estimation over heterogeneous surfaces using multitemporal land-surface temperature observations, Water Resour. Res., doi:10.1002/2015WR017587, 2016 (in press).
- Chen, F., Schwerdtfeger, P., 1989. Flux-gradient relationships for momentum and heat over a rough natural surface. Quarterly Journal of the Royal Meteorological Society 115, 335-352.
- Choudhury, B. J., and Monteith, J. L.: Implications of stomatal response to saturation deficit for the heat balance of vegetation, Agric. For. Meteorol., 36, 215 225, 1986.
- Ershadi, A., et al.: Impact of model structure and parameterization on Penman–Monteith type evaporation models. J Hydrology, 525, 521 535, 2015.
- Foken, T.: 50 Years of the Monin-Obukhov similarity theory, Boundary-Layer Meteorol., 2, 7–29, 2006.
- Gibson, L. A., Münch, Z., and Engelbrecht, J.: Particular uncertainties encountered in using a pre-packaged SEBS model to derive evapotranspiration in a heterogeneous study area in South Africa, Hydrol. Earth Syst. Sci., 15, 295-310, doi:10.5194/hess-15-295-2011, 2011.

- Gokmen, M., et al.: Integration of soil moisture in SEBS for improving evapotranspiration estimation under water stress conditions, Remote Sens. Environ., 121, 261–274, 2012.
- Harman, I.N., Finnigan, J.J.: Scalar concentration profiles in the canopy and roughness sublayer, Bound. Layer Meteorol. 129 (3), 323–351, 2008.
- Harman, I.N., Finnigan, J.J., 2007. A simple unified theory for flow in the canopy and roughness sublayer. Bound.-Layer Meteorol. 123 (2), 339–363.
- Holwerda, F., Bruijnzeel, L., Scatena, F., Vugts, H., Meesters, A.: Wet canopy evaporation from a Puerto Rican lower montane rain forest: the importance of realistically estimated aerodynamic conductance, J. Hydrol. 414, 1–15, 2012.
- Hong, J. and Kim, J.: Numerical study of surface energy partitioning on the Tibetan plateau: comparative analysis of two biosphere models, Biogeosciences, 7, 557-568, doi:10.5194/bg-7-557-2010, 2010.
- Jarvis, P.G., and McNaughton, K.G.: Stomatal control of transpiration: scaling up from leaf to region, Adv. Ecol. Res., 15, 1 49, 1986.
- Jiang, L., and Islam, S.: Estimation of surface evaporation map over Southern Great Plains using remote sensing data, Water Resour. Res., 37 (2), 329–340, 2001.
- Kleidon, A., Renner, M., and Porada, P.: Estimates of the climatological land surface energy and water balance derived from maximum convective power, Hydrol. Earth Syst. Sci., 18, 2201-2218, 2014.
- Mallick, K., Jarvis, A.J., Boegh, E., et al.: A surface temperature initiated closure (STIC) for surface energy balance fluxes, Remote Sens. Environ., 141, 243 261, 2014.
- Mallick, K., Boegh, E., Trebs, I., Alfieri, J.G., Kustas, W.P., Prueger, J.H., Das, N.N., Drewry, D., Hoffmann, L., and Jarvis, A.J.: Reintroducing radiometric surface temperature into the Penman-Monteith equation. Water Resources Research, 51, 6214 6243, doi:10.1002/2014WR016106, 2015.
- Matheny, A.M., Bohrer, G., Stoy, P., Baker, I.T., et al.: Characterizing the diurnal patterns of errors in the prediction of evapotranspiration by several land-surface models: An NACP analysis, J. Geophys. Res.- Biogeosci., 119, doi:10.1002/2014JG002623, 2014.
- Monteith, J.L.: Evaporation and surface temperature, Quart. J. Royal Met. Soc., 107, 1–27, 1981.
- Monteith J.L.: A reinterpretation of stomatal responses to humidity, Plant, Cell & Environment, 18, 357–364, 1995.
- Monteith, J.L., Unsworth, M.H., 2008. Principles of Environmental Sciences, Elsevier, Amsterdam.
- Morillas, L., García, M., Nieto, H., Villagarcia, L., Sandholt, I., Gonzalez-Dugo, M.P., Zarco-Tejada, P.J., Domingo, F.: Using radiometric surface temperature for energy flux estimation in Mediterranean drylands from a two-source perspective, Remote Sens. Environ., 136, 234 – 246, 2013.
- O'Grady, A.P., Eamus, D., and Hutley, L. B.: Transpiration increases during the dry season: patterns of tree water use in eucalypt open-forests of northern Australia, Tree Physiol., 19, 591—597, 1999.

- Prihodko, L., Denning, A.S., Hanan, N.P., Baker, I.T., and Davis, K.: Sensitivity, uncertainty and time dependence of parameters in a complex land surface model, Agric. For. Meteorol., 148 (2), 268–287, 2008.
- Raupach, M.R.:: Influence of local feedbacks on land-air exchanges of energy and carbon, Global Change Biol., 4, 477 494, 1998.
- Renner, M., Hassler, S. K., Blume, T., Weiler, M., Hildebrandt, A., Guderle, M., Schymanski, S. J., and Kleidon, A.: Dominant controls of transpiration along a hillslope transect inferred from ecohydrological measurements and thermodynamic limits, Hydrol. Earth Syst. Sci. Discuss., doi:10.5194/hess-2015-535, 2016.
- Schulz, K., Beven, K.J.: Data-supported robust parameterisations in land surfaceatmosphere flux predictions: towards a top–down approach, Hydrol. Process. 17, 2259– 2277, 2003.
- Simpson, I.J., Thurtell, G.W., Nuemann, H.H., den Hartog, G., Edwards, G.C.: The validity of similarity theory in the roughness sublayer above forests, Boundary- Layer Meteorology 87, 69-99, 1998.
- Thom, A.S., Stewart, J.B., Oliver, H.R., Gash, J.H.C.: Comparison of aerodynamic and energy budget estimates of fluxes over a pine forest, Quart. J. Royal Met. Soc., 101, 93-105, 1975.
- Timmermans, J., Su, Z., van der Tol, C., Verhoef, A., and Verhoef, W.: Quantifying the uncertainty in estimates of surface–atmosphere fluxes through joint evaluation of the SEBS and SCOPE models, Hydrol. Earth Syst. Sci., 17, 1561-1573, doi:10.5194/hess-17-1561-2013, 2013.
- Villani, M.G., Schmid, H.P,Su, H.B., Hutton, J.L., and Vogel, C.S.: Turbulence statistics measurements in a northern hardwood forest, Boundary-Layer Meteorology 108: 343–364, 2003.
- Van Dijk, A.I.J.M., et al.: Rainfall interception and the couple surface water and energy balance, Agric. For. Meteorol., 214 215, 402 415, 2015.
- Zhang, Q., Manzoni, S., Katul, G., Porporato, A., and Yang, D.: The hysteretic evapotranspiration-vapor pressure deficit relation, J. Geophys. Res. Biogeosci., 119, 125–140, doi:10.1002/2013JG002484, 2014.

---

## Author Comment (AC2)

**Reviewer 2 (R2):**

This manuscript describes a study that infers stomatal and aerodynamic conductances from eddy flux observations. I think in general, this study is innovative and presents novel material, so that in principle it should be published. I hesitate recommendation for publication mostly because I am not entirely convinced by the approach and I feel that this needs revision. Hence, I recommend major revisions, although I do not think that it necessarily involves a lot of work to address the points below.

Response: We thank R2 for the encouraging comments and for appreciating the novelty of the aerodynamic and canopy conductance ( $g_A$  and  $g_C$ ) retrieval to assess their controls on evaporation and transpiration. We appreciate the valuable suggestions which will further improve the manuscript.

**Major points:**

(1) My major problem with the manuscript is that I do not understand the approach, so that it is difficult to assess its plausibility. While the main equations are provided in the manuscript (eqn. 2-5), there is no more description on where these equations come from, except for references to prior papers by the authors. I think it is necessary to at least provide a description at a qualitative level where these equations come from.

Response: We agree to include the derivations of eqns. 2-5 in the Appendix of the revised manuscript. A detailed derivation of the STIC1.2 state equations (eqns. 2-5) is given below.

Formulation of STIC was based on the goal to find an analytical solution of the two unobserved 'state variables' (i.e., aerodynamic and canopy conductances) ( $g_A$  and  $g_C$ ) in the Penman-Monteith (PM) equation while exploiting the radiative (net radiation and ground heat flux), meteorological (air temperature, humidity), and radiometric surface temperature ( $T_R$ ) as external inputs. The fundamental assumption in STIC is the first order dependence of  $g_A$  and  $g_C$  on the aerodynamic temperature ( $T_0$ ) and soil moisture (through the radiometric surface temperature,  $T_R$ ). These assumptions allow direct integration of  $T_R$  into the PM equation and simultaneously constrain the conductances. Given  $T_R$  is the direct signature of the soil moisture availability, inclusion of  $T_R$  in the PM equation also works to add water stress controls in  $g_C$ .

Neglecting horizontal advection and energy storage, the surface energy balance equation is written as follows:

$$\phi = \lambda E + H \tag{1}$$

Where  $\phi \cong R_N - G$ , with  $R_N$  being net radiation, and G being the conductive surface heat flux or ground heat flux, H is the sensible heat flux and  $\lambda E$  is the latent heat flux (or evapotranspiration, E).

According to Figure A1 in the manuscript, while the sensible heat flux is controlled by a single aerodynamic resistance  $(r_A)$  (or  $1/g_A$ ); the water vapor flux is controlled by two resistances in series, the surface resistance  $(r_C)$  (or  $1/g_C$ ) and the aerodynamic resistance to vapor transfer  $(r_C + r_A)$ . For simplicity, it is implicitly assumed that the aerodynamic resistance of water vapor and heat are equal (Raupach, 1998), and both the fluxes are transported from the same level from near surface to the atmosphere. The sensible and

latent heat flux can be expressed in the form of aerodynamic transfer equations (Boegh et al., 2002; Boegh and Soegaard, 2004) as follows:

$$H = \rho c_P g_A (T_o - T_A) \tag{2}$$

$$\lambda E = \frac{\rho c_P}{\gamma} g_A(e_0 - e_A) = \frac{\rho c_P}{\gamma} g_C(e_0^* - e_0)$$
(3)

Where  $\rho$  is the density of dry air (kg m-3),  $c_P$  is the specific heat of dry air (MJ kg-1 K-1),  $\gamma$  is the psychrometric constant (hPa K-1),  $T_A$  is the air temperature at the reference height ( $z_R$ ),  $e_A$  is the atmospheric vapor pressure (hPa) at the level at which  $T_A$  is measured,  $e_0$  and  $T_0$  are the atmospheric vapor pressure and air temperature at the source/sink height (i.e., aerodynamic temperature), or at the so-called roughness length ( $z_0$ ), where wind speed is zero. They represent the vapor pressure and temperature of the quasi-laminar boundary layer in the immediate vicinity of the surface level (Figure A1), and  $T_0$  can be obtained by extrapolating the logarithmic profile of  $T_A$  down to  $z_0$ .  $e_0^*$  is the saturation vapor pressure at  $T_0$  (hPa).

By combining eq. 1, 2 and 3 and solving for  $g_A$ , we get the following equation.

$$g_A = \frac{\phi}{\rho c_P \left[ (T_o - T_A) + \left(\frac{e_0 - e_A}{\gamma}\right) \right]}$$
(4)

Combining the aerodynamic expressions of  $\lambda E$  in eq. 3 and solving for  $g_C$ , we can express  $g_C$  in terms of  $g_A$ ,  $e_0^{*}$ ,  $e_0$ , and  $e_A$ .

$$g_{C} = g_{A} \frac{(e_{0} - e_{A})}{(e_{0}^{*} - e_{0})}$$
(5)

While deriving the expressions for  $g_A$  and  $g_C$ , two more unknown variables are introduced ( $e_0$  and  $T_0$ ), thus there are two equations and four unknowns. Therefore, two more equations are needed to close the system of equations.

An expression for  $T_0$  is derived from the Bowen ratio ( $\beta$ ) (Bowen, 1926) and evaporative fraction ( $\Lambda$ ) (Shuttleworth et al., 1989) equation.

$$\beta = \left(\frac{1-\Lambda}{\Lambda}\right) = \frac{\gamma(T_0 - T_A)}{(e_0 - e_A)}$$
(6)

$$T_o = T_A + \left(\frac{e_0 - e_A}{\gamma}\right) \left(\frac{1 - \Lambda}{\Lambda}\right) \tag{7}$$

This expression for  $T_0$  introduces another new variable ( $\Lambda$ ); therefore, one more equation that describes the dependence of  $\Lambda$  on the conductances ( $g_A$  and  $g_c$ ) is needed to close the system of equations. The detailed derivation of an expression for  $\Lambda$  is described in Mallick et al. (2014, 2015) and this is briefly described below. Estimation of  $e_0$  is based on numerical iteration as described in the Appendix of the manuscript and is also described in the response (3) below.

In order to express  $\Lambda$  in terms of  $g_A$  and  $g_C$ , we had adopted the advection – aridity hypothesis (Brutsaert and Stricker, 1979) with a modification introduced by (Mallick et al.,

2015). Although the advection–aridity hypothesis leads to an assumed link between  $g_A$  and  $T_0$ , the effects of surface moisture (or water stress) was not explicit in the advection–aridity equation. We implemented a moisture constraint in the original advection-aridity hypothesis for deriving an expression of  $\Lambda$ . The logic of using the advection-aridity hypothesis for finding an expression of  $\Lambda$  is described in Mallick et al. (2014). A modified form of the original advection-aridity hypothesis is written as follows.

$$E_{PM}^{*} = 2E_{PT}^{*} - E$$
 (8)

Here  $E_{PM}^*$  is the potential evapotranspiration according to Penman-Monteith (Monteith, 1965) for any surface, and  $E_{PT}^*$  is the potential evapotranspiration according to Priestley-Taylor (Priestley and Taylor, 1972). Dividing both sides by E we get,

$$\frac{E}{E_{PM}^{*}} = \frac{E}{2E_{PT}^{*} - E}$$
(9)

and dividing the numerator and denominator of the right hand side of eqn. 9 by E\*T we get,

$$\frac{E}{E_{PM}^{*}} = \frac{\frac{E}{E_{PT}^{*}}}{2 - \frac{E}{E_{PT}^{*}}}$$
(10)

Again assuming the Priestley-Taylor equation for any surface is a variant of the PM potential evapotranspiration equation, we will derive an expression of  $E_{PT}^*$  for any surface.

$$E_{PM}^{*} = \frac{s\phi + \rho c_{P} g_{A} D_{A}}{s + \gamma \left(1 + \frac{g_{A}}{g_{cmax}}\right)}$$
(11)
$$= \frac{s\phi}{s + \gamma \left(1 + \frac{g_{A}}{g_{cmax}}\right)} \left(1 + \frac{\rho c_{P} g_{A} D_{A}}{s\phi}\right)$$
$$= \frac{as\phi}{s + \gamma \left(1 + \frac{g_{A}}{g_{cmax}}\right)}$$
(12)
$$= E_{PT}^{*}$$

Here  $\gamma$  is the psychrometric constant (hPa K-1), s is the slope of the saturation vapor pressure versus air temperature (hPa K-1),  $\alpha$  is the Priestley-Taylor parameter ( $\alpha$  =1.26 under non-limiting moisture conditions), DA is the vapor pressure deficit of air (hPa). gCmax is defined as the maximum possible gC under the prevailing atmospheric conditions whereas gC is limited due to the moisture availability (M) and hence gCmax = gC/M (Monteith, 1995; Raupach, 1998). We assume that M is a significant controlling factor for the ratio of actual and potential evapotranspiration (or transpiration for a dry canopy), and the interactions between the land and environmental factors are substantially reflected in M. Since, Penman (1948) derived his equation over the open water surface and gCmax over the water surface is very high (Monteith, 1965; 1981), gA/gCmax was assumed to be negligible.

Expressing  $\phi$  as  $\phi$  = E/A and expressing  $E_{PT}^*$  according to eqn. 12 gives the following expression of E/ $E_{PT}^*$ .

$$\frac{E}{E_{PT}^*} = \frac{\Lambda \left[ s + \gamma \left( 1 + \frac{g_A}{g_{Cmax}} \right) \right]}{\alpha s}$$
(13)

Now substituting  $E/E_{PT}^*$  from eqn. 13 into eq. 10 and after some algebra we obtain the following expression.

$$\frac{E}{E_{PM}^{*}} = \frac{\Lambda \left[ s + \gamma \left( 1 + \frac{g_{A}}{g_{cmax}} \right) \right]}{2\alpha s - \Lambda \left[ s + \gamma \left( 1 + \frac{g_{A}}{g_{cmax}} \right) \right]}$$
(14)

According to the PM equation (Monteith, 1965) of actual and potential evapotranspiration,

$$\frac{E}{E_{PM}^{*}} = \frac{\frac{s\phi + \rho c_{p}g_{A}D_{A}}{s + \gamma \left(1 + \frac{g_{A}}{g_{c}}\right)}}{\frac{s\phi + \rho c_{p}g_{A}D_{A}}{s + \gamma \left(1 + \frac{g_{A}}{g_{cmax}}\right)}}$$
(15)

Combining eqn. 14 and 15 (eliminating  $E/E_{PM}^*$ ) gives an expression for  $\Lambda$  in terms of the conductances.

$$\frac{s+\gamma\left(1+\frac{Mg_A}{g_C}\right)}{s+\gamma\left(1+\frac{g_A}{g_C}\right)} = \frac{\Lambda\left[s+\gamma\left(1+\frac{Mg_A}{g_C}\right)\right]}{2\alpha s-\Lambda\left[s+\gamma\left(1+\frac{Mg_A}{g_C}\right)\right]}$$
(16)

After some algebra the final expression of  $\Lambda$  is as follows.

$$\Lambda = \frac{2\alpha s}{2s + 2\gamma + \gamma \frac{g_A}{g_c}(1+M)}$$
(17)

Given the information of  $R_N$ , G,  $T_A$ , and  $R_H$  or  $e_A$ , the four state equations (eqns. 4, 5, 7, and 17) can be solved simultaneously to derive analytical solutions for the four state variables. The analytical solutions to the state equations 4, 5, 7, and 17 still have four additional unknowns; M,  $e_0$ ,  $e_0^*$ , and  $\alpha$ , and these variables are iteratively estimated as described in the Appendix of the current manuscript. For estimating M, we have extensively used the radiometric surface temperature (TR) in a physical retrieval framework in STIC1.2, thus treating TR as an external input.

(2) The point where I really got confused is that eqn. 5 uses the Priestley-Taylor coefficient, which is an empirical coefficient in an evaporation equation that is rather different from the Penman Monteith equation. Where does this coefficient suddenly come from? I find this quite confusing, and it needs at least a minimum of explanation as it is not obvious.

Response: Good point indeed and we apologise for the confusion.

From the derivation of the equation 17 above (eqn. 5 in the manuscript), it is apparent that the Priestley-Taylor coefficient ( $\alpha$ ) appeared due to the use of the Advection-Aridity hypothesis for deriving the state equation of the evaporative fraction. However, instead of assuming  $\alpha$  as a 'fixed parameter', we have developed a physical equation of  $\alpha$  (eqn. A8 in the manuscript) and numerically estimated  $\alpha$  as a 'variable'. **The derivation of the equation** for  $\alpha$  is described is the following response (also in the Appendix of the manuscript in line 609 to 611).

We will make this description more explicit in the revised version of the manuscript.

**(3) What I also do not understand is why an iterative scheme is needed.**

Response: The analytical solution to the above state equations 4, 5, 7, and 17 (eqn. 2 – 5 in the manuscript) have four accompanying unknowns; M (surface moisture availability),  $e_0$  (vapor pressure at the source/sink height),  $e_0^*$  (saturation vapor pressure at the source/sink height), and  $\alpha$ , and as a result there are 4 equations with 8 unknowns. Consequently an iterative solution is needed to determine the four unknown variables (as described below).

An estimate of  $e_0^*$  is obtained by inverting the aerodynamic transfer equation of  $\lambda E$ .

$$e_0^* = e_A + \left[\frac{\gamma \lambda E(g_A + g_C)}{\rho c_P g_A g_C}\right]$$
(18)

Following Shuttleworth and Wallace (1985) (SW85, hereafter), the vapor pressure deficit  $(D_0)$  (= $e_0^* - e_0$ ) and vapor pressure ( $e_0$ ) at the source/sink height are expressed as follows.

$$D_0 = D_A + \left[\frac{\{s\phi - (s+\gamma)\lambda E\}}{\rho c_P g_A}\right]$$
(19)

$$e_0 = e_0^* - D_0 \tag{20}$$

A physical equation of  $\alpha$  is derived by expressing the evaporative fraction ( $\Lambda$ ) as function of the aerodynamic equations of H [ $\rho c_P g_A (T_0 - T_A)$ ] and  $\lambda E \left[\frac{\rho c_P}{\gamma} \frac{g_A g_C}{g_A + g_C} (e_0^* - e_A)\right]$  as follows.

$$\Lambda = \frac{\lambda E}{H + \lambda E} \tag{21}$$

$$= \frac{\frac{\rho c_P}{\gamma} \frac{g_A g_C}{g_A + g_C} (e_0^* - e_A)}{\rho c_P g_A (T_0 - T_A) + \frac{\rho c_P}{\gamma} \frac{g_A g_C}{g_A + g_C} (e_0^* - e_A)}$$
(22)

$$=\frac{g_{C}(e_{0}^{*}-e_{A})}{[\gamma(T_{0}-T_{A})(g_{A}+g_{C})+g_{C}(e_{0}^{*}-e_{A})]}$$
(23)

Combining eqn. 23 and eqn. 17 (eliminating  $\Lambda$ ), we can derive a physical expression of  $\alpha$ .

$$\alpha = \frac{g_C(e_0^* - e_A) \left[ 2s + 2\gamma + \gamma \frac{g_A}{g_C} (1+M) \right]}{2s[\gamma(T_0 - T_A)(g_A + g_C) + g_C(e_0^* - e_A)]}$$
(24)

Following Venturini et al. (2008), M can be expressed as the ratio of the vapor pressure difference to the vapor press deficit between surface to atmosphere as follows.

$$M = \frac{(e_0 - e_A)}{(e_0^* - e_A)} = \frac{s_1(T_{SD} - T_D)}{s_2(T_0 - T_D)}$$
(25)

Where  $T_{SD}$  is the dewpoint temperature of the evaporating front (at source/sink height) and  $T_D$  is the air dewpoint temperature,  $s_1$  and  $s_2$  are the psychrometric slopes of the saturation vapor pressure and temperature between  $(T_{SD} - T_D)$  versus  $(e_0 - e_A)$  and  $(T_0 - T_D)$  versus  $(e_0^* - e_A)$  relationship (Venturini et al., 2008). Since  $T_0$  is not available and  $T_R$  and  $e_A$  are available, we compute  $s_2$  as  $s_2 = (e_S^* - e_A)/(T_R - T_D)$  with the assumption that errors due to any inequality between  $T_0$  versus  $T_R$  and  $e_0^*$  versus  $e_S^*$  tend to be cancelled out in this ratio. This appears to be a valid assumption due to the close relationship between  $T_0$  and  $T_R$  (Huband and Monteith, 1986). Despite  $T_0$  drives the sensible heat flux, the comprehensive dry-wet signature of underlying surface due to soil moisture variations is directly reflected in  $T_R$  (Kustas and Anderson, 2009). Therefore, using  $T_R$  in the denominator of eqn. 25 gives a direct signature of the surface moisture availability (M). In eqn. 25,  $T_{SD}$  computation is challenging because both  $e_0$  and  $s_1$  are unknown. By decomposing the aerodynamic equation of  $\lambda E$ ,  $T_{SD}$  can be expressed as follows.

$$\lambda E = \frac{\rho c_P}{\gamma} g_A(e_0 - e_A) = \frac{\rho c_P}{\gamma} g_A s_1 (T_{SD} - T_D)$$

$$T_{SD} = T_D + \frac{\gamma \lambda E}{\rho c_P g_A s_1}$$
(26)

In the earlier STIC versions,  $s_1$  was approximated at  $T_D$ ,  $T_{SD}$  was estimated from  $s_1$ ,  $T_D$ ,  $T_R$ , and related saturation vapor pressures (Mallick et al., 2014; 2015), and M was estimated from eqn. 25 (estimation of  $T_{SD}$  and M was stand-alone earlier). However, since  $T_{SD}$  depends on  $\lambda E$  and  $g_A$ , an iterative procedure is applied in STIC1.2 to estimate  $T_{SD}$  and M as described below, which is another modification of the STIC1.0 and STIC1.1.

In STIC1.2, an initial value of  $\alpha$  is assigned as 1.26 and initial estimates of  $e_0^*$  and  $e_0$  are obtained from  $T_R$  and M as  $e_0^* = 6.13753e^{(T_R+237.3)}$  and  $e_0 = e_A + M(e_0^* - e_A)$ . Initial  $T_{SD}$  and M were estimated as described above. With the initial estimates of these variables; first estimate of the conductances,  $T_0$ ,  $\Lambda$ , and  $\lambda E$  are derived. The process is then iterated by updating  $D_0$  (using eqn. 19),  $e_0^*$  (using eqn. 18),  $e_0$  (using eqn. 20),  $T_{SD}$  (using eqn. 26 with  $s_1$  estimated at  $T_D$ ), M [M =  $s_1(T_{SD} - T_D)/s_2(T_R - T_D)$ ], and  $\alpha$  (using eqn. 24), with the first estimates of  $g_C$ ,  $g_A$ , and  $\lambda E$ , and recomputing  $g_A$ ,  $g_C$ ,  $T_0$ ,  $\Lambda$ , and  $\lambda E$  in the subsequent iterations with the previous estimates of  $e_0^*$ ,  $e_0$ ,  $T_{SD}$ , M, and  $\alpha$  until the convergence  $\lambda E$  is achieved. Stable values of  $\lambda E$ ,  $e_0^*$ ,  $e_0$ ,  $T_{SD}$ , M, and  $\alpha$  are obtained within ~25 iterations.

The above equations are previously included in the appendix of the current manuscript.

(4) Can't one simply use the observations and use a simple partitioning based on the Bowen ratio?

Response: Here we intended to partition evapotranspiration into component water fluxes. Although the Bowen ratio (Bowen, 1926) is an energy partitioning ratio to understand the relative apportioning between sensible and latent heat flux, it is not relevant for the latent heat flux partitioning into transpiration and evaporation. In this context an aggregated surface moisture availability (or water stress factor) is a better metric for dry-wet latent heat flux partitioning and we used the retrieved surface moisture availability (M) for partitioning of the latent heat flux.

(5) It would be good to describe what the differences and similarities are to previous approaches. As the authors propose a new approach, they should provide a better description that is easier to follow of what is being done.

Response: We assume R2 is intending to the differences of STIC with other approaches that earlier attempted to understand the biophysical controls of evapotranspiration, which is briefly described in the table below.

| Biophysical states | Modeling principles                                                                                                                                                                                                                                                                                                                                                                                                                                                                                                                                                                                                           |                                                                                                                                                                                                                                 |  |
|--------------------|-------------------------------------------------------------------------------------------------------------------------------------------------------------------------------------------------------------------------------------------------------------------------------------------------------------------------------------------------------------------------------------------------------------------------------------------------------------------------------------------------------------------------------------------------------------------------------------------------------------------------------|---------------------------------------------------------------------------------------------------------------------------------------------------------------------------------------------------------------------------------|--|
|                    | Parametric
( Ma et al., 2015; Kumagai et al., 2004 )                                                                                                                                                                                                                                                                                                                                                                                                                                                                                                                                                                | Nonparametric (STIC)                                                                                                                                                                                                            |  |
| g₄                 | Either $g_A$ is assumed to be the momentum
conductance $(g_M)$ or estimated as a sum of
$g_M$ and quasilaminar boundary-layer
conductance $(g_B)$ .
$1/g_A = 1/g_M + 1/g_B$
$g_M = f\{u, wind speed\}$
$g_B = f\{Nusselt number, leaf dimension, thermalconductivity of air in boundary layer, wind speed,kinematic viscosity, Reynolds number}If u* is available from EC tower, it is directlyused, otherwise u* is estimated using Monin-Obukhov Similarity Theory (MOST). MOSTis only valid for an extended, uniform, andflat surface (Foken, 2006)$                                                     | Analytically retrieved by solving 'n' state
equations and 'n' unknowns, with explicit
convective feedback.                                                                                                                |  |
| gс                 |  <li>(a) If λE measurements are available from the EC towers, gC is estimated by inverting the PM equation. This leads to circularity. Since λE observations are used to obtain gC, the same gC should not be used to assess the biophysical controls of λE.</li> <li>(b) If λE measurements are not available from the EC towers (i.e., at grid-scale or spatial scale), gC is modelled either by coupling with leaf-scale photosynthesis models (Ball et al., 1987; Leuning, 1995) or gC is estimates from standalone empirical models (Jarvis, 1976)</li>  | Analytically retrieved by solving 'n' state
equations and 'n' unknowns where
physical feedbacks of g A , soil moisture,
and vapor pressure deficit are embedded
(as explained in the STIC1.2 equations). |  |

If R2 is intending the differences between STIC1.2 with other previous versions, we propose to include a table in the appendix to describe the fundamental differences between STIC1.0, STIC1.1, and STIC1.2. The Table is given below.

[revised manuscript text omitted]

**Minor points:**

- The authors refer to  $\lambda E$  as evaporation, which, technically speaking, is the latent heat flux, not evaporation.

Response: Necessary corrections will be incorporated in the revised manuscript.

- Abstract: dry and wet conditions  $\lambda E_T$ , do you mean conditions in which water is not limiting vs. limiting, or precipitation vs. radiation driven conditions?

Response: It is the precipitation vs. radiation driven conditions. We will clarify this in the abstract.

- Biophysical control of  $\lambda E_T$  should be briefly explained by what this means.

Response: Aerodynamic (physical) and stomatal (biological) conductances ( $g_A$  and  $g_C$ ) together impose substantial biophysical controls on  $\lambda E_T$ . At large  $g_A$  and small  $g_C$ , the vapor pressure deficit close to the canopy source/sink height ( $D_0$ ) changes in response to the transpiration rate caused due to changes in the atmospheric vapor pressure deficit ( $D_A$ ) or  $g_A$ . This results in strong canopy-atmosphere coupling and such condition is prevalent under soil moisture deficient conditions. On the other hand large  $g_C$  minimizes the gradients of vapor pressure deficit just above the canopy, such that  $D_0$  tend towards zero and remains independent of any change in transpiration rate caused by changes in  $D_A$  or  $g_A$ . This substantially weakens the canopy-atmosphere coupling and such situation prevails under predominantly wet conditions.

We shall include this description in the introduction of the revised manuscript.

- Line 145: I wonder why approaches that directly link stomatal conductance to photosynthesis are not mentioned, such as Ball-Berry?

Response: We shall include references to photosynthesis-dependent stomatal conductance models in the revised manuscript.

- Line 194: Where do these "state equations" come from? Referring to previously published work is fine for derivations, but the description should still mention what the concepts are that are behind these equations.

Response: As discussed earlier we agree to include a description of the derivation in the revised manuscript.

- A table of variables would help.

Response: A table of variables will be included.

- Line 238: I think the authors assume that the conductances to momentum, sensible and latent heat are identical. If this is the case, it should be mentioned, as there are also approaches to surface exchange that do not treat them as being identical.

Response: Yes, the conductances of momentum for the sensible and latent heat flux are assumed identical. We will mention this in the revised manuscript after equation 7.

- Line 331: As the typical readers of HESS are not micrometeorologists, it would be useful to explain the decoupling coefficient in some more detail. This will help to interpret the following results.

Response: The decoupling coefficient or factor Omega ( $\Omega$ ) is a dimensionless coefficient ranging from 0.0 to 1.0 (Jarvis and McNaughton, 1986) and considered as an index of the degree of stomatal control on transpiration. The equation of  $\Omega$  is as follows.

$$\Omega = \frac{\frac{s}{\gamma} + 1}{\frac{s}{\gamma} + 1 + \frac{g_A}{g_C}}$$

The  $\Omega$  form of the Penman-Monteith (PM) equation for evapotranspiration is as follows.

$$\lambda E = \Omega \lambda E_{eq} + (1 - \Omega) \lambda E_{imp}$$
$$\lambda E_{eq} = \frac{s\phi}{s + \gamma}$$
$$\lambda E_{imp} = \frac{\rho c_P}{\gamma} g_C D_A$$

Where,  $\lambda E_{eq}$  is the equilibrium evapotranspiration, which depends only on the net available energy and would be obtained over an extensive surface of uniform moisture availability (Jarvis and McNaughton, 1986; Kumagai et al., 2004).  $\lambda E_{imp}$  is the imposed evapotranspiration, which is 'imposed' by the atmosphere on the vegetation surface through the effects of vapor pressure deficit (triggered under limited soil moisture availability) and evapotranspiration is proportional to gc.

When the  $g_C/g_A$  ratio is very small (i.e., water stressed conditions), stomata principally control the water loss and a change in  $g_C$  will result in a nearly proportional change in transpiration. In this case the  $\Omega$  value approaches zero, and vegetation is believed to be fully coupled to the atmosphere. In contrast, for a high  $g_C/g_A$  ratio (i.e., water unstressed conditions), changes in  $g_C$  will have little effect on the transpiration rate, and transpiration is predominantly controlled by the net available radiative energy. In this case the  $\Omega$  value approaches unity, and vegetation is considered to be poorly coupled to the atmosphere. We will add this description in the revised version of the manuscript.

- Line 422: To what extent could these discrepancies between how conductances are derived also relate to actual differences in the conductances for momentum vs. heat?

Response: This is indeed a good point addressed by R2 (although beyond the scope of this manuscript) and will be clarified in the revised version. However, a detailed investigation using data on atmospheric profiles of wind speed, temperature etc. are needed to actually quantify such differences.

Momentum transfer is associated with pressure forces and not identical to heat and mass transfers (Massman, 1999). In principle, the aerodynamic conductances for heat and mass transfers are assumed equal (Monteith, 1965, 1981). In STIC1.2,  $g_A$  is directly estimated (as described previously) and is a robust representative of the resistance to heat/water vapor transfer. The parametric  $g_A$  estimates based on the friction velocity and wind speed is more representative for momentum transfer. Therefore, the difference between the two different  $g_A$  estimates (Fig. 2) is primarily due to the actual difference in the conductances for momentum and heat/water vapor.

Response: We will correct this.

- Line 498: The authors should stick to the same ratio gA/gC for easier interpretation.

**Reference:**

[revised manuscript text omitted]

---

## Author Response (AR1)

**Response to Review of HESSD-552-2015 by Anonymous Reviewer #1**

*Note: Original reviewer comments are in black and author's responses are in blue throughout. The*
*changes in manuscript are in track change. The line numbers mentioned are according to the*
*revised version of the manuscript*

Major comments:

(1) My main concern is that the manuscript does not present any new theory (and on top of
that uses an approach (STICS) that in my opinion is misguided, despite the fact that it has
been published).

> **Summary response:** We thank R1 for the comments and firmly object the statement on 'misguidance' which appears to be potentially misleading particularly when a large part of the community have outright enthusiasm in developing analytical approaches for estimating terrestrial evapotranspiration (E) (or latent heat flux, $\lambda$E) and sensible heat fluxes (H) to overcome the ambiguities associated with parameterizations of aerodynamic ($g_A$) and canopy surface conductances ($g_C$) (**Kleidon et al., 2014; Matheny et al., 2014; Ershadi et al., 2015**). Besides, we would like to clarify that the abbreviation for our model framework is "STIC" and not "STICS" as referred to by R1.
>
> **R1's claim is flawed because STIC (Surface Temperature Initiated Closure) introduced a novel analytical method to integrate radiometric surface temperature ($T_R$) into the Penman-Monteith model to overcome the limitations associated with empirical (uncertain) leaf-scale parameterizations of the aerodynamic and canopy surface conductances ($g_A$ and $g_C$) which are not directly measurable either at the canopy-scale or at the large spatial grid-scale. To our knowledge, this research objective is unquestionably novel and the behavior of the analytically retrieved canopy-scale conductances as well as transpiration are compliant with the theory earlier postulated in the literatures (Jarvis and McNaughton, 1986; Monteith, 1995, Raupach, 1998). In addition to its simplicity, STIC has the capabilities for generating spatially explicit surface energy fluxes and independent of submodels for boundary layer developments.**

The characteristic features of STIC are explicitly stated in section 2.1 (Theory) and 2.2 (State
equations).

**Detailed response:** The most tangible accomplishment and uniqueness of STIC (STIC1.2)
is the physical integration of land surface temperature (i.e., radiometric surface temperature,
$T_R$) into a combined framework of the Penman-Monteith (PM) and Shuttleworth-Wallace
(SW) model for simultaneously estimating E, H, $g_A$, $g_C$, surface moisture status, and E
components (evaporation, $E_E$ and transpiration, $E_T$). The intrinsic link between the PM-SW
model and $T_R$ emanates through the first-order dependence of the biophysical conductances
($g_A$ and $g_C$) on the aerodynamic temperature ($T_0$) (through $T_R$) and soil moisture (through $T_R$).
However, until now the explicit use of $T_R$ in the PM-SW model was hindered due to the
unavailability of any direct method to integrate $T_R$ into these models, and, furthermore, due
to the lack of physical models expressing biophysical states of vegetation as a function of
$T_R$. Therefore, the majority of the E modeling approaches strongly rely on surface
reflectance and meteorology; and thermal approaches require significant parameterization of
land surface properties (e.g., $g_A$ and $g_C$) which are very empirical in nature (Schulz and
Beven, 2003; Prihodko et al., 2008; Bonan et al., 2014; Ershadi et al., 2015).

To bridge this gap, the STIC methodology was developed as a novel thermal-based
biophysical scheme for directly estimating E over terrestrial ecosystems by leveraging the
combined strength of $T_R$ observations and physically-based models (Mallick et al., 2014;
2015). In addition to physically integrating $T_R$ observations into a combined PM-SW
framework, STIC1.2 also establishes of a feedback loop describing the relationship between
$T_R$ and E, coupled with canopy-atmosphere components relating E to aerodynamic
temperature ($T_0$) and vapor pressure ($e_0$) (in STIC1.2). By integrating $T_R$ with standard
surface energy balance (SEB) theory and vegetation biophysical principles, STIC formulates
multiple state equations in order to eliminate the need of exogenous parametric submodels
for the surface and aerodynamic conductances, aerodynamic temperatures, and land-
atmosphere coupling. Instead these 'internal states' are numerically retrieved. Originally
designed for application to thermal remote sensing data from Earth observation sensors, the
STIC framework exploits observations of **$T_R$**, radiative, and meteorological variables
including net radiation ($R_N$), ground heat flux (G), air temperature ($T_A$), relative humidity ($R_H$)
or vapor pressure ($e_A$) at a reference level above the surface, and can be applied over any
ecosystem, provided the necessary input variables are available.

**(2)** STICs is misguided because it ends up with an aerodynamic conductance that does not
depend on wind speed.

Summary response: R1 claims that STIC is misguided due to two reasons. According to R1, the first reason should be that aerodynamic conductance does not depend on wind speed (u). **It should be noted that, in one of the hallmark papers by Choudhury and Monteith (1986), it is clearly stated that 'aerodynamic conductance determined by wind speed and roughness is assumed to be unaffected by buoyancy'. Strictly, the aerodynamic conductance should be replaced by a term which accounts for radiative as well as convective heat transfer'.** Although incorporation of u data has almost become a dogma (Foken, 2006) in the field of land surface energy balance modelling, there are several widely accepted evapotranspiration estimation approaches that do not incorporate $W_s$, for example, maximum entropy production approach (Kleidon et al., 2014), evaporative fraction approach (Jiang and Islam, 2001; Batra et al., 2006), complementary relationship approach (Venturini et al., 2008) etc.

A table is included in the Appendix (Table A2) which describes the fundamental differences
in $g_A$ modeling between the conventional approaches and STIC.

**Detailed response: Given the importance of $g_A$ for evapotranspiration (E) estimates**
**there are overriding cases for getting this 'right' in the surface energy balance models**
**(Prihodko et al., 2008; Hong et al., 2010; Gibson et al., 2011; Holwerda et al., 2012;**
**Gokmen et al., 2012; Morillas et al., 2013).** However, if the empirical $g_A$ models currently
provide accurate estimates of E for the wrong reasons then this status quo has to be
questioned, especially as errors like this might become important when predicting E under
future boundary conditions. **Furthermore, it is not obvious that $W_S$-based models**
**currently provide accurate estimates of $g_A$, in particular at the grid-scale (e.g., 1 km**
**and above) where bundles of site specific parameters are required (which cannot be**
**measured).**

We would like to bring forward the following arguments concerning $W_S$-based $g_A$ estimation.

**(a)** As highlighted in several studies (Monteith and Unsworth, 2008; Holwerda et al., 2012), the momentum transfer equation for $g_A$ estimation based on the Monin-Obukhov Similarity Theory (MOST) only holds for an extended, uniform, and flat surface (Foken, 2006). MOST tends to fail over rough surfaces due to breakdown of the similarity relationships for heat and water vapour transfer in the roughness sub-layer, which results in an underestimation of the 'true' $g_A$ by a factor 1-3 (Thom et al., 1975; Chen and Schwerdtfeger, 1988; Simpson et al., 1998; Holwerda et al., 2012). Despite some of the boundary layer studies based on parameterized friction velocity (u*) demonstrated the validity of MOST (subjected to tuning and calibration) (Harman and Finnigan, 2007; 2008), a considerable number of studies have casted scepticism on the validity of u* parameterization in the framework of MOST (Foken, 2006; Holwerda et al., 2012; van Dijk et al., 2015). **It is imperative to mention that $g_A$ is one of the main anchors in the PM-SW model because it not only appears in the numerator and denominator of these models, $g_A$ also provides feedback to $g_C$, aerodynamic temperature, and vapor pressure (seminal paper of Jarvis and McNaughton, 1986).** Therefore, the estimates of E and interception evaporation (Ei) in the PM-SW framework are robustly sensitive to parameterization of $g_A$ and stable E estimates might be possible if $g_A$ estimation is unambiguous (Holwerda et al., 2012; van Dijk et al., 2015). Consequently, our aim was to find analytical solution of $g_A$, and through algebraic reorganization of surface energy balance equation we are able to do so. **Given the lack of consensus in the community on the 'true' $g_A$, we treat STIC1.2 derived non-parametric $g_A$ to be the aerodynamic conductance that satisfies the PM-SW equation for estimating evaporative fluxes.**

(b) **In the state-of-art E modeling, the parametric $g_A$ sub-models are stand alone and empirical**, **and do not provide any feedback to the canopy (or surface) conductances ($g_C$), aerodynamic temperature ($T_0$), and aerodynamic vapor pressure deficit ($D_0$).** However, **$g_A$** is an internal state that provides physical feedback to E and H by influencing **$T_0$, $D_0$, and $g_C$**. Large $g_A$ indicates **small gradients of vapor pressure deficit between the air and canopy boundary layer** and hence strong coupling between canopy and atmosphere (Jarvis and McNaughton, 1986). **These biophysical interactions are entirely overlooked in the land surface parameterizations of $g_A$ (but are included in STIC).** **STIC1.2 consists of a feedback describing the relationship between $T_R$ and E, coupled with canopy-atmosphere components relating E to $T_0$ and $e_0$. The equations are explicitly stated in the Appendix (A2) (eq. A9 to A17) of the manuscript and the detailed descriptions are in L764 to L808.**

(c) **Additional challenges** in grid-scale or spatial-scale $g_A$ estimation are the **requirements of numerous site specific parameters** (e.g., vegetation height, measurement height, vegetation roughness, leaf size, soil roughness) and **coefficients** needed to correct the atmospheric stability conditions (Raupach, 1998). **These informations are required to fulfil the set of assumption established around 1960's**, that can and should be questioned by the community if we want to make science advance in the field of surface energy balance modeling.

(d) The enhanced errors in E estimates in water-limited regions due to uncertain $g_A$ parameterization (Gibson et al., 2011; Timmermans et al., 2013; Morillas et al., 2013; Castellvi et al., 2016) and repeated adjustment of different vegetation as well as soil parameters in the conductance equations to obtain a better E validation (Gokmen et al., 2012) questions the validity of wind driven non-stationary $g_A$ parameterizations. **It solicits for revisiting the state-of-art $g_A$ parameterizations and rethinking to develop a calibration independent $g_A$ modelling framework**.

(e) The credibility of STIC1.2 $g_A$ estimates is shown in the figures (Fig. 1 and Fig. 2). While
Fig. 1a in the manuscript illustrates the differences in the $g_A$ magnitude between forest and
pasture, Fig. 2a displays an independent comparison of STIC-$g_A$ versus u*-based $g_A$. Fig.
2e and 2f showed that $T_R$, vapor pressure deficit ($D_A$), and net available energy ($\phi$)
(difference between net radiation, $R_N$ and ground heat flux, G) can explain 42% to 83%
variability of the u*-based $g_A$. **These correlations and scatterplots between u*-based $g_A$**
**with radiative and meteorological variables clearly emphasize the explanatory power**
**of these variables to characterize wind-driven $g_A$ and the appropriateness of deriving**
**an analytical $g_A$ without wind speed. This also supports the findings of Villani et al.**
**(2003) which stated that during unstable surface layer conditions the major source**
**of net available energy is located at the canopy top and drives the convective motion**
**in the layers above. Hence, the value of $g_A$ is not only controlled by wind speed as**
**advocated by R1.**

(3) STICs is misguided because it introduces a soil moisture stress term that only depends on
atmospheric variables.

Response: R1's claim is not substantiated because the water stress factor was estimated by
combining the radiometric surface temperature ($T_R$) with air temperature ($T_A$), dewpoint
temperature ($T_D$), and near surface dewpoint temperature ($T_{SD}$) as explained in Mallick et al.
(2015).

The methodology is explained in section 2.1 (L205 to L229), section 2.2, and in the
Appendix (A2) of the revised manuscript. In STIC1.2, $T_{SD}$ was estimated in an iterative mode
to establish a feedback between the water stress, $T_R$, $T_{SD}$, and evapotranspiration (explained
in Appendix A2).

(4) The paper then uses Amazonian micrometeorological data to compare a range of gA and
gC terms. No measurements of gC are used to provide verification.

Response: The reasons are explicitly stated is section 2.4 (L337 to L343).

This exercise could not be performed as direct canopy-scale $g_C$ observations are not
possible with current measurement techniques. Although leaf-scale measurements of $g_C$ are
relatively straightforward, these values are not comparable to values retrieved at the canopy-
scale. However, assuming u*-based $g_A$ as baseline aerodynamic conductance, we had
estimated canopy-scale $g_C$ by inverting the PM equation ($g_{C-INV}$) to evaluate $g_{C-STIC}$. The
comparison between $g_{C-STIC}$ and $g_{C-INV}$ over forest and pasture is illustrated in Fig. 3a, and
the results are discussed in L305 to L315 of the earlier manuscript (L425 to L434 of the
revised manuscript).

(5) A large number of plots are then presented where STICS variables are plotted against
meteorological variables in a host of different ways. I am not surprised to see that these
dependencies exist as all of them are intrinsic to the model. Also, because all of them are
interdependent I am not sure how much realism there ultimately is in the findings.

**Summary response:** We do not agree with this statement of R1. Firstly, comparing different
$g_A$ estimates, linking the wind driven $g_A$ estimates with some independent variables (Fig. 2),
and STIC driven $g_A$ estimates with some interdependent variables (Fig. 6 to Fig. 8) is not a
matter of choice, but a necessity, as it evident to any reader. The same is applicable to Fig.

to Fig. 8 for $g_C$. Secondly, despite the transpiration and evaporation estimates are interdependent with $g_C$ and $g_A$ (as shown in Fig. 6 to Fig. 8); the figures reflect the credibility of the conductances as well as transpiration estimates by realistically capturing the hysteretic behavior between biophysical conductances and water vapor fluxes which is frequently observed in natural ecosystems (Zhang et al., 2014, Renner et al., 2016). Fig. 8 (a, b) also affirms that the conductance-transpiration-vapor pressure deficit relationships are compliant with the stomatal feedback-response theory earlier postulated from observational evidences (Monteith, 1995).

**Necessary explanations are included in the revised manuscript (L662 to L672)**

**Detailed response:** Fig. 2 illustrates the diagnostic potential of thermal ($T_R$), radiative ($\phi$),
and meteorological ($D_A$) variables to explain the wind driven $g_A$ variability (wind driven $g_A$ is
independently estimated).

**Fig. 6 and 7** explains the 'hysteresis' between transpiration and conductances which shows
the degree of hysteresis was larger in the dry season than in the wet season. These results
are compliant with the theories earlier postulated from observations that the magnitude of
hysteresis depends on the radiation-vapor pressure deficit lag, while the soil moisture
availability is a key factor modulating the hysteretic transpiration-vapor pressure deficit
relation as soil moisture declines (Zhang et al., 2014; O'Grady et al., 1999; Jarvis and
McNaughton, 1986). **This shows that despite independent of any predefined hysteretic
function, the interdependent conductance-transpiration hysteresis is still captured in
STIC1.2 (which are generally observed in natural ecosystems).**

**Fig. 8 (a and b) confirms the 'stomatal feedback-response' hypothesis as postulated
by Lt. John Monteith (Monteith, 1995)**, which states that a decrease in stomatal
conductance with increasing vapor pressure deficit is caused by a direct increase in
transpiration (Monteith, 1995) and stomata responds to the changes in the air humidity by
sensing transpiration, rather than vapor pressure deficit. This feedback mechanism is found
because of the influence of vapor pressure deficit on both stomatal conductance and
transpiration, which in turn changes vapor pressure deficit by influencing the air humidity
(Monteith, 1995).

**Fig. 8c** shows the complex interaction between $g_C$, radiometric surface temperature ($T_R$) and
vapor pressure deficit ($D_A$). This also answers why different parametric $g_C$ models produce
divergent results.

**Fig. 8d** emphasizes the behavior of $g_A$ according to existing theory that under extremely high
atmospheric turbulence (i.e., high $g_A$), a close coupling exists between the surface and the
atmosphere, which causes $T_R$ and $T_A$ to converge (i.e., $T_R - T_A \rightarrow 0$).

(6) Furthermore STICS assumes that T0 = TR and yet the manuscript does not mention the
potential implications of this assumption, nor the fact that considerable errors can be made
when measuring TR.

> **Summary response:** This comment by R1 is incorrect. **In STIC1.2, $T_0$ is analytically estimated by integrating $T_R$ into a combined PM-SW framework. The analytical expression of $T_0$ is dependent on M and the estimation of M is based on $T_R$ as described in the Appendix (A2) of the current manuscript. $T_0$ is a function of $T_R$ in STIC and they are not assumed equal (section 2.2, L286 to L290).** To further address R1's point on the assumption that $T_0=T_R$, we show here an intercomparison of retrieved $T_0$ versus $T_R$ for forest and pasture (figure below). This indicates the distinct difference of the retrieved $T_0$ from $T_R$ for the two different biomes, which proves that R1's claim to be invalid.
>
> We have included this figure of $T_0$ versus $T_R$ and in the Appendix of revised manuscript (**Fig. A2**). **We have addressed this point explicitly (i.e., $T_0 \neq T_R$) in section 2.2 of the manuscript.**

[Figure]

[Figure]

**Figure:** Aerodynamic temperature obtained from STIC1.2 ($T_{0\text{-STIC}}$) versus radiometric surface temperature ($T_R$) over two different biomes in the Amazon basin. The regression equation of line of best fit is $T_{0\text{-STIC}} = 0.67(\pm 0.10)T_R + 10.59 (\pm 2.79)$ with $r = 0.65$

**Detailed response:** One of the core objectives of the original STIC formulation was to
physically integrate $T_R$ into the PM model to constrain the conductances. **This is done by**
**estimating an aggregated surface moisture availability (or water stress factor) which**
**is an emphatic function of $T_R$.** A detailed description of the STIC state equations is given in
Mallick et al. (2015) and novel part of STIC1.2 is described in the Appendix (A2) of the
revised manuscript.

(7) In the end I feel I have learned little new and what has been presented is tentative and
therefore potentially misleading. This is underlined by sentences (line 325-329) such as
"The evaluation of the conductances and surface energy fluxes indicates some efficacy for
the STIC derived fluxes and conductance estimates ..... As a result we feel some
justification for exploring the canopy-scale biophysical controls on ET and EE generated
through the STIC framework".

Response: We do not agree with the reviewer's impression. The major novelties of the
present manuscript are as follows:

The bafflement about estimating $\lambda E$ originates from complex supply-demand interactions,
where net radiation and soil moisture represents the supply and the atmospheric vapor
pressure deficit represents the demand. This supply-demand interaction accelerates the
biophysical feedbacks in $\lambda E$ and understanding these biophysical feedbacks is necessary to assess the terrestrial biosphere response to water availability. This is now explicitly mentioned in L98 to L102 (also L175 to L188, section 2.1 and section 2.2) of the revised manuscript.

**In this context, the entire manuscript is about understanding the canopy-scale biophysical controls on transpiration and evaporation over the Amazon basin.** The two critical biophysical state variables (i.e., $g_A$ and $g_C$) in the PM equation are the unobserved components which cannot be measured directly. Therefore, we explored the radiative (net radiation and ground heat flux), meteorological (air temperature and relative humidity), and thermal (radiometric surface temperature) information, and developed the STIC framework to analytically estimate these variables in an internally consistent manner (as described in the manuscript). However, before understanding the controls of $g_A$ and $g_C$ on transpiration and evaporation, some indirect evaluation of these two biophysical states was necessary. The sentence is changed as ''The evaluation of the conductances and surface energy fluxes indicates some efficacy for the STIC derived fluxes and conductance estimates which represent a weighted average of these variables over the source area around EC tower''.

**Detailed comments:**

**Line 81-82: "An intensification of the Amazon hydrological cycle was observed in the past two decades characterised by increased temperatures and more frequent droughts and floods" How are increased air (?) temperatures directly linked to hydrological cycle? If it is surface temperatures then say this, but this would mean a decreased ET (hence the floods?)**

Response: Necessary corrections are made in L84 to L85.

**Line 86: "the Amazon forest may become an increasing carbon source". Should this be "increasingly become a net source of carbon?**

Response: Necessary corrections are made in L88 to L91.

**Line 97-104: I disagree with the final point made in this section: GC does not include the conductance relating to bare soil. If you would have called it the surface conductance instead and defined it via the PM Big leaf equation I would have agreed.**

Response: We do not agree and texts are included in L110 to L112 and L236 to L239.

For a dense canopy, $g_C$ in the PM equation represents the canopy surface conductance. Although it is not equal to the canopy stomatal conductance, it contains integrated information of the stomata. For a heterogeneous landscape, $g_C$ in the PM equation is an aggregated surface conductance containing information of canopy and soil.

**Lines 111-126 are stating the obvious. Where is this going?**

Response: These sentences (L127 to L142 in the revised manuscript), and explained the unresolved challenges and problems associated with $g_A$ and $g_C$ parameterisations. **If these are obvious, R1's previous claims on $g_A$ appear to be unfounded.**

These statements are needed to recognize the need of a non-parametric $g_A$ and $g_C$ modeling framework.

**Line 136: Why is the partitioning between soil evaporation and transpiration deemed so important in the Amazon? Soil evaporation must only make up a small part of total ET. Will this soil term affect flooding, atmospheric circulation etc. I highly doubt this.**

Response: **We intended to address 'evaporation', not 'soil evaporation' (L152 in the revised manuscript)**. In the Amazon forest, although the soil evaporation has negligible contribution, it is the interception evaporation that has substantial contribution in the total evaporative fluxes, and, therefore the partitioning of 'evaporation ($\lambda E_E$)' and 'transpiration ($\lambda E_T$)' is significant.

**Line 141-143: "Given the persistent risk of deforestation, the ecophysiological changes of different plant functional types (PFTs) are expected to be reflected in gA and gC and EE and ET". I really do not understand what is meant by this sentence.**

Response: This is now L157 to L161 and necessary changes are incorporated.

The persistent risk of deforestation is likely to alter the radiation interception, surface temperature, surface moisture, associated meteorological conditions, and vegetation biophysical states of different plant functional types (PFTs). Conversion from forest to pasture is expected to change the gC/gA ratio of these ecosystems and impact the evapotranspiration components.

**Line 154-157: The surface temperature is already implicit in the PM equation as it combines the energy balance with bulk transfer equations.**

Response: The surface temperature ($T_R$) was eliminated from the derivation of the PM equation by expressing the slope of the saturation vapor pressure at ambient air temperature. However, in the seminal paper titled 'Evaporation and Surface Temperature' (Monteith, 1981), Lt. John Monteith described the role of leaf temperature in constraining the biophysical conductances. Although $T_R$ is implicit in the net radiation (Rn), which appears in the numerator of the PM equation, it may be noted that Rn has a relatively weak dependence on $T_R$ (compared to $T_R$ sensitivities of soil moisture and E). No universally agreed formulation is available that physically constrains $g_A$ and $g_C$ by using $T_R$. Development of STIC is based on the assumption that the intrinsic link between the PM-SW model and $T_R$ emanates through the first-order dependence of the biophysical conductances on aerodynamic temperature ($T_0$) and soil moisture (through $T_R$). Hence, the conductances are explicitly constrained by using $T_R$ information as described in the manuscript.

**Detailed explanations are given in L213 to L229 and in section 2.2.**

**Line 179-181: "The retrieval of gA, gC, and E are based on finding a 'closure' of the PM equation using the STIC framework". In my opinion, the PM is already closed, see my point above. It calculates ET from Rn-G, and H is implicitly in there. Please study books such as those by Hamlyn Jones to see how PM equation is derived.**

Response: The PM equation is 'closed' upon the availability of canopy-scale measurements of the two unobserved biophysical conductances ($g_A$ and $g_C$) and if we assume the empirical models of $g_A$ and $g_C$ to be reliable. However, neither $g_A$ nor $g_C$ can be measured at the canopy-scale or at larger spatial scales. Furthermore, as shown by several recent studies (Matheny et al., 2014; van Dijk et al., 2015) a most appropriate or correct $g_A$-$g_C$ model is currently not available. This implies that **a true 'closure' of the PM equation is only**
**possible upon analytical estimation of the conductances. Necessary explanations are**
**given in L242 to L248.**

**Line 184: This should be 'radiative temperature'.**

Response: Necessary correction is made (L210)**.**

**Line 203: You have now tacitly assumed that T0 = TR. There is a host of literature**
**references that will tell you otherwise.**

Response: The explanation is already provided above; there is no assumption on the
equality between $T_R$ and $T_0$. **We have addressed this point explicitly (i.e., $T_0 \neq T_R$) in**
**section 2.2 of the manuscript.**

**Line 204-205: PM equation is already closed. This assumption of energy balance**
**closure is implicit in its derivation. But maybe I do not understand what you mean by**
**this statement.**

Response: As mentioned earlier, the PM equation is closed if measurements of the two
unobserved biophysical conductances ($g_A$ and $g_C$) are available. However, $g_A$ and $g_C$ cannot
be directly measured at the canopy-scale and there is no universally agreed $g_A$ and $g_C$
model. By the term 'closure', we mean **actual 'closure'** of the PM equation by finding
analytical solutions of $g_A$ and $g_C$. This was done by solving 'n' equations and 'n' unknowns as
described in equation 2 to 5 in the manuscript. The derivation of these equations are
explained in the Appendix (A1) of the revised manuscript.

**Line 225-227: "The estimates of EE in the current method consists of aggregated**
**contribution from both interception and soil evaporation, and no further attempt is**
**made to separate these two components". This is a considerable weakness in the**
**approach seeing leaf area index and hence interception is so large for large parts of**
**the Amazon and soil evaporation will be negligible. You are making this point yourself**
**a few sentences later (line 232) Also: these two types of evaporation fluxes take place**
**at very different source heights, so their GA will be very different, further weakening**
**your approach.**

**Response: We do not agree. This is not a considerable weakness, but a fact which is**
**clearly stated instead of withholding it.** At the outset, the biophysical controls on
evaporation and transpiration are mentioned, and no claim is made on understanding soil
evaporation, interception evaporation etc.

We agree that different $g_A$ exists for soil-canopy, sun-shade, and dry-wet conditions; which is
currently integrated into a lumped $g_A$ (given the big-leaf nature of STIC). From the big-leaf
perspective, it is generally assumed that the aerodynamic conductance of water vapor and
heat are equal (Raupach, 1998). However, for obtaining partitioned aerodynamic
conductances, explicit partitioning of evapotranspiration is needed, which is beyond the
scope of the current manuscript. This is mentioned in L315 to L320 (section 2.3) of the
revised manuscript.

**Line 285: "The conductances showed a marked diurnal variation expressing their**
**overall dependence on net radiation, vapor pressure deficit, and surface**

**temperature". What conductance are you referring to here? gA or gC? Or both? Note that gA generally does not depend on net radiation or VPD etc., although it does in STICS.**

**Response: Here, we are referring to both $g_A$ and $g_C$ as clearly stated in Fig. 1 and the related descriptions as stated in L399 to L408.**

**The role of $g_A$ is associated with the role of convection (Choudhury and Monteith, 1986) according to the surface energy balance principle as follows.**

Neglecting horizontal advection and energy storage, the surface energy balance equation is written as follows:

$$\phi = \lambda E + H \tag{1}$$

Where $\phi \cong R_N - G$, with $R_N$ being net radiation, and G being the conductive surface heat flux or ground heat flux, H is the sensible heat flux and $\lambda E$ is the latent heat flux.

The sensible and latent heat flux can be expressed in the form of aerodynamic transfer equations (Boegh et al., 2002; Boegh and Soegaard, 2004) as follows:

$$H = \rho c_P g_A (T_o - T_A) \tag{2}$$

$$\lambda E = \frac{\rho c_P}{\gamma} g_A (e_0 - e_A) = \frac{\rho c_P}{\gamma} g_C (e_0^* - e_0) \tag{3}$$

Where $T_A$ is the air temperature at the reference height ($z_R$), $e_A$ is the atmospheric vapor pressure (hPa) at the level at which $T_A$ is measured, $e_0$ and $T_0$ are the atmospheric vapor pressure and air temperature at the source/sink height, or at the so-called roughness length ($z_0$), where wind speed is zero. They represent the vapor pressure and temperature of the quasi-laminar boundary layer in the immediate vicinity of the surface level (Fig. A1), and $T_0$ can be obtained by extrapolating the logarithmic profile of $T_A$ down to $z_0$. $e_0^*$ is the saturation vapor pressure at $T_0$ (hPa).

By combining eq. 1, 2, and 3 and solving for $g_A$, we get

$$g_A = \frac{\phi}{\rho c_P \left[ (T_o - T_A) + \left( \frac{e_0 - e_A}{\gamma} \right) \right]} \tag{4}$$

**Equation 4 clearly portrays the dependency of $g_A$ on net available energy and vapor pressure.**

Given R1's disposition on the wind speed dependent empirical $g_A$ models based on the Monin-Obukhov Similarity Theory (MOST), it is important to mention that the Monin-Obukhov Length (L) is a function of evapotranspiration (E) (Brutsaert, 1982), and E is strongly dependent on the net available energy as well as vapor pressure deficit. The functions below describes the dependence of $g_A$ on net available energy ($\phi$) (= net radiation – ground heat flux) and vapor pressure deficit in addition to $T_0-T_A$, despite $g_A$ being generally estimated from wind speed information.

$$g_A = f\{L\} \tag{5}$$

$$L = \frac{u^* \rho C_P T_A}{g(H + 0.61 C_P T_A E)} \tag{6}$$

$$u^* = f\{L, E, specific\ humidity\ gradieant, wind\ speed\} \tag{7}$$

$$E = f\{R_N, D_A, soil\ moisture, T_R\} \tag{8}$$

Here $u^*$ is the friction velocity (m s$^{-1}$), $\rho$ is the air density (kg m$^{-3}$), $c_P$ is the specific heat of air
(1004 j kg$^{-1}$ K$^{-1}$), $T_A$ is the air temperature (K), $D_A$ is the vapor pressure deficit (hPa). Rest all
the variables are explained earlier.

According to equations 5 to 8, the dependence of $g_A$ on net radiation and $D_A$ is obvious.
Wind is generated as a result of the differences in atmospheric pressure which is a result of
uneven surface radiative heating. Therefore, the aerodynamic conductance (and wind as
well) is an effect of net radiative heating and therefore, there should be a physical
relationship between these two.

**Necessary explanations are given in Table (A2).**

 **Response to Review of HESSD-552-2015 by Anonymous Reviewer #2**

This manuscript describes a study that infers stomatal and aerodynamic conductances from
eddy flux observations. I think in general, this study is innovative and presents novel
material, so that in principle it should be published. I hesitate recommendation for publication
mostly because I am not entirely convinced by the approach and I feel that this needs
revision. Hence, I recommend major revisions, although I do not think that it necessarily
involves a lot of work to address the points below.

Response: We thank R2 for the encouraging comments and for appreciating the novelty of
the aerodynamic and canopy conductance ($g_A$ and $g_C$) retrieval to assess their controls on
evaporation and transpiration. We appreciate the valuable suggestions which will further
improve the manuscript.

Major points:

(1) My major problem with the manuscript is that I do not understand the approach, so that it is
difficult to assess its plausibility. While the main equations are provided in the manuscript
(eqn. 2-5), there is no more description on where these equations come from, except for
references to prior papers by the authors. I think it is necessary to at least provide a
description at a qualitative level where these equations come from.
Response: We agree and included the detailed derivations of the 'state equations' (eqn. 2 to
5) in the Appendix (A1) of the revised manuscript.

(2) The point where I really got confused is that eqn. 5 uses the Priestley-Taylor coefficient,
which is an empirical coefficient in an evaporation equation that is rather different from the
Penman Monteith equation. Where does this coefficient suddenly come from? I find this
quite confusing, and it needs at least a minimum of explanation as it is not obvious.
Response: Good point indeed and we apologize for the confusion. This description is made
explicit in the revised version of the manuscript (L279 to L283).
From the derivation of the equation S10 (described in Supplement in the manuscript), it is
apparent that the Priestley-Taylor coefficient ($\alpha$) appeared due to the use of the Advection-
Aridity hypothesis for deriving the state equation of the evaporative fraction. However,
instead of assuming $\alpha$ as a 'fixed parameter', we have developed a physical equation of $\alpha$
(eqn. A15 in the manuscript) and numerically estimated $\alpha$ as a 'variable'. **The derivation of**
**the equation for $\alpha$ is described in the** Appendix A2 of the manuscript in L780 to L782.

(3) What I also do not understand is why an iterative scheme is needed.
Response: The analytical solution to the 'state' equations (eqn. 2 – 5 in the manuscript) have
four accompanying unknowns; M (surface moisture availability), $e_0$ (vapor pressure at the
source/sink height), $e_0^*$ (saturation vapor pressure at the source/sink height), and $\alpha$, and as
a result there are 4 equations with 8 unknowns. Consequently an iterative solution is needed to determine the four unknown variables as stated in L265 to L271 of the revised manuscript
and described in the Appendix A2.

(4) Can't one simply use the observations and use a simple partitioning based on the
Bowen ratio?

Response: Here we intended to partition evapotranspiration into component water fluxes.
Although the Bowen ratio (Bowen, 1926) is an energy partitioning ratio to understand the
relative apportioning between sensible and latent heat flux, it is not relevant for the latent
heat flux partitioning into transpiration and evaporation. In this context an aggregated
surface moisture availability (or water stress factor) is a better metric for dry-wet latent heat
flux partitioning and we used the retrieved surface moisture availability (M) for partitioning of
the latent heat flux.

(5) It would be good to describe what the differences and similarities are to previous
approaches. As the authors propose a new approach, they should provide a better
description that is easier to follow of what is being done.

Response: We assume R2 is intending to the differences of STIC with other approaches that
earlier attempted to understand the biophysical controls of evapotranspiration, which is
described in Table (A2).

If R2 is intending the differences between STIC1.2 with other previous STIC versions, we
included Table (A1) in the appendix to describe the fundamental differences between
STIC1.0, STIC1.1, and STIC1.2.

**Minor points:**

- The authors refer to $\lambda E$ as evaporation, which, technically speaking, is the latent heat flux,
not evaporation.

Response: Necessary corrections are made (L37 to L38, L106 to L107) in the revised
manuscript.

- Abstract: dry and wet conditions $\lambda E_T$, do you mean conditions in which water is not limiting
vs. limiting, or precipitation vs. radiation driven conditions?

Response: It is the precipitation vs. radiation driven conditions and we have clarified this in
the abstract (L47 to L52) of the revised manuscript.

- Biophysical control of $\lambda E_T$ should be briefly explained by what this means.

Response: At large $g_A/g_C$, the vapor pressure deficit close to the canopy source/sink height
($D_0$) approximates the atmospheric vapor pressure deficit ($D_A$) due to aerodynamic mixing
and/or low transpiration. This results in a strong canopy-atmosphere coupling and such
condition is prevalent under soil moisture deficient conditions. On the contrary, large $g_C$
influences the gradients of vapor pressure deficit just above the canopy, such that $D_0$ tend
towards zero and thus remains different from $D_A$ (Jarvis and McNaughton, 1986). This situation reflects a weak canopy-atmosphere coupling and such situation prevails under
predominantly wet conditions and/or poor aerodynamic mixing due to wetness induced low
aerodynamic roughness.
We have included this description in the introduction (L112 to L120) of the revised
manuscript. Additionally, section 2.5 also described the details about biophysical controls.

- Line 145: I wonder why approaches that directly link stomatal conductance to
photosynthesis are not mentioned, such as Ball-Berry?

Response: We have included references to photosynthesis-dependent stomatal
conductance models in the revised manuscript (L162 to L163).
- Line 194: Where do these "state equations" come from? Referring to previously published
work is fine for derivations, but the description should still mention what the concepts are
that are behind these equations.
Response: We have included a detailed description of the derivation of the 'state equations'
in the Appendix (A1) of the revised manuscript.
- A table of variables would help.
Response: A table of variables has been included (Table 1).
- Line 238: I think the authors assume that the conductances to momentum, sensible and
latent heat are identical. If this is the case, it should be mentioned, as there are also
approaches to surface exchange that do not treat them as being identical.
Response: Yes, the conductances of momentum for the sensible and latent heat flux are
assumed identical as mentioned in the revised manuscript (L329 to L330).

- Line 331: As the typical readers of HESS are not micrometeorologists, it would be useful to
explain the decoupling coefficient in some more detail. This will help to interpret the following
results.
Response: A detailed description of the decoupling coefficient is now included in section 2.5.
The decoupling coefficient or 'Omega' ($\Omega$) is a dimensionless coefficient ranging from 0.0 to
1.0 (Jarvis and McNaughton, 1986) and considered as an index of the degree of stomatal
control on transpiration. The equation of $\Omega$ is as follows:

$$\Omega = \frac{\frac{s}{\gamma} + 1}{\frac{s}{\gamma} + 1 + \frac{g_A}{g_C}}$$

Introducing $\Omega$ in the Penman-Monteith (PM) equation for $\lambda E$ results in:

$$\lambda E = \Omega \lambda E_{eq} + (1 - \Omega)\lambda E_{imp}$$

$$\lambda E_{eq} = \frac{s\phi}{s + \gamma}$$

$$\lambda E_{imp} = \frac{\rho c_P}{\gamma} g_C D_A$$

Where, $\lambda E_{eq}$ is the equilibrium evapotranspiration, which depends only on the net available energy and would be obtained over an extensive surface of uniform moisture availability (Jarvis and McNaughton, 1986; Kumagai et al., 2004). $\lambda E_{imp}$ is the imposed evapotranspiration, which is 'imposed' by the atmosphere on the vegetation surface through the effects of vapor pressure deficit (triggered under limited soil moisture availability) and evapotranspiration is proportional to $g_C$.

When the $g_C/g_A$ ratio is very small (i.e., water stressed conditions), stomata principally control the water loss and a change in $g_C$ will result in a nearly proportional change in transpiration. In this case the $\Omega$ value approaches zero, and vegetation is believed to be fully coupled to the atmosphere. In contrast, for a high $g_C/g_A$ ratio (i.e., water unstressed conditions), changes in $g_C$ will have little effect on the transpiration rate, and transpiration is predominantly controlled by the net available radiative energy. In this case the $\Omega$ value approaches unity, and vegetation is considered to be poorly coupled to the atmosphere.

- Line 422: To what extent could these discrepancies between how conductances are derived also relate to actual differences in the conductances for momentum vs. heat?

Response: This is indeed a good point addressed by R2 (although beyond the scope of this manuscript) and is clarified in the revised manuscript (L537 to L546).

- Line 498: The authors should stick to the same ratio gA/gC for easier interpretation.

Response: In the entire manuscript we maintain gC/gA ratio for easier interpretation, uniformity and also for the clarity to the reader.

Additional references:

Batra, N., Islam, S., Venturini, V., Bisht, G., and Jiang, L.: Estimation and comparison of evapotranspiration from MODIS and AVHRR sensors for clear sky days over the southern great plains, Remote Sens. Environ., 103, 1–15, 2006.

Brutsaert, W.: Evaporation Into the Atmosphere, Reidel Pub. Comp., Dordrecht, Holland, 299 pp, 1982.

Castellví, F., Cammalleri, C., Ciraolo, G., Maltese, A. and Rossi, F.: Daytime sensible heat flux estimation over heterogeneous surfaces using multitemporal land-surface temperature observations, Water Resour. Res., doi:10.1002/2015WR017587, 2016 (in press).

Gokmen, M., et al.: Integration of soil moisture in SEBS for improving evapotranspiration estimation under water stress conditions, Remote Sens. Environ., 121, 261–274, 2012.

Harman, I.N., Finnigan, J.J.: Scalar concentration profiles in the canopy and roughness sublayer, Bound. Layer Meteorol. 129 (3), 323–351, 2008.

Harman, I.N., Finnigan, J.J., 2007. A simple unified theory for flow in the canopy and roughness sublayer. Bound.-Layer Meteorol. 123 (2), 339–363.

Hong, J. and Kim, J.: Numerical study of surface energy partitioning on the Tibetan plateau:
comparative analysis of two biosphere models, Biogeosciences, 7, 557-568,
doi:10.5194/bg-7-557-2010, 2010.

Jiang, L., and Islam, S.: Estimation of surface evaporation map over Southern Great Plains
using remote sensing data, Water Resour. Res., 37 (2), 329–340, 2001.

Kleidon, A., Renner, M., and Porada, P.: Estimates of the climatological land surface energy
and water balance derived from maximum convective power, Hydrol. Earth Syst. Sci., 18,
2201-2218, 2014.

Morillas, L., García, M., Nieto, H., Villagarcia, L., Sandholt, I., Gonzalez-Dugo, M.P., Zarco-
Tejada, P.J., Domingo, F.: Using radiometric surface temperature for energy flux
estimation in Mediterranean drylands from a two-source perspective, Remote Sens.
Environ., 136, 234 – 246, 2013.

Prihodko, L., Denning, A.S., Hanan, N.P., Baker, I.T., and Davis, K.: Sensitivity, uncertainty
and time dependence of parameters in a complex land surface model, Agric. For.
Meteorol., 148 (2), 268–287, 2008.

Schulz, K., Beven, K.J.: Data-supported robust parameterisations in land surface-
atmosphere flux predictions: towards a top–down approach, Hydrol. Process. 17, 2259–
2277, 2003.

Timmermans, J., Su, Z., van der Tol, C., Verhoef, A., and Verhoef, W.: Quantifying the
uncertainty in estimates of surface–atmosphere fluxes through joint evaluation of the
SEBS and SCOPE models, Hydrol. Earth Syst. Sci., 17, 1561-1573, doi:10.5194/hess-17-
1561-2013, 2013.

[revised manuscript text omitted]

---

## Referee Report (RR1)

Canopy-scale biophysical controls of transpiration and evaporation in the Amazon Basin

by Mallick et al., Revised Version

I believe that the authors benefit from reviewers' comments and address them satisfactorily. Definitely, inclusion of variables in Table 1, derivation of equations and more detailed information about Omega Theory improved readability and are useful for readers of HESS. I found some minor points to improve the readability of the paper.

This is a humble recommendation to the authors, they may be benefit from Zuecco et al. (2016) about the magnitude of hysteresis (L667) for their future study. I agree with the authors about their comparison underlining area or size of the hysteresis (L494-5). However, if they want to parameterize the magnitude of hysteresis, it may be good to check this aforementioned study for their future study.

> Zuecco et al., 2016. A versatile index to characterize hysteresis between hydrological variables at the runoff event timescale, *Hydrological Processes*, 30, 1449-1466.

Minor Comments:

L99. Delete 's' after 'represent'. Insert comma after 'supply'.

…net radiation and soil moisture represent the supply, and ….

L111. Insert comma after 'cover'.

L154. Replace 'are' with 'is'. My understanding you are expressing about the investigation.

L205. I recommend plural because of more than one retrieval. The retrievals…

L221. I recommend re-write the parenthesis. My suggestion:

….(compared to the sensitivity of $T_R$ to soil moisture and $\lambda E$).

L223. Insert a dash after 'water'. …water-stress controls…

L268. Ensure unity. Write numbers in words as you did in L393, L399 etc. '4' and '8' should be replaced by four and eight, respectively.

L304. Delete 's'. Plural verb. …consist of…

L367. I recommend 'was' instead of 'is'. It depends on authors.

L375. Word choice. Replace 'from' with 'during'. …during 1995-2005….

L383. I recommend taking $T_R$ outside the radiation properties. Pay attention to L550, you use TR as a thermal variable. So, it will be good to take it out from the radiation variables. It depends on authors. My recommendation:

…radiation ($R_N$, shortwave and longwave), thermal ($T_R$), meteorological…

L401. I think 'over' should be replaced with 'for' as seen in L403.

L404. Delete 'that'. … less than half of those….

L424. Insert 'respectively' after '$g_{A\text{-}BM13}$' …….$g_{A\text{-}BM13}$, respectively.

L439. Insert 'respectively' after 'fluxes' …..…fluxes, respectively (Fig.4).

L468. …relativeLY higher….

L469. I may be confused here due to 'of'. My understanding which defines 'reasons', so is should be are. Please check the sentence.

L555. I think 'is' should be 'are'. … the accuracies….ARE limited….

L588. 'results' should be 'result'.     ….changes …. RESULT in….

L622. I think 'is' should be 'are'. My understanding, 'which' refers to forests, $g_A$ of forests are higher than that of pastures.

L626. I recommend plural form of 'forest' due to using plural form of 'pasture'. …forestS than in the pastures.

L679. Insert comma after 'model'.

L776. Replace 'is' with 'are'.   ……differences…..ARE….

L778. I recommend using (SW) instead of SW85. You used throughout the paper PM-SW.

L791. Singular verb. …tendS to….

L1130. Table 2. Please insert degree sign (°) beneath latitude and longitude. Please define in the caption, (-) refers to (S) and (W) for latitude and longitude, respectively.

S1P2L3. Introduced by Mallick et al (2015).

S1P2L5. 'was' should be 'were'. ….effects…..WERE…

---

## Author Response (AR2)

**1 Reviewer 4 (R4):**

I believe that the authors benefit from reviewers' comments and address them satisfactorily.
Definitely, inclusion of variables in Table 1, derivation of equations and more detailed

4 information about Omega Theory improved readability and are useful for readers of HESS. I

5 found some minor points to improve the readability of the paper.

6 Response: We thank R4 for the detailed reading and comments to improve the manuscript.

This is a humble recommendation to the authors, they may be benefit from Zuecco et al.
(2016) about the magnitude of hysteresis (L667) for their future study. I agree with the
authors about their comparison underlining area or size of the hysteresis (L494-5). However,
if they want to parameterize the magnitude of hysteresis, it may be good to check this
aforementioned study for their future study.

Response: We have included the reference of Zuecco et al. (2016) in the revisedmanuscript.

14 Minor Comments:

L99. Delete 's' after 'represent'. Insert comma after 'supply'....net radiation and soil moisture represent the supply, and ....

- 17 Response: 's' is deleted now.
- 18 L111. Insert comma after 'cover'.
- 19 Response: A comma is inserted.
- L154. Replace 'are' with 'is'. My understanding you are expressing about the investigation.
- 21 Response: 'Are' is replaced with 'is'.
- 22 L205. I recommend plural because of more than one retrieval. The retrievals...
- 23 Response: Done, as suggested.

L221. I recommend re-write the parenthesis. My suggestion:....(compared to the sensitivity of TR to soil moisture and  $\lambda E$ ).

- 26 Response: Response: Necessary changes are made.
- 27 L223. Insert a dash after 'water'. ...water-stress controls...
- 28 Response: a dash is inserted.
- L268. Ensure unity. Write numbers in words as you did in L393, L399 etc. '4' and '8' should
  be replaced by four and eight, respectively.
- 31 Response: Necessary changes are made.
- 32 L304. Delete 's'. Plural verb. ...consist of...
- 33 Response: 's' deleted.

- 34 L367. I recommend 'was' instead of 'is'. It depends on authors.
- 35 Response: 'is' replaced by 'was'.
- 36 L375. Word choice. Replace 'from' with 'during'. ...during 1995-2005....

37 Response: 'from' is replaced with 'during'.

L383. I recommend taking TR outside the radiation properties. Pay attention to L550, you use TR as a thermal variable. So, it will be good to take it out from the radiation variables. It depends on authors. My recommendation:

- 41 ...radiation (RN, shortwave and longwave), thermal (TR), meteorological...
- 42 Response: Necessary changes are made.
- 43 L401. I think 'over' should be replaced with 'for' as seen in L403.
- 44 Response: 'over' is replaced with 'for'.
- 45 L404. Delete 'that'. ... less than half of those....
- 46 Response: 'that' is deleted.
- 47 L424. Insert 'respectively' after 'gA-BM13' ......gA-BM13, respectively.
- 48 Response: 'respectively' is inserted in appropriate place.
- 49 L439. Insert 'respectively' after 'fluxes' ......fluxes, respectively (Fig.4).
- 50 Response: 'respectively' is inserted in appropriate place.
- 51 L468. ...relativeLY higher....
- 52 Response: Corrected now.
- L469. I may be confused here due to 'of'. My understanding which defines 'reasons', so isshould be are. Please check the sentence.
- 55 Response: The sentence is modified as...

'Interestingly, coupling was relatively higher in pasture during the dry seasons and thereasons are detailed in the following section and discussion.'

- 58 L555. I think 'is' should be 'are'. ... the accuracies....ARE limited....
- 59 Response: 'is' replaced with 'are'.
- 60 L588. 'results' should be 'result'. ....changes .... RESULT in....
- 61 Response: The sentence is corrected as,.....
- 62 'Here, fractional change in  $g_C$  results in an equivalent fractional change in  $\lambda E_T$ .'

- L622. I think 'is' should be 'are'. My understanding, 'which' refers to forests, gA of forests are
   higher than that of pastures.
- Response: Here 'is' seems to be appropriate. Here, 'which' refers to the coupling.
- L626. I recommend plural form of 'forest' due to using plural form of 'pasture'. ...forestS thanin the pastures.
- 68 Response: Necessary correction is made.
- 69 L679. Insert comma after 'model'.
- 70 Response: A comma is inserted.
- 71 L776. Replace 'is' with 'are'. .....differences.....ARE....
- 72 Response: 'is' replaced by 'are'.
- 73 L778. I recommend using (SW) instead of SW85. You used throughout the paper PM-SW.
- 74 Response: 'SW85' replace by 'SW'.
- 75 L791. Singular verb. ...tendS to....
- 76 Response: Necessary correction is made.
- 77 L1130. Table 2. Please insert degree sign (°) beneath latitude and longitude. Please define
- in the caption, (-) refers to (S) and (W) for latitude and longitude, respectively.
- 79 Response: Necessary correction is made.
- 80 S1P2L3. Introduced by Mallick et al (2015).
- 81 Response: Necessary correction is made.
- 82 S1P2L5. 'was' should be 'were'. ....effects.....WERE...
- 83 Response: 'was' replaced by 'were'.
- 84
- 85
- 86
- 00
- 87
- 88
- 89
- 90
- 91
- 92

**93 Canopy-scale biophysical controls of transpiration and 94 evaporation in the Amazon Basin**

Kaniska Mallick1, Ivonne Trebs1, Eva Boegh2, Laura Giustarini1, Martin Schlerf1, Darren T.
Drewry3, Lucien Hoffmann1, Celso von Randow4, Bart Kruijt5, Alessandro Araùjo6, Scott Saleska7, James R. Ehleringer8, Tomas F. Domingues9, Jean Pierre H. B. Ometto4, Antonio D. Nobre4, Osvaldo Luiz Leal de Moraes10, Matthew Hayek11, J. William Munger11, Steve Wofsy11

[revised manuscript text omitted]
         | $R^2$   | Slope               | Offset         | Ν    | RMSD                         | $\mathbf{R}^2$ | Slope    | Offset       |
|      | $(m s^{-1})$ |         | •                   | $(m s^{-1})$   |      | $(m s^{-1})$                 |                |          | $(m s^{-1})$ |
| TRF  | 0.013        | 0.41    | 1.07                | 0.0031         | 1159 | 0.012                        | 0.14           | 0.39     | 0.0097       |
|      |              | (±0.03) | $(\pm 0.047)$       | $(\pm 0.0008)$ |      |                              | $(\pm 0.04)$   | (±0.039) | (±0.0007)    |
| TMF  | 0.012        | 0.55    | 0.81                | 0.0006         | 1927 | 0.009                        | 0.55           | 0.85     | 0.0032       |
|      |              | (±0.12) | (±0.023)            | (±0.0006)      |      |                              | (±0.12)        | (±0.025) | (±0.0005)    |
| TDF  | 0.007        | 0.49    | 0.89                | 0.0019         | 787  | 0.012                        | 0.33           | 0.30     | 0.0050       |
|      |              | (±0.15) | (±0.041)            | (±0.0006)      |      |                              | (±0.19)        | (±0.022) | (±0.0005)    |
| PAS  | 0.012        | 0.22    | 1.03                | 0.0059         | 288  | 0.007                        | 0.58           | 0.65     | 0.0024       |
|      |              | (±0.18) | (±0.083)            | (±0.0007)      |      |                              | (±0.12)        | (±0.025) | (±0.0003)    |
| Mean | 0.012        | 0.44    | 0.76                | 0.0047         | 4161 | 0.010                        | 0.39           | 0.63     | 0.0046       |
|      |              | (±0.10) | (±0.016)            | (±0.003)       |      |                              | $(\pm 0.08)$   | (±0.016) | (±0.0003)    |

1261 N = number of data points; RMSD = root mean square deviation between predicted (P) and observed (O) 1262 variables =  $\left[\frac{1}{N}\sum_{i=0}^{N}(P_i - O_i)^2\right]^2$ .

1263

1264

**Table 4**: Comparative statistics for the STIC and tower-derived hourly  $\lambda E$  and *H* for a range of PFTs in the Amazon Basin (LBA tower sites). Values in parenthesis are ±one standard deviation (standard error for correlation).

| DET. |              | 11             |               |              |              |          | 11       |              |      |
|------|--------------|----------------|---------------|--------------|--------------|----------|----------|--------------|------|
| PFIS |              | ΛI             | 5             |              | Н            |          |          |              |      |
|      | RMSD         | $\mathbb{R}^2$ | Slope         | Offset       | RMSD         | $R^2$    | Slope    | Offset       | N    |
|      | $(W m^{-2})$ |                |               | $(W m^{-2})$ | $(W m^{-2})$ |          |          | $(W m^{-2})$ |      |
| TRF  | 28           | 0.96           | 1.10          | -16          | 34           | 0.52     | 0.60     | 29           | 1159 |
|      |              | (±0.007)       | $(\pm 0.008)$ | (±2)         |              | (±0.030) | (±0.025) | (±2)         |      |
| TMF  | 20           | 0.98           | 1.08          | -11          | 23           | 0.71     | 0.61     | 20           | 1927 |
|      |              | $(\pm 0.004)$  | $(\pm 0.004)$ | (±1)         |              | (±0.019) | (±0.014) | (±1)         |      |
| TDF  | 26           | 0.96           | 0.96          | -7           | 30           | 0.66     | 0.89     | 20           | 787  |
|      |              | (±0.009)       | $(\pm 0.008)$ | (±2)         |              | (±0.032) | (±0.035) | (±3)         |      |
| PAS  | 31           | 0.96           | 1.14          | -2           | 33           | 0.88     | 0.67     | 9            | 288  |
|      |              | (±0.009)       | (±0.010)      | (±2)         |              | (±0.016) | (±0.011) | (±1)         |      |
| Mean | 33           | 0.94           | 1.04          | -1           | 37           | 0.61     | 0.58     | 24 (±2)      | 4161 |
|      |              | (±0.005)       | (±0.005)      | (±1)         |              | (±0.021) | (±0.009) |              |      |

1268

1269

1270

1271

**Figure 1**. Examples of monthly averages of the diurnal time series of canopy-scale (a)  $g_A$  and (b)  $g_C$ estimated for two different biomes (forest and pasture) in the Amazon Basin (LBA sites K34 and FNS). The time series of four different  $g_A$  estimates and their corresponding  $g_C$  estimates are shown here.